psychology

COVID-19, risk communication, risk perception

**Author for correspondence:**
Alexandra L. J. Freeman
e-mail: alex.freeman@maths.cam.ac.uk

# Communicating personalized risks from COVID-19: guidelines from an empirical study

Alexandra L. J. Freeman[1], John Kerr[1,2], Gabriel Recchia[1], Claudia R. Schneider[1,2], Alice C. E. Lawrence[1], Leila Finikarides[1], Giulia Luoni[1], Sarah Dryhurst[1,2] and David Spiegelhalter[1]

[1]Winton Centre for Risk and Evidence Communication, Department of Pure Mathematics and Mathematical Statistics, and [2]Department of Psychology, University of Cambridge, Cambridge, UK

ALJF, 0000-0002-4115-161X; JK, 0000-0002-6606-5507;
GR, 0000-0002-0210-8635; CRS, 0000-0002-6612-5186;
ACEL, 0000-0002-8771-4844; LF, 0000-0002-0414-3207;
GL, 0000-0003-3582-5953; SD, 0000-0002-7772-8492;
DS, 0000-0001-9350-6745

As increasing amounts of data accumulate on the effects of the novel coronavirus SARS-CoV-2 and the risk factors that lead to poor outcomes, it is possible to produce personalized estimates of the risks faced by groups of people with different characteristics. The challenge of how to communicate these then becomes apparent. Based on empirical work (total $n = 5520$, UK) supported by in-person interviews with the public and physicians, we make recommendations on the presentation of such information. These include: using predominantly percentages when communicating the absolute risk, but also providing, for balance, a format which conveys a contrasting (higher) perception of risk (expected frequency out of 10 000); using a visual linear scale cut at an appropriate point to illustrate the maximum risk, explained through an illustrative 'persona' who might face that highest level of risk; and providing context to the absolute risk through presenting a range of other 'personas' illustrating people who would face risks of a wide range of different levels. These 'personas' should have their major risk factors (age, existing health conditions) described. By contrast, giving people absolute likelihoods of other risks they face in an attempt to add context was considered less helpful. We note that observed effect sizes generally were small. However, even small effects are meaningful and relevant when scaled up to population levels.

# 1. Introduction

When a new threat emerges, such as that presented by the SARS-CoV-2 virus, it is very difficult for individuals to assess the risk it poses to them personally: how likely they are to be affected by it, and how severely. These two aspects are important components of a person's 'risk perception'—a subjective feeling—which also incorporates emotional components such as worry. The emotional components of risk perception are affected by feelings of control, knowledge about the risk and other aspects, which can cumulatively make a risk be perceived as a 'dread risk' [1,2]. Risk perceptions are key drivers of behaviour, which can be protective [3–5] but high-risk perceptions can lead to worry, anxiety and behaviours whose harms may be greater than their benefits [6–8], and low ones to inadequate protective behaviours, which has both individual and societal consequences. Providing information which can influence people's perception of a risk, then, has to be done with care and be based on an understanding of what effect it is likely to have.

As countries accumulate more data on mortality and hospitalization rates from COVID-19, as well as the proportions who suffer long-term effects, it is possible to produce increasingly personalized risk calculators (e.g. [9]). The issue, then, is how to communicate this potentially highly emotional information, which may challenge people's prior perceptions about the risk, triggering identity-protective cognition (e.g. [10]), to diverse audiences—and what effect different presentations of such a risk are likely to have.

Risks from COVID-19 fall into one of the more difficult areas to communicate: the thought of the disease can provoke strong emotion (or 'dread' [1]) which is known to affect risk perceptions [11–14]; for many people, the magnitude of the risk is low (less than 0.1% chance of dying from the disease even if you catch it) making the numbers difficult to comprehend [15–17]; but with a very wide variation, meaning that it is difficult to represent the range of risks on a single, linear scale. It is well known that even relatively subtle changes in methods of communication can have profound effects on audiences' perceptions of risks and behaviours (e.g. [18–20]) and hence careful empirical work alongside qualitative work with the intended audience is key to designing effective communication messages.

In this set of experiments, we set out to produce empirical evidence-based guidelines to help support the designers of personalized COVID-19 risk communications, recognizing that they may have different aims and core audiences. Before embarking on such work, it is vital to define these aims, audiences and medium of the communication as these determine the outcome measures and constrain the design.

## 1.1. Aim of communication

Communication messages lie on a continuum from a purely persuasive design (e.g. many public health messages) where the desired outcome is behavioural change; to purely informative (e.g. informed consent processes) where the outcome of interest is objective comprehension only. Some authors have described 'risk communication' as concentrating on informing and 'crisis communication' as concentrating on behaviour change (e.g. [21]).

Many trying to communicate the risk from COVID-19 might be aiming for some level of behavioural change (people adopting either more or fewer actions to mitigate the risks from the disease), which might mean, for instance, placing an individual into a risk band (e.g. 'high risk') with tailored behavioural advice. Others may be aiming to be as neutral as possible, allowing individual interpretation of the risk, which will naturally vary between individuals.

The first approach communicates less information to the audience and requires less from them, so may be preferred by some, while others may find its persuasive intent less trustworthy. The second approach avoids some practical difficulties. For example, the same additional absolute risk of death would present a very different prospect to, say, a 9-year-old and a 90-year-old because of their different background levels of risk, making automated categorization of risks difficult. Simply presenting the absolute risks and allowing the audience to interpret them avoids that difficulty, but makes the communication element more challenging.

We approached this study from the perspective of those wishing purely to inform their audience, to support those trying to meet this communication challenge, but also collected data on the public's opinion and preferences on where they would expect individualized COVID-19 risk communication to fall along this spectrum.

## 1.2. Audience for communication

The main audience for our purposes was the general public. This presents the greatest communication challenge, and means that any communication produced should then also be suitable for other

contexts (e.g. use by a healthcare professional in checking a patient's risk while making treatment decisions, or communicating risk to an individual patient).

## 1.3. Medium of communication

We designed the communications for use online, optimized for mobile phone screens and possible to print out, but hope that our findings will be more broadly generalizable.

Our study was carried out during the COVID-19 pandemic with the aim of providing information in real time to communicators who were producing personalized COVID-19 risk calculators, so our methods involved some pragmatic design choices.

We ran a series of qualitative interviews, first with members of the public and then with primary care physicians, which fed into design choices throughout the process. After the initial rounds of qualitative interviews, we started potential designs of the communication and refined these through further interviews and, simultaneously, through a set of quantitative experiments. Overall we focused on a number of key research questions:

## 1.4. What are the information needs of the public?

Even taking into account the potential range of aims that different communicators might have when approaching the communication of personalized risk from COVID-19, it is important to understand what the public's current state of knowledge on the subject is, and what information they particularly want to have (or specifically not to have).

One method of approaching risk communication is the 'mental models' approach, whereby researchers use both qualitative and quantitative methods to build up an understanding of the audience's intuitive model of causality in a situation, as well as the probabilities they ascribe to the likelihoods of different events and the strengths of causal relationships. The idea is then to provide the audience with information to help bring their understanding of the situation closer to the expert understanding of it, correcting any misunderstandings or providing missing pieces of information [22,23]. In this study, we did not set out to complete a full 'mental models' approach as our aim was to communicate a specific piece of information (an individual's risk of dying from COVID-19 should they catch it), but we did want to capture the audience's existing knowledge about this topic, and their information priorities and preferences, in order to tailor communications.

## 1.5. Which format should probabilities be presented in?

Probabilistic information can be represented numerically in different ways, and research has shown that these different formats can affect the perception of likelihoods as well as affecting the ease with which people can make mental comparisons and manipulations of the information [24–28]. Since in this particular example the probabilities being represented were often very small, we wished to explore explicitly how different possible formats affected people's perceptions of the risks portrayed across the range of probabilities, from 1 in 10 000 (a very low risk) to 1 in 5 (a very high risk).

## 1.6. How should context be provided for the numbers?

Numbers on their own do not convey a perception of quantity—they need to be given context, particularly when the numbers are small [29]. However, the choice of the context (such as comparators for the risk) just like the choice of frame (discussed below) can clearly influence the perception of a risk [30–33]. Choosing contextual information that is 'informative' without being 'persuasive' is hard, if not impossible. A risk is more than just a number, a likelihood, and attempting to present risks to compare with each other which, to the audience, represent very different concepts can be perceived as unhelpful at best—manipulative at worst [33]. There is a paucity of empirical literature on the effects of different kinds of comparator risks on risk perception (e.g. [34]). Slovic [35] warns that the choice needs to take into account people's differing feelings about different risks. It is, therefore, important to work with the intended audience to find ways to provide context to the numbers that are acceptable and to be aware of potential biases that they may introduce.

## 1.7. Should the numbers be visualized and how?

Most people do not 'think' in numbers. Visually showing the comparative difference between numerical quantities could help subjective comprehension of them. People have an innate sense of quantity, before even being taught about number symbols, and the use of number lines and other graphics has been proposed to help people assess and compare quantities without the need for formal symbolic number sense [36,37]. Work into the best ways to visualize numbers for risk communication purposes is ongoing and varies by context and with the magnitude of the numbers [19]. Small probabilities cannot easily be visualized using icon arrays, one of the most popular methods of visualizing risk, as the denominators need to be so large, but visual number lines, called risk ladders, can be useful—although research on them has proved inconsistent [17].

When constructing a graphical scale such as a risk ladder, the question arises of whether to represent the numbers on a linear or nonlinear scale. When the probabilities being portrayed are very small and/or vary over several orders of magnitude, a logarithmic scale such as the Paling perspective scale [38]—a chart designed to try to help put new risks into a context of everyday experiences by plotting a number of 'familiar' risks on a logarithmic scale—is popular. However, it is not clear whether such scales allow people to judge the relative sizes of risks [31,39].

## 1.8. Should positive or negative framing of the numbers be used?

The fact that the framing of numbers (whether they represent a positive outcome or gain versus whether they represent a negative outcome or loss) affects the perception of risks is one of the most robust findings in psychology and risk perception [40,41]. However, to understand the magnitude of its effect in any given situation and with any given audience requires empirical testing.

Across all our experiments we were interested in a number of outcome measures. Because our aim was not explicit behaviour change, we decided instead to use Weinstein and Sandman's effective message evaluation measures as a starting point: *objective comprehension*, *agreement with recommendations/advice*, *dose–response consistency* (do people facing a higher dose of the hazard perceive greater risk?), *hazard–response consistency* (do people facing a hazard that is higher in risk perceive greater risk?), *uniformity in response* (do people exposed to the same level of risk tend to have similar responses to it?), *audience evaluation* (subjective measures) and a regard for *types of failure of the communication* and how acceptable those might be [42].

Alongside these, we were interested in how different methods of communication affected participants' risk perception (as measured by their assessment of the likelihood, severity and worry about the risk), their perception of the uncertainty in the estimate, and their trust in the communication.

By addressing this series of research questions and different endpoints, we hoped to be able to produce an evidence-based set of guidelines for practitioners attempting to communicate people's individual risk from COVID-19 in such a way as to suit their own aims.

This study consisted of four online large-scale quantitative surveys, supported by four rounds of interviews with members of the public and one round of interviews with primary care physicians. The qualitative interviews were run as an iterative process in parallel with the quantitative surveys, each informing design of the risk communications, which were refined constantly in the light of findings.

We outline specific pre-registered analysis plans below as we set out each part of the study.

# 2. Participants and practicalities

This study was approved by the Psychological Research Ethics Committee at the University of Cambridge (PRE.2020.070).

Qualitative interviews were carried out via video conferencing due to the pandemic. Recruitment initially used convenience and snowball sampling techniques. Participants from previous studies, and individuals who were otherwise known to the study team were contacted. They were asked to share an invitation to participate with family and friends. This led to a list of around 85 participants for whom we had basic demographic data (age, ethnicity and sometimes health status). Purposive sampling from this participant list led to interviews with a broad demographic range of individuals in each round of testing.

Prior to interviews, participants were given relevant information about the study so they could determine whether they wanted to take part. If participants had any questions pertaining to the study,

these were answered by the study team. Informed consent was taken at the beginning of each interview via an online survey platform. Participants received a £10 store voucher as payment for taking part.

All quantitative experiments were carried out online, with participants recruited through the ISO-accredited polling company Respondi. Quotas were set to ensure recruitment of UK participants representative of the national population on age and sex. Participants were paid £2.50 and each survey lasted 20–25 min.

Participants in the quantitative surveys were quota sampled so as to be representative of the UK population on age and sex. In survey 1, they were also quota sampled by ethnicity to be representative of the UK (we prioritized ethnicity representation in this survey over representation by sex, as we believed it more important when collecting information to be representative on ethnicity in this context, hence there was a slight difference in sex ratio in survey 1).

Participants were excluded from participating in more than one survey. In addition, demographic information was collected: age, sex registered at birth, COVID-19 risk perception (measured by six items from [5]), numeracy (using a sum of their score on the adaptive Berlin numeracy test [43], three items from [44] and a single item from [45], electronic supplementary material, appendix S4), household income, employment status, whether they had, or suspected they had, coronavirus, perceived social status [46], health literacy (using subscale 9, 'Understanding health information well enough to know what to do', of the Health Literacy Questionnaire [47], electronic supplementary material, appendix S4), ethnicity and education level. See table 1 for a summary of participant characteristics.

All stages of the study were carried out between the 3 June 2020 and 23 July 2020.

# 3. Qualitative interviews and survey 1

## 3.1. Introduction

The first stage of a good risk communication process, as already described, is to understand the audiences' information needs and their existing understanding of the subject. We approached this both through qualitative methods and a quantitative survey.

Qualitative methods, in the form of semi-structured interviews, were used as they are suited to exploratory research, including hypothesis generation, as well as producing detailed and rich descriptions of the phenomenon under study [48]. The flexible nature of semi-structured interviews has many benefits, including enhancing the flow of the interview. However, this means that at times not all participants are asked every question in the interview guide. Interviews were split into two phases: discovery and alpha. Discovery interviews aimed to understand the user, their needs and their expectations of a tool for communicating personalized risks from COVID-19. The iterative alpha phase allowed design and testing of potential communications, concentrating on the aspects already introduced: what format to present probabilities in (frequencies or percentages), how to convey a context for the numbers presented, and—on visual representations—whether to use a linear or logarithmic scale, and the effects of colours. As per the use of semi-structured interviews, the interview guide was adjusted for clarity, and questions changed according to the focus of the research, between each round. Care was also taken to allow other themes to emerge from the conversations that would help the development of communications.

The quantitative survey was designed to capture a snapshot of the risk perception and understanding of individuals' personal risk, as well as information desires of the UK public. The perception of a risk and the numerical understanding of its likelihood and severity are two very different things. Risk perception takes into account emotional factors such as worry about the potential outcomes and therefore depends on an individual's circumstances (e.g. what impact 'being too ill to work for two weeks' will have on an individual; what the relative increase in their risk is). Knowing the audiences' prior perceptions of the risk helps understand the potential impact of communicating about it. There were no planned statistical analyses.

## 3.2. Methods

The discovery phase of the qualitative work consisted of one round of interviews each with the general public ($n = 6$) and primary care physicians ($n = 7$). The alpha phase consisted of four rounds of interviews (R1: $n = 6$, R2a: $n = 4$, R2b: $n = 4$, R3: $n = 8$) with the general public only. For details of participants, see electronic supplementary material, appendix S2.

**Table 1.** Characteristics of participants in the quantitative surveys.

| | | survey 1 (n = 500) | survey 2 (n = 700) | survey 3 (n = 1820) | survey 4 (n = 2500) |
|---|---|---|---|---|---|
| sex (%) | male | 234 (46.8) | 336 (48) | 873 (48) | 1199 (48) |
| | female | 266 (53.2) | 364 (52) | 947 (52) | 1301 (52) |
| ages (%) | 18–24 | 56 (11.2) | 78 (11.2) | 229 (12.6) | 303 (12.1) |
| | 25–34 | 91 (18.2) | 136 (19.4) | 337 (18.5) | 454 (18.1) |
| | 35–44 | 98 (19.6) | 134 (19.1) | 344 (18.9) | 480 (19.2) |
| | 45–54 | 99 (19.8) | 133 (19) | 346 (19) | 480 (19.2) |
| | 55–64 | 86 (17.2) | 118 (16.9) | 310 (17) | 429 (17.2) |
| | 65+ | 70 (14) | 101 (14.4) | 254 (14) | 354 (14.2) |
| ethnicity (%) | white | 422 (84.4) | 636 (90.9) | 1597 (87.7) | 2174 (87) |
| | black | 26 (5.2) | 6 (0.9) | 38 (2.1) | 64 (2.6) |
| | Asian | 17 (3.4) | 37 (5.3) | 123 (6.8) | 160 (6.4) |
| | mixed | 11 (2.2) | 9 (1.3) | 30 (1.6) | 55 (2.2) |
| | other | 7 (1.4) | 5 (0.7) | 6 (0.3) | 15 (0.6) |
| education (%) | no qual | 21 (4.2) | 25 (3.6) | 78 (4.3) | 116 (4.6) |
| | GCSE-level | 116 (23.2) | 163 (23.2) | 407 (22.4) | 578 (23.1) |
| | A-level | 140 (28) | 178 (25.4) | 495 (27.2) | 692 (27.7) |
| | bachelors | 133 (26.6) | 207 (29.6) | 525 (28.8) | 717 (28.7) |
| | masters | 49 (9.8) | 72 (10.3) | 189 (10.4) | 250 (10) |
| | doctoral | 13 (2.6) | 17 (2.4) | 44 (2.4) | 49 (2.0) |
| | other | 19 (3.8) | 29 (4.1) | 55 (3.0) | 71 (2.9) |
| experience of COVID-19 | confirmed infection | 1 (0.2) | 4 (0.6) | 18 (1.0) | 14 (0.6) |
| | unconfirmed infection | 77 (15.4) | 99 (14.1) | 275 (15.1) | 373 (15.0) |
| | no infection | 422 (84.4) | 597 (85.3) | 1526 (83.8) | 2113 (84.5) |
| numeracy[a] (%) | 1 | 19 (3.9) | 47 (7) | 171 (9.7) | 166 (6.7) |
| | 2 | 59 (12.1) | 67 (9.9) | 177 (10) | 250 (10.1) |
| | 3 | 80 (16.4) | 100 (14.8) | 280 (15.8) | 363 (14.7) |
| | 4 | 75 (15.4) | 120 (17.8) | 355 (20.1) | 445 (18) |
| | 5 | 93 (19.1) | 119 (17.7) | 287 (16.2) | 467 (18.9) |
| | 6 | 93 (19.1) | 121 (18) | 279 (15.8) | 428 (17.3) |
| | 7 | 24 (4.9) | 40 (5.9) | 83 (4.7) | 137 (5.5) |
| | 8 | 44 (9) | 60 (8.9) | 138 (7.8) | 217 (8.8) |

[a]Score 1–8 on using a sum of their score on the adaptive Berlin numeracy test [43], three items from [44] and a single item from [45].

Five interviewers carried out semi-structured interviews using video call technology and, where appropriate, screen share functions (e.g. when getting feedback on visualizations in alpha rounds). Interviews took on average 1 h to complete (range 0.5–2.25 h). Calls were recorded and partially transcribed by two members of the team (AL and LF). Data analysis was conducted by one member of the team (AL) and was descriptive in nature [49]. Analysis was done by populating a table per round of interviews with data from the partial transcriptions, whereby each question, or appropriate groups of questions, represented a row of the table, with each column representing a participant. Once the table had been populated, the researcher summarized each row of answers. In conjunction with this approach, there was some use of quantitative content analysis methods [50] so the data could more easily be used for making decisions regarding risk communication. If multiple participants made similar comments, these comments formed inductive 'codes', which were then

weighted through the use of descriptive statistics. While undertaking the descriptive analysis, themes were informally identified within the data. As this was a purely qualitative analysis, there was no need to assign descriptive statistics to each theme. A formal thematic analysis [51] was not carried out as time was limited and this was not considered a primary output of the research. Once all the analyses for each round had taken place, it was discussed with two other team members (LF and GL). These two other team members had either partially transcribed the interviews or had carried out their own rapid descriptive analysis using the same populated table. Within these discussions, the researchers identified and resolved any discrepancies between their impressions of the data, as well as adding any missing points to the existing analyses. The alterations and additions made were minimal.

Since the discovery phase interviews with primary care physicians were carried out after the discovery phase interviews with members of the public, they were also shown mock-up visualizations and asked to give comments. The main questions asked during these interviews are shown in electronic supplementary material, appendix S1 (grouped by theme).

In quantitative survey 1, participants were asked a series of questions about their information needs ('I would like to know now what my personal risk of dying from COVID-19 would be if I were to catch the virus'; 'I think that people are entitled to know now what their personal risk of dying from COVID-19 would be if they were to catch it'; 'I would not like my employer to know my personal risk of dying from COVID-19'; 'I feel that I have enough information already about my personal risks from COVID-19'; 'I would like to know by how much each personal behavioural change (e.g. wearing face masks, washing hands) reduces my personal risk of catching the virus'; 'I would not trust any information about my personal risk of dying from COVID-19 as I don't believe enough is known about it': each answered on a 7-point Likert scale marked from 'completely disagree' to 'completely agree' with 'I don't know' marked as the mid-point). They were also asked to estimate their own chance of catching and dying from COVID-19 on a 9-point Likert scale marked 'not at all likely' to 'almost certain' (and asked why they rated their chance at this level in free text), and in the demographics section were asked a series of health-related questions which allowed a rough estimation of their actual chance of dying from COVID-19 if they caught it.

### 3.2.1. Power calculation

Given that there were no planned statistical analyses, we based our participant numbers for survey 1 on past experience. In our prior research [5], we found that a sample size of 700 per country was more than adequate to characterize predictors of COVID-19 risk perception in that country. Further analysis of the earliest UK dataset in [5] indicated that the mean value of 700 participants versus just the first 500 differed by less than a quarter of a point on every 7-point Likert item, suggesting that 500 participants would be adequate for a descriptive study to characterize attitudes on similar topics.

## 3.3. Results

### 3.3.1. Information needs

In interviews, when asked how they felt about a tool in which they would see a risk of death, and whether it would be useful to them, participants said they would use the tool for decision-making (e.g. relating to going to work, school, social events, travel) (11 out of 22; 50%) or out of curiosity ('…it would just be interesting to know if it was something you could beat, or if it would beat you.') (7 out of 22; 32%), but recognized others might not want to as it may be anxiety-inducing (6 out of 22; 27%). All participants felt comfortable seeing it in the context of the study.

Other information that participants mentioned being useful included: how infectious an individual with COVID-19 is; what the treatment would be if you did catch it; how distance affects infection rate; likelihood of hospitalization; likelihood of long-term effects; likelihood of catching it and how much each suggested piece of mitigation advice reduces that; likelihood of transmitting it (and again, effectiveness of preventative measures); and an idea of the likely severity of symptoms if you get it.

In the quantitative survey, we found widespread demand for personalized information on people's risk of dying from COVID-19 if they caught it, and their risk of catching it. A minority thought that they would not want such information (figure 1). The majority also stated that they would want that information in numerical form, rather than as simple 'low, medium, high' categories (figure 2). The reasons given in interviews in favour of a simple categorization were to help people with low

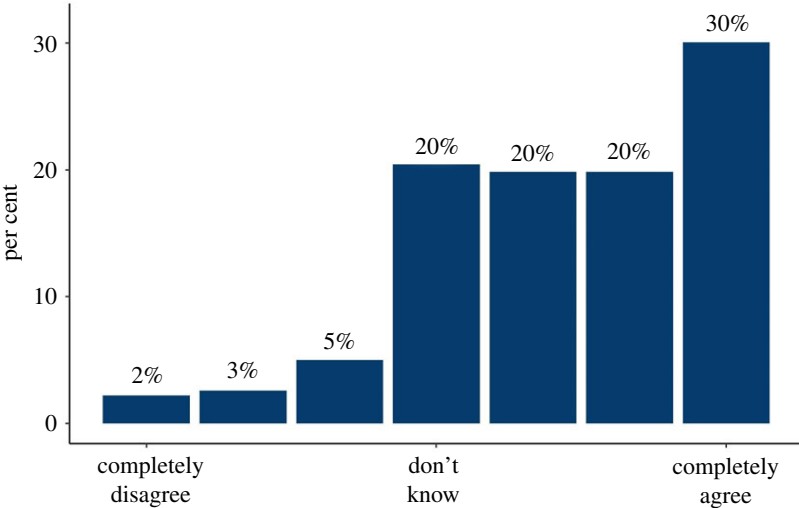

**Figure 1.** The proportion of people answering 1–7 on a Likert scale 'I think that people are entitled to know now what their personal risk of dying from COVID-19 would be if they were to catch it.' (survey 1, $n = 500$).

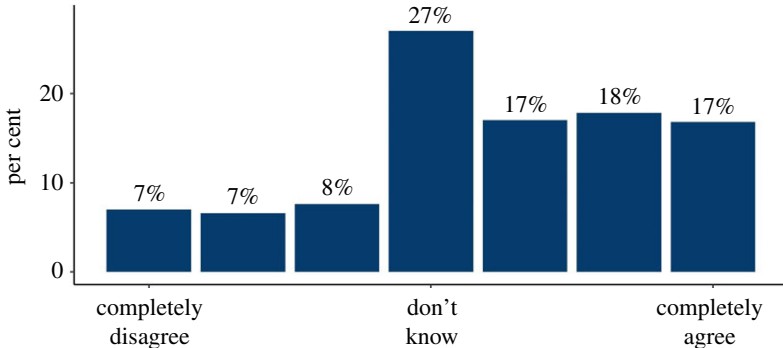

**Figure 2.** Participants' rating on a 7-point Likert scale of how much they agreed or disagreed with the statement 'If my doctor was going to tell me now my personal risk of dying from COVID-19 if I caught the virus, I would like to know that risk as a precise number, rather than as a category (e.g. low, medium, high)'—asked before people had seen any mock-up showing risk as an exact number (survey 1, $n = 500$).

numeracy, and to be more persuasive in terms of getting people to change their behaviour. Reasons given against included that it might be more frightening for those who were told they were at high risk.

One of the themes identified within the interviews with primary care physicians was that of infantilization and empowerment. Some physicians spoke strongly about how they felt the public in the UK were being 'infantilized' by clinicians who implied that '*every risk can be managed away*' and by the government for not being honest about the risks that some socializing can bring. They also recounted how patients ask them what they should do, wanting an authority instead of '*the risk of thinking for themselves*'. However, they also felt it was important that patients made their own decision about certain risks, as only the patient's values can inform that. Additionally, one physician felt such risk information could empower patients to '*have those conversations with the people who are making the decisions*', for example, employers.

Another theme recognized within the primary care physician interviews was related to conflicting medical advice whereby the participants spoke of experiencing contention between the advice given by the government, which they viewed as political, and the advice they wanted to give, which they viewed as clinical.

Preferences expressed by the public in our online surveys and interviews also showed a desire for trustworthy, personalized risk information.

During interviews, when asked to quantify risks from COVID-19, participants rarely used numbers without being prompted, and instead naturally described the characteristics of people they would

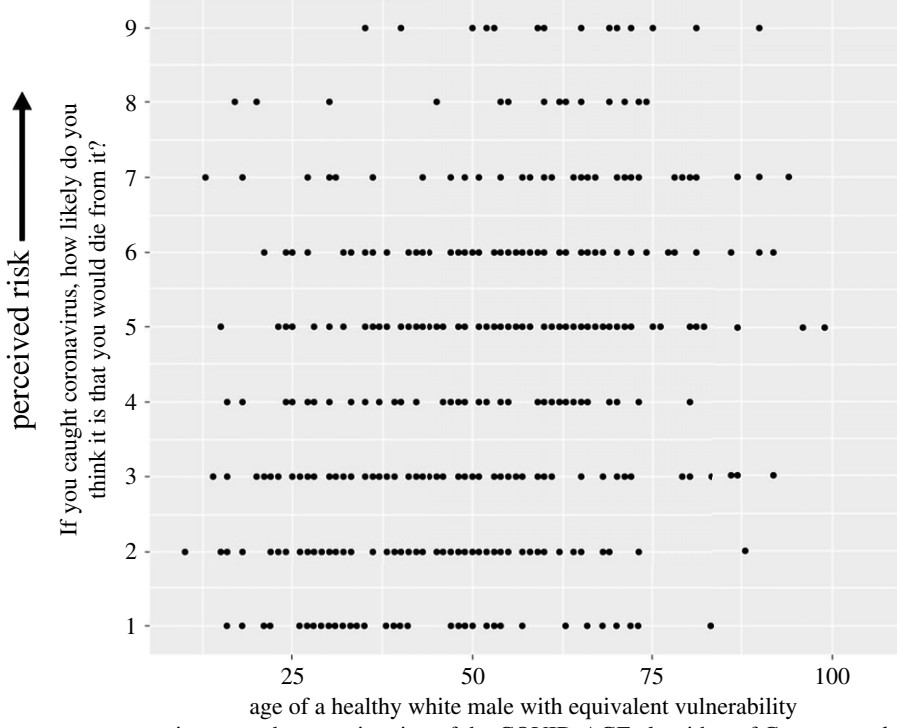

**Figure 3.** The ratings of personal risk of participants (on a 1–9 Likert scale) against their approximate actual risk (presented as 'COVID-AGE') as calculated approximately by the algorithm by Coggon *et al.* [9] as a means of assessing the degree to which people's assessments of their risk from COVID-19 correlate with their actual risk (survey 1, $n = 500$).

consider 'high' or 'low' risk (11 out of 22; 50%), for example, *'the older you get the more at risk you are'* or, low-risk individuals are *'in good health'*. However, when pushed to quantify the risk, participants gave estimates as seen in electronic supplementary material, appendix S3.

### 3.3.2. Personal risk perception

Overall, participants marked their worry about COVID-19 higher than their worry about any other subject they were asked about, in line with previous findings about the perception of the UK population around this time [5].

To get a sense of how individuals' perceptions of dying from COVID-19 if they caught it compared with their actual risk, we estimated the latter by applying an approximation of the online COVID-AGE algorithm by Coggon *et al.* [9] to the demographic and health data provided by participants. This approximation was necessarily imprecise as we did not collect the full breadth of health information that the full COVID-AGE uses, but nevertheless provided a rough estimate of actual risk levels. The correlation between perceived risk and actual risk was weak (Spearman's $\rho = 0.4$), suggesting that perceived and actual risk are not closely related (figure 3).

### 3.3.3. Which format to present probabilities in

In interviews, 7 out of 14 (50%) participants mentioned finding frequencies hard to understand: *'I think it's a lot clearer with the percentages. I mean 190 out of 1000, I think it's hard to imagine'* and felt the constant repetition of frequencies on the visual scale they were shown contributed to this (4 out of 14; 29%): *'It's just all zeros everywhere!'*. Some participants also commented that, in order to understand their result given as a frequency, they would have to convert it into a percentage (3 out of 14; 21%). Lastly, in line with previous research findings (e.g. [25]), some participants suggested that the use of frequencies made the risk seem higher (3 out of 14; 21%).

### 3.3.4. How to provide context for the numbers

Participants in interviews were asked about several types of contextual information (the risk for a healthy person of a specified age; the risk of other causes of death; the risk for a person with specified health conditions and age; the risk for the individual getting the results if they did not have any health conditions; the risk for an 'average' person of a specified age; what proportion of the population had a higher or lower risk than the individual getting the result). One suggestion for context to attempt to make a person's 'relative risk' clear has been to compare an individual's own risk with that of someone who was 'like them but without health conditions' [52].

Participants found these sorts of hypothetical scenarios confusing and/or felt that they raised more questions than they answered (7 out of 8; 88%). Those options that compared the participant's risk of dying from COVID-19 to dying from other causes were thought likely to be helpful but only if those risks seemed similar enough to COVID-19: when asked if a comparison with seasonal 'flu was useful and understandable, 7 out of 8 (88%) said yes, while 5 out of 6 (83%) found risks of accidental injury irrelevant and not useful. Those presentations that tried to communicate a concept that wasn't an absolute risk (e.g. a person's position within a population, such as a percentile, or the number of people who had such a risk score) were easily misunderstood as an absolute risk (e.g. '75% of people have a higher risk than you' being read as 'you have a 75% risk') (3 out of 8; 38%), or were felt to be confusing (5 out of 8; 63%). Again, these concepts were difficult for participants to understand as they already had their own individual risk in their head and were, therefore, not thinking at a population level.

### 3.3.5. Whether to use a logarithmic or linear scale

In the qualitative interviews, 21 out of 29 participants (72%), including health professionals, were confused by a logarithmic scale and/or considered it to be unfairly representing the risks, for example: *'I don't like the scale because it's supposed to be a scale but it's not TO scale! Spatially it's not right!', The 12% has been manipulated to look higher!'*

The estimated risks of an individual dying from COVID-19 if they become infected with SARS-CoV-2 very rarely exceed 30% (or, conversely, their chance of surviving it is rarely below 70%).

If a linear scale is used it is likely that communicators will want to cut the axis and display half or less than the full 0–100% scale. In interviews, participants felt that this was justified (6 out of 7; 86%), but wanted it to be made clear in the visualization, and to know why the particular cut-off point was chosen. Knowing 'the highest risk possible' was helpful for them to calibrate their perception of the numerical risks, and 5 out of 7 participants (71%) with whom it was discussed again found it helpful to have that 'maximum risk' presented in terms of a persona: the type of person (in terms of risk factors such as age, health conditions, etc.) who might be at that sort of level of risk.

### 3.3.6. Responses to different colour schemes

In interviews, participants expressed varied opinions about a 'traffic light' system of green, orange and red used to colour some mock-up scales. Those shown these colours felt this coloration was easily understood, but 2 of the 6 participants (33%) shown the colours spontaneously added that it could be misinterpreted: *'The green colour could make people think that they don't need to worry that much and not undertake proper behaviours'*. It became clear that particular colours could be interpreted as an indication of what 'should be' considered high risk and what low risk. The decision over colouring, therefore, depends on how pointedly the communicator wants to guide interpretation.

### 3.3.7. Emerging theme: trust

In a descriptive analysis (purely qualitative, with no descriptive statistics), we extracted emerging themes from the interviews.

In interviews, we specifically asked the participants if they would trust an outcome from the tool, and what would make them trust it more. Sub-themes (which were unprompted) related to trust were identified:

#### 3.3.7.1. Trust related to the data

Participants questioned whether the data were collected in a rigorous way, how accurate the data were and whether it was being 'tampered with' once collected.

### 3.3.7.2. Trust related to the source

Some participants identified the University of Cambridge logo displayed on the mock-up; knowing that the source of the tool was a university institution that undertook research gave them trust in the outcome.

### 3.3.7.3. Trust related to the methods behind the results

Suggestions that the risk information was produced through careful research (indicating the high quality of underlying evidence) affected the participants' perceptions. One said: *'It's important that people know it's not been plucked out the air, like they think it might be, but that it's actually based on data'*.

### 3.3.7.4. Trust related to the medium

Referring to increasing incidences of online scams and other malicious online activity, one participant commented: *'People are careful these days of anything online'*, going on to suggest that the inclusion of institutional logos was particularly important in gaining people's trust when communicating online.

### 3.3.8. Emerging theme: uncertainty

Uncertainty was identified as a theme within interviews, with many participants commenting that the risk score itself was inherently uncertain. This did not seem to affect their feelings of how useful the tool was, nor their trust in the guidance it might provide. Some participants indicated that the range around the risk score presented in the mock-ups was superfluous, possibly because they instinctively acknowledged the uncertainty of the result. Some also felt that the data which could be used to calculate the risk score was uncertain. This uncertainty of the data, and whether it was accurate and reliable, did seem to affect trust. These findings were also broadly similar within the primary care physicians, though unlike the public, uncertainty about the risk score itself was attributed to applying population-level data to individuals.

### 3.3.9. Emerging theme: worry

Throughout the general public interviews, numerous participants spoke about how the tool may worry them or others. Many primary care physicians also commented on how the tool could have a negative effect on the mental well-being of patients. By contrast, one primary care physician detailed how a tool like this would empower them to talk openly with their patients who were unnecessarily concerned, which for some had resulted in deteriorating mental health.

## 3.4. Interim discussion

The majority of participants clearly showed a desire for quantitative information about COVID-19 (much of which was not, at the time of this survey, available). The comments from the primary care physicians suggested that they viewed patients as perhaps becoming used to being given simplified instructions rather than information on which to base their own decisions. However, the results from survey 1 implied that the majority of the public would actually appreciate being given more detailed information on which to base their own risk decisions.

The quantitative survey emphasized that there was a clear public concern about the virus and its potential consequences, and people's perception of chances of themselves dying of the virus if they caught it were not highly correlated with the actual chances as calculated from a personalized risk calculator given the health information participants gave us, with only a weak relationship between the two.

Some of this discrepancy seemed very likely to be due to the way that most people think about 'risks'—as a subjective impression, not easily translated into a numerical concept [53]. Even when pushed to quantify their feeling of risk, most members of the public in our interviews described the risk in terms of a 'persona' that they had in their mind to represent that level of risk: a person who embodied the major risk factors that would exemplify that level of risk (such as age and relevant health attributes).

Given that only a minority of people in survey 1 wanted simply to be told that their risk fell into a simple category such as 'low' (as well as the practical and ethical difficulties of doing that given that an absolute risk and relative risk would need to be combined in some way), we explored possible comparators to help the public put their own personal risk from COVID-19 into context. The most commonly suggested comparators (such as other causes of death) were generally not deemed helpful, but using insights from the way people

described their concepts of 'high' and 'low' risks from COVID-19 in terms of imaginary people, we chose to experiment further with the idea of these 'personas' as comparators (see below).

The responses to a variety of mock-up visual risk ladders also gave us insights into the likely responses to the use of different colours. We decided not to pursue this dimension further in our research as we deemed it likely to be highly influential in how people interpreted the numbers and we wanted to investigate the more subtle and less persuasive cues that could be provided in a communication. The responses also suggested that a logarithmic scale on a risk ladder could prove misleading or untrustworthy, but we wanted to investigate that further in quantitative surveys.

Equally, the information about frequencies and percentages was in line with previous findings, but we thought it worth quantifying the differences in perception between different numerical formats in this particular context and range of magnitudes as the choice of format is one that communicators need to make and would want to do so based on empirical evidence.

The other themes emerging from the interviews were useful in determining important endpoints to measure throughout our study: levels of perceived uncertainty, perceptions of the quality of the underlying evidence, trust (in the data itself and in the source of the data) and degree of the worry associated with the results.

# 4. Survey 2

## 4.1. Introduction

Building on the knowledge gained from the first interviews and survey 1, survey 2 was designed to start investigating the concept of 'personas' as a method of giving context to the risk, which had arisen as an idea as a result of the qualitative interviews. Firstly, we wanted to investigate how giving information about the person *whose risk it was*—in the form of a persona—affected the perception of that risk, and whether this had a different influence on people's perception of the risk than simply giving them the context of a range of risks for an 'average' person of different ages which they could compare with the numerical risk they were presented with.

In addition, we wanted to investigate how strong the influence of information about personas might be compared with numerical information. To do this we decided to present personas alongside risk numbers that were discordant with the description of the person (i.e. a very high-risk percentage displayed alongside the description of someone who would naturally be perceived as being low risk and vice versa), to see how much this affected people's assessment of the risk.

We also wanted to quantify any potential effects of the presentation of numbers as percentages versus frequencies, and to investigate alternative ways of putting risks into context visually. Although the interviews had suggested that percentiles were confusing, the concept of showing a person's absolute risk as a position within the distribution of absolute risks across the population *visually* seemed a way of getting at the same concept in perhaps a more intuitive way. We, therefore, designed an experiment to test the influence of showing where a person lay in a population-wide distribution of risk.

## 4.2. Methods

Participants were asked a series of questions relating to their information needs. Since these were shared with the concurrent survey 3, they are described and reported together in the Survey 3 section of this manuscript. Participants were then presented with two experimental sections of the survey.

### 4.2.1. Experiment 2.1: interpretation of numerical risks with and without context

Each participant was shown a set of five hypothetical risk results, one after the other; the order of presentation was randomized.

Participants were also randomized to one of 8 conditions in a 2 (format) × 4 (context) factorial design (resulting in an overall 5 (risk level; within subjects) × 2 (format; between subjects) × 4 (context; between subjects) mixed design). The *format* factor referred to the format in which participants were presented with numeric risks: as a percentage or as a frequency. The *context* factor had four levels (*number only*; *ages*; *concordant*; *discordant*), and referred to the context that participants were provided before being asked how they would interpret a risk. The context corresponding to each level is shown in table 2. (Note that the numeric risks were shown to all participants within-subjects in random order, and that

**Table 2.** Wording presented in Experiment 2.1. Each participant was presented with five hypothetical risk levels (shown in the rows), and was assigned to one of eight conditions in a 4 (context condition) × 2 (percentage versus frequency format) design.

| risk level (percentage/ frequency) | context condition | | | |
|---|---|---|---|---|
| | number only | ages | concordant | discordant |
| 0.1%/1 in 1000 | imagine that you were told that if you caught COVID-19, your risk of dying of it were: 0.1%[a] | if Mel catches COVID-19, Mel's risk of dying is 0.1%. For comparison: The average 85-year-old man with no health problems has a risk of 12%. The average 75-year-old man with no health problems has a risk of 1.7%. The average man under 55 with no health problems has a risk of below 0.5%. | if Mel catches COVID-19, Mel's risk of dying is 0.1%. For context: They are a white man aged 30 with no underlying health conditions | if Mel catches COVID-19, Mel's risk of dying is 0.1%. For context: They are an Asian man aged 85 with a heart condition and diabetes |
| 1%/10 in 1000 | imagine that you were told that if you caught COVID-19, your risk of dying of it were 1% | if Ali catches COVID-19, Ali's risk of dying is 1%. For comparison: [Same as above] | if Ali catches COVID-19, Ali's risk of dying is 1%. For context: They are a mixed race man aged 30 with two underlying health issues | if Ali catches COVID-19, Ali's risk of dying is 1%. For context: They are a black woman aged 75 and with certain underlying health issues. |
| 5%/50 in 1000 | imagine that you were told that if you caught COVID-19, your risk of dying of it were 5% | if Jo catches COVID-19, Jo's risk of dying is 5%. For comparison: [Same as above] | if Jo catches COVID-19, Jo's risk of dying is 5%. For context: They are a white woman aged 40 with a high BMI and undergoing cancer treatment | if Jo catches COVID-19, Jo's risk of dying is 5%. For context: They are a white woman aged 40 with a high BMI and undergoing cancer treatment |
| 12%/120 in 1000 | imagine that you were told that if you caught COVID-19, your risk of dying of it were 12% | if Alex catches COVID-19, Alex's risk of dying is 12%. For comparison: [Same as above] | if Alex catches COVID-19, Alex's risk of dying is 12%. For context: They are a black woman aged 75 and with certain underlying health issues | if Alex catches COVID-19, Alex's risk of dying is 12%. For context: They are a mixed race man aged 30 with two underlying health issues |
| 20%/200 in 1000 | imagine that you were told that if you caught COVID-19, your risk of dying of it were 20% | if Sam catches COVID-19, Sam's risk of dying is 20%. For comparison: [Same as above] | if Sam catches COVID-19, Sam's risk of dying is 20%. For context: They are an Asian man aged 85 with a heart condition and diabetes | if Sam catches COVID-19, Sam's risk of dying is 20%. For context: They are a white man aged 30 with no underlying health conditions |

[a]Text for percentage format condition shown throughout, in frequency format condition risk level presented as 'x in 1000' as shown in the first column.

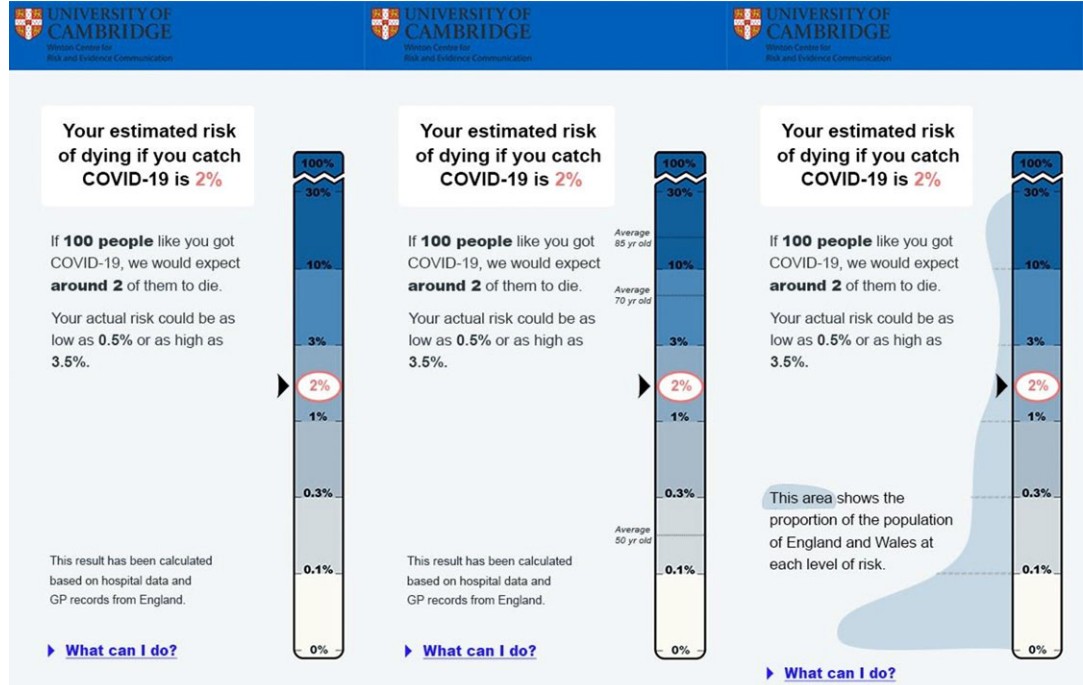

**Figure 4.** The three formats tested in Experiment 2.2: each shows a 2% risk on a logarithmic scale. One shows no additional contextual information, one shows the comparative risks of 'average' people of different ages, and one attempts to illustrate the proportion of the UK population that experiences each level of risk. Please note, the risks illustrated on these visualisations are all fictional (including the age comparators) this was made clear to participants. Please take care not to reproduce these visualisations in contexts where they may be taken as genuine.

participants who were randomized to view frequencies rather than percentages were shown these risks in the format 'x in 1000', e.g. '120 in 1000' rather than 12%).

In each case, participants were asked to give their rating of risk on a visual slider with no numerical cues, marked only 'very low risk' and 'very high risk' at the endpoints.

### 4.2.2. Experiment 2.2: interpretation of visual scales with and without context

Each participant was shown a single mock-up of a hypothetical risk result alongside a visual scale.

They were randomized to see one of three mock-ups, each using a (logarithmic) scale illustrating the result. The control group saw no further information. A second group saw a scale with the risks for 'an average' 50-, 70- and 85-year-old shown alongside it. The third group saw the scale with, alongside it, an illustration of the approximate distribution of the different absolute risk percentages in the UK population (figure 4). A logarithmic scale had to be used for this experiment because a realistic illustration of the population distribution of risk was impossible on a linear scale due to the very large number of people with a very low risk.

Participants were asked to answer the following questions:

— *How well did you understand the information in the mock-up? How clear is the information in the mock-up?* (both answered on 7-point Likert scales marked Not at all—completely); we intended to combine these into an index measure of *comprehension* if their correlation exceeded $r = 0.7$.
— *If the person who got this result caught COVID-19, how likely do you think it is that they would die as a result?* (answered on a 7-point Likert scale: Very unlikely; Unlikely; Somewhat unlikely; Neither likely nor unlikely; Somewhat likely; Likely; Very likely). *How would you describe the risk of this person dying from COVID-19 if they caught it?* (answered via a slider with no numerical cues: 'very low risk' and 'very high risk' as the end points). We intended to combine these into an index measure of *cognitive risk perception* if their correlation exceeded $r = 0.7$.
— *If this result applied to you, how worried would you be?* (answered on a 7-point Likert scale: Not at all worried—Very worried). We treated this as a measure of *emotional risk perception*.

(Although the graphics represented the risk as 'your estimated risk', in all experiments for ethical reasons it was heavily emphasized that the representations were completely hypothetical and did not represent the

participants' risk, even though they were giving us information such as their age. Hence the questions were phrased carefully to re-emphasize that the results participants were seeing were not related to them personally. Please also note that it was incorrect of us to represent the risk at any point as 'your risk' as it is only estimated on the basis of the characteristics entered and thus could never be truly personalized.)

### 4.2.3. Power calculation

Our power calculation for survey 2 was based on the requirements of the experiment that we felt would produce the most actionable information for our collaborators who were producing personalized COVID-19 risk calculators, Experiment 2.2. Determining an appropriate effect size to power for is always a somewhat subjective decision. One approach is to look at systematic reviews of the literature in the same subdomain to see what effect sizes are typical in the published literature. In one systematic review of one-to-one risk communication interventions in health-related contexts, the mean effect size reported was equivalent to a Cohen's $d$ of 0.38—equivalent to a common language effect size (CLES) of 0.61. The CLES is sometimes considered to be easier to interpret than Cohen's $d$, and represents the probability that a data point selected at random from one distribution will be higher than a data point selected at random from a second distribution [54]. The authors assessed that due to publication bias, the true size of risk communication interventions was likelier to be closer to $d = 0.1$–0.2 (CLES = 0.53–0.56) [55]. However, this review prioritized behavioural measures, only quantifying effect sizes of risk perception (our primary focus) when behavioural measures were not available. Generally speaking, the effects of experimental manipulations on risk perception tend to be larger. For example, Portnoy *et al.* [56] conducted a meta-analysis of health interventions, including interventions categorized as 'deliberative' (presenting factual or numeric information, including risk calculators), 'affective' (interventions explicitly intended to provoke emotional responses) and 'decision science-based' (e.g. interventions that changed only the format or framing of risk information). The overall mean effect size on the perceived risk of developing health problems was $d = 0.5$ (95% CI 0.36–0.63), equivalent to CLES = 0.64 (95% CI 0.60–0.67). Only a manipulation's status as a 'deliberative' intervention had a significant effect on effect size, being associated with higher effect sizes; decision science-based interventions were only marginally and non-significantly negatively associated with effect size. Ultimately, we concluded that achieving 95% power to detect an effect size of $d = 0.3$ (CLES = 0.58), using an alpha of 0.05, would be a reasonable approach for this series of studies. GPower was used to determine the number of participants required for a one-way ANOVA (with three groups) at this power, alpha and effect size. This provided an estimated sample size of 690 which we rounded up to 700. *Post hoc* power analysis in GPower also suggested that 700 participants would be adequate to achieve 95% power at the same alpha and effect size in Experiment 2.1 as long as there were not extreme levels of non-sphericity.

## 4.3. Results

### 4.3.1. Experiment 2.1

Participants' risk ratings did appear to be influenced by the independent variables: A 5 (risk levels; within) × 2 (format; between) × 4 (group; between) mixed three-way ANOVA revealed a significant three-way interaction between all factors, $F_{12,2748} = 2.56$, $p < 0.01$, $\eta_G^2 = 0.004$, which we decomposed by running separate 5 × 2 mixed ANOVAs within in each group. Greenhouse–Geisser corrections were applied to correct for non-sphericity ($\varepsilon$s 0.526–0.710). All groups showed an expected significant main effect of risk level, which we will not examine further here for brevity, except where interactions with other factors were detected.

In the 'number only' group, there was a significant main effect of numerical format, $F_{1,183} = 11.41$, $p < 0.001$, $\eta_G^2 = 0.045$. Tukey's *post hoc* tests indicated that across all risk levels, frequencies were rated as higher risk than the equivalent percentage (all $p < 0.05$).

In the group presented with 'ages' comparison information, we report a significant interaction between risk level and format $F_{2.51,440.02} = 3.78$, $p < 0.01$, $\eta_G^2 = 0.006$. *Post hoc* tests revealed that frequencies were only rated significantly higher than percentages in the 1% risk level ($p < 0.05$).

In the groups presented with an individual's description, there was no significant main effect of format ($F_{1,161} = 0.44$, $p = 0.51$ in the 'concordant' group, $F_{1,168} = 1.45$, $p = 0.23$ in the 'discordant' group), or interaction with the risk level. However, *post hoc* comparisons within the discordant group revealed a significant difference between the frequency and per cent groups ratings of the 20% risk value ($p < 0.01$).

Overall, as can be seen in figure 5, the difference in risk perception between those given a risk in a percentage compared with a frequency was diminished in the presence of additional information. It is also clear that the presence of the contextual information made a difference to people's estimations of

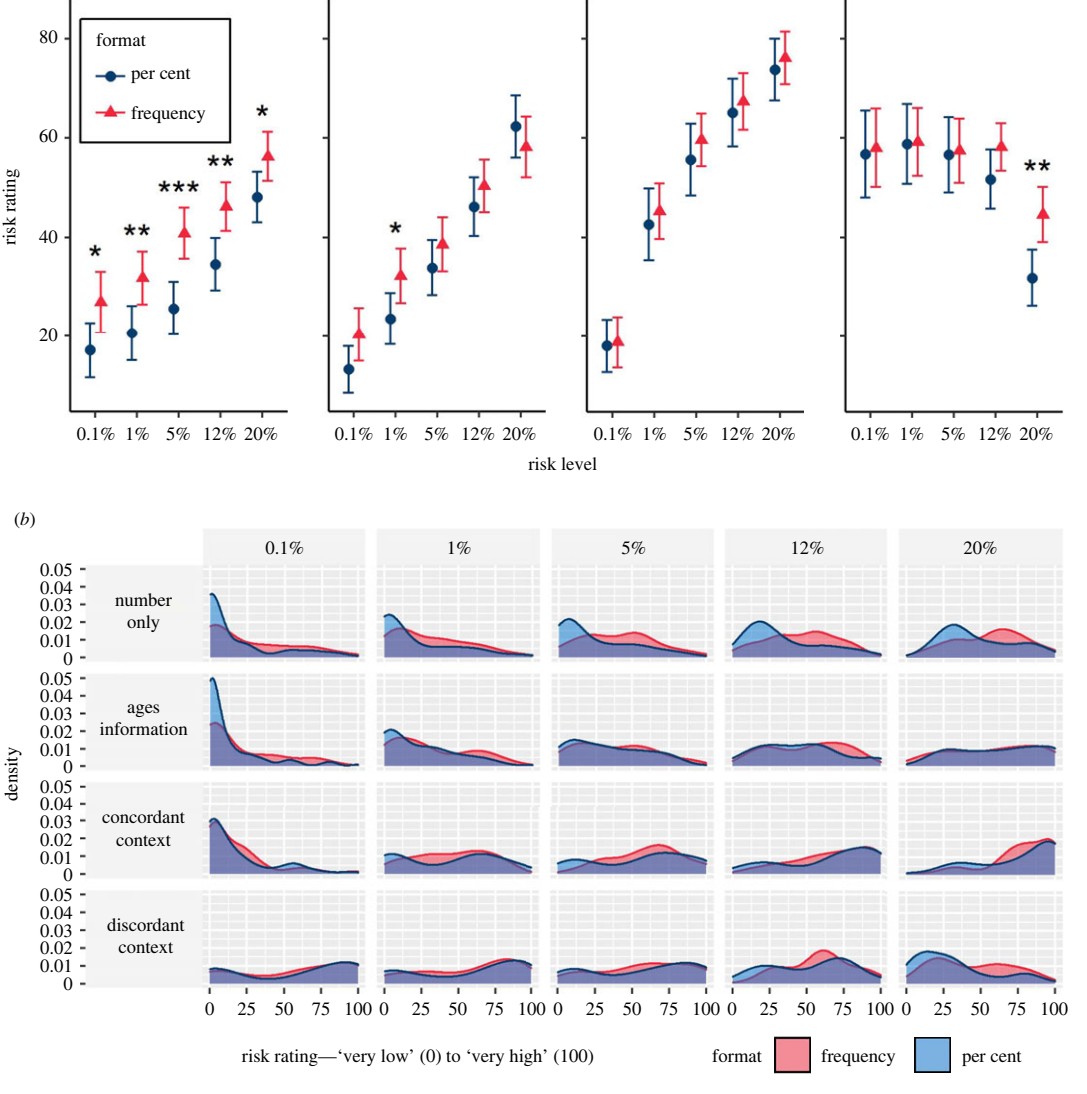

**Figure 5.** Means (*a*) (95% CI) and distributions (*b*) of ratings ('very low risk' (0) to 'very high risk' (100)) of five different risk levels presented as a percentage or frequency (out of 1000), with or without additional information. Asterisks indicate a significant difference between per cent and frequency formats, $^*p < 0.05$, $^{**}p < 0.01$, $^{***}p < 0.001$ (survey 2, $n = 700$).

the risks (compared with the 'no additional information' group), and the effect of giving a description of the risk factors of the individual was so strong that in the 'dissonant context' condition, a 0.1% risk (given as the risk for an 85-year-old man with two health conditions) was judged higher than a 20% risk (given as the risk for a 30-year-old man with no health conditions).

To examine the relationship between demographic factors and risk ratings (e.g. whether males or older people had a lower perception of the risks), we fitted a linear model to the data from the 'no information group' regressing risk ratings on risk level, age, sex and level of numeracy.

Accounting for experimental manipulations, we found no significant effect of age ($\beta = -0.04$, $p = 0.15$) or sex ($\beta = -0.04$, $p = 0.15$) on risk ratings. Numeracy did not moderate the effect of format but did interact with risk level, ($\beta = 0.38$, $p < 0.001$). Lower numeracy individuals tended to rate the risks presented as higher risk compared with higher numeracy individuals, with this difference decreasing as the risk level increased (see electronic supplementary material, appendix S5).

### 4.3.2. Experiment 2.2

The correlation between the items on clarity and ease of understanding exceeded 0.7 ($r_{695} = 0.82$) and were thus combined into an index measure of comprehension. The same was true of the two items

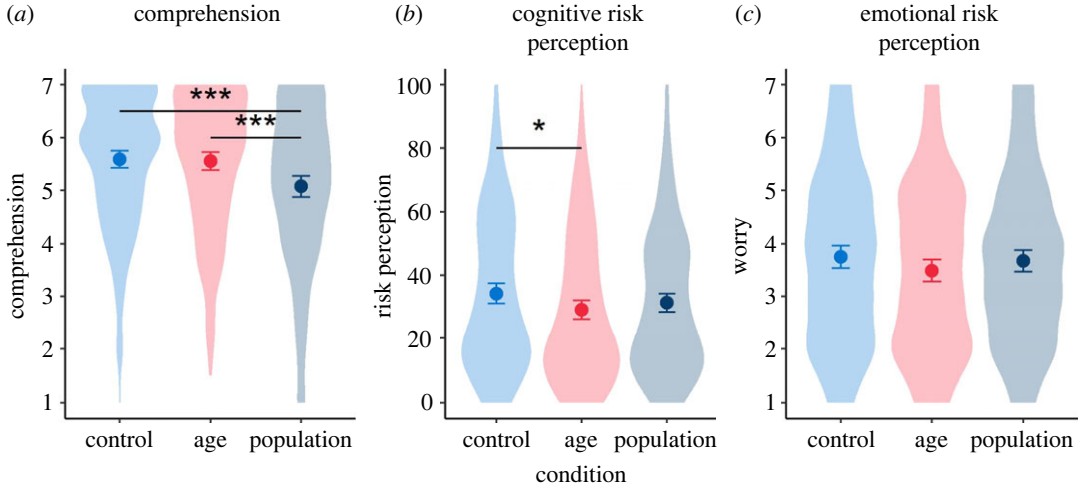

**Figure 6.** (a–c) Mean (95% CI) participant ratings of the three formats shown in figure 4 (risk result shown alongside: no additional information (control), average risk for different ages (age) or risk distribution for UK population (population). 'Violin' plots indicate underlying distribution. Horizontal bars indicate significant difference between conditions, $^*p < 0.05$, $^{***}p < 0.001$ (survey 2, $n = 700$).

asking about the risk of death ($r_{692} = 0.73$), which were, therefore, combined into an index measure of cognitive risk perception. Mean responses of participants asked to rate the visualization that they were presented with on comprehension, cognitive risk perception and emotional risk perception are shown in figure 6.

Cognitive risk perception was slightly decreased by giving the age comparators. A one-way ANOVA suggested a possible difference between groups ($F_{2,691} = 2.81$, $p = 0.061$, $\eta_G^2 = 0.008$.[1] Follow-up *post hoc* testing using Tukey HSD correction for multiple comparisons showed a significant difference between the 'comparator by age' (age) group and the 'no comparator' (control) group ($p = 0.049$, $d = 0.22$; CLES = 0.56). The three visualizations did not lead to a significant difference in emotional risk perception (worry about the result). The population distribution presentation was less well understood than the other two visualizations ($F_{2,694} = 10.07$, $p < 0.001$, $\eta_G^2 = 0.028$; Tukey HSD *post hoc* pairwise comparisons significant for this group versus the other two, both $p < 0.001$: $d_{\text{Population-Age}} = 0.33$, CLES = 0.59; $d_{\text{Population-Control}} = 0.37$, CLES = 0.61).[2]

## 4.4. Interim discussion

Participants' estimates of the likelihood of death from COVID-19 were consistent with an understanding of the major risk factors for the disease: the rank ordering of the risks of the different personas described was in line with experts' estimations of their risks. This has been observed previously within people's mental models of risks (summarized in [57]).

Higher numeracy was associated with lower risk perception, particularly as risks became lower, which might indicate a greater familiarity with numerical risks and hence a more 'realistic' view of the numbers associated with the risks in this instance.

In accordance with previous findings (e.g. [25,27,28]), the same number expressed as a frequency (x out of 1000) was perceived as expressing a higher level of risk that when expressed as a probability. However, the fact that the gap between the two decreased as participants were given contextual information to the number shows the importance of this context to their perception and judgement.

The power of the persona information can be seen by its effect in the discordant condition, where the numerical and the persona information fight each other. The density plots for this condition show the broad spread of participants' responses across the spectrum. However, this is also true of some of the other conditions. For example, whereas the comparison information appeared to help people

[1]Note that the ANOVA merely revealed a trend. However, non-parametric testing to account for the skew in the data revealed a significant effect (Kruskal–Wallis rank sum test, $p = 0.048$). Therefore, *post hoc* analysis was performed and reported.

[2]Non-parametric testing results, performed as a robustness check for the mild skew in the distributions, were in line with the reported parametric results.

assess the lowest risks, the (concordant) persona risks appeared more helpful at the higher end of risk likelihood. This could be because it combines information about multiple risk factors (not just age, as is given in the comparison information, but also health conditions which make clearer what the expected magnitudes of the highest risks might be).

The results of Experiment 2.2 suggested that the graphic attempting to show an individual's risk in the context of the UK population distribution of risks was not clear enough. It may have been mainly a matter of design, but the qualitative interviews suggested that the concept of switching between an individual's absolute risk and the population of absolute risks was confusing, so we decided not to pursue this further, and instead to concentrate on refining potential comparators for an individual's risk.

# 5. Survey 3

## 5.1. Introduction

Survey 3 shared the initial survey elements of survey 2 (and was run concurrently) adding to our investigation of people's information priorities. With only a limited amount of space within a communication, and issues of cognitive load for the audience, we were keen to start identifying what participants considered to be the 'core' information and what could be included as a 'deeper dive'.

To inform the development of a means of displaying an individual's risk of dying from COVID-19 if they caught it, and given that a visual means of display would be helpful to test, the experiment within survey 3 was designed to investigate, quantitatively, the potential impacts of a logarithmic versus a linear scale on a visual risk ladder. Previous research [32] suggested that the visual appearance of the position of a risk on a ladder can influence the perception of that risk. Since a logarithmic scale emphasizes the lower portion of a scale and hence tends to move all but the lowest risks up the scale, we anticipated that a logarithmic scale might affect the perception of, and worry about, the risks. After the initial suggestions in the qualitative interviews that the logarithmic scale may be confusing and potentially less trustworthy, we also anticipated that measures of understanding and trustworthiness might reflect this. The experiment also investigated the potential impacts of both percentage and frequency formats of numbers, this time as they might be presented on a risk ladder format.

## 5.2. Methods

Surveys 2 and 3 both contained two questions designed to help determine information priorities. These are described here in order to report their combined results (see electronic supplementary material, appendix S4 for a summary of key questions in each survey).

After being shown the mock-up of a personalized risk communication tool (Experiments 2.2 and 3.1), participants were asked how much they agreed or disagreed on a 7-point Likert scale (marked from 'completely disagree' to 'completely agree' with 'don't know' labelled as the mid-point) with the following statements: 'I would not like my employer to know a result like this about my personal risk from COVID-19', and 'I would not like to see information like this about my own risks from COVID-19'.

They were also asked:

'Which of the following pieces of information would you most want to know about your risk, if it were available? Please drag and drop into the order you feel puts the most important at the top: The risk of you dying of COVID-19 if you caught it; The risk of you catching COVID-19 given where you live and how many other cases have been reported nearby (will vary day-to-day a bit like a pollen count); The risk of you being hospitalized by COVID-19 if you caught it; Whether you might suffer long term complications from COVID-19 if you caught it.'

'Please drag and drop the following into the order you feel puts the most important at the top: The person's risk score: the chances of that person dying if they caught COVID-19; Information about what data was used to calculate that risk score (e.g. whether it was data from the UK, how much data the researchers had); Information about who developed the maths behind the calculation; Information about the certainty and precision of the risk score: how much lower or higher the actual risk for that person could be; Context about that risk score: how it rates compared with a similar person at different ages (e.g. "someone like you but aged 20"; "someone like you but aged 60"); Context about that risk score: where that person's risk lies in comparison with everyone else in the UK (e.g. a graph of the risks of everyone in the UK with an arrow showing "you are here" to make it clear how many

people have a higher risk than that person, and how many a lower risk); Link to where people can get more information about things they can do to reduce their chances of catching COVID-19 (e.g. hand washing, social distancing, wearing face masks, etc.); Information reminding people that as well as the risk to themselves, they also pose a risk to others (in case they catch COVID-19 and pass it on); Details about how the risk score was calculated (e.g. what factors makes someone's risk higher or lower, and how important each of those factors are); Information about things they can do to reduce their chances of dying from COVID-19 if they caught it (such as losing weight).'

Additionally, participants were presented with a single experiment intended to investigate their interpretation of logarithmic versus linear visual scales, as well as the effect of different number formats (percentages and frequencies) in this context:

### 5.2.1. Experiment 3.1: interpretation of visual scales with linear and logarithmic scales/frequencies and percentages

In Experiment 3.1, participants were randomized to one of 8 groups in a 2 (high or low risk) × 2 (percentage or frequency) × 2 (logarithmic or linear) between-subjects design. They were shown a risk of either 12% or 0.1%, expressed either as a percentage or a frequency, and indicated on either a log or a linear scale (figure 7).

As in Experiment 2.2, participants were asked questions to measure their comprehension, cognitive risk perception and emotional risk perception (worry). As described in our analysis plan (pre-registered at https://osf.io/xpyk9), our key dependent measures were:

— *Cognitive risk perception* (perceived likelihood and perceived risk level): 'If the person who got this result caught COVID-19, how likely do you think it is that they would die as a result?', answered on a 7-point Likert scale, marked 'very unlikely' to 'very likely'; 'How would you describe the risk of this person dying from COVID-19 if they caught it?'—slider with no numerical cues, very low to very high, coded 0–100).
— Subjective *comprehension* of the information provided ('How well did you understand the information in the mock-up?', and 'How clear is the information in the mock-up?'; answered on a 7-point Likert scale marked 'not at all' to 'completely').
— *Emotional risk perception* (worry) about COVID-19 ('If this result applied to you, how worried would you be?', answered on a 7-point Likert scale marked 'not at all worried' to 'very worried').

As an exploratory measure of *trustworthiness*, we also asked participants the extent to which they found the number in the mock-up: accurate, reliable and trustworthy (each answered on a 7-point Likert scale marked 'not at all' to 'very').

More details on these measures, and others included in the survey but not reported here, can be found in electronic supplementary material, appendix S4.

### 5.2.2. Power calculation

GPower was used to determine the number of participants required to provide 95% power for small to medium-sized effects ($d = 0.3$, equivalent to CLES = 0.58) for the planned *t*-tests between subgroups in Experiment 3, for an alpha of 0.05, adjusted to correct for multiple hypothesis tests with the Benjamini–Hochberg procedure. This led to a sample size of 910 participants in the 'high risk' condition, or 1820 participants overall.

## 5.3. Results

### 5.3.1. Surveys 2 and 3: information priorities questions

In both surveys 2 and 3, participants were asked—after seeing a hypothetical result on a risk ladder, whether they would like to see such information. More people said that they would than that they would not. The results are shown in figure 8. When asked about their employer seeing such information, the majority of participants chose 'I don't know'.

When asked what numerical risk result was most important to them, the risk of dying from COVID-19 if they caught it was ranked first. Risks of hospitalization or long-term consequences of COVID-19 were further down people's priority lists, although they could be made available as 'extra information' for

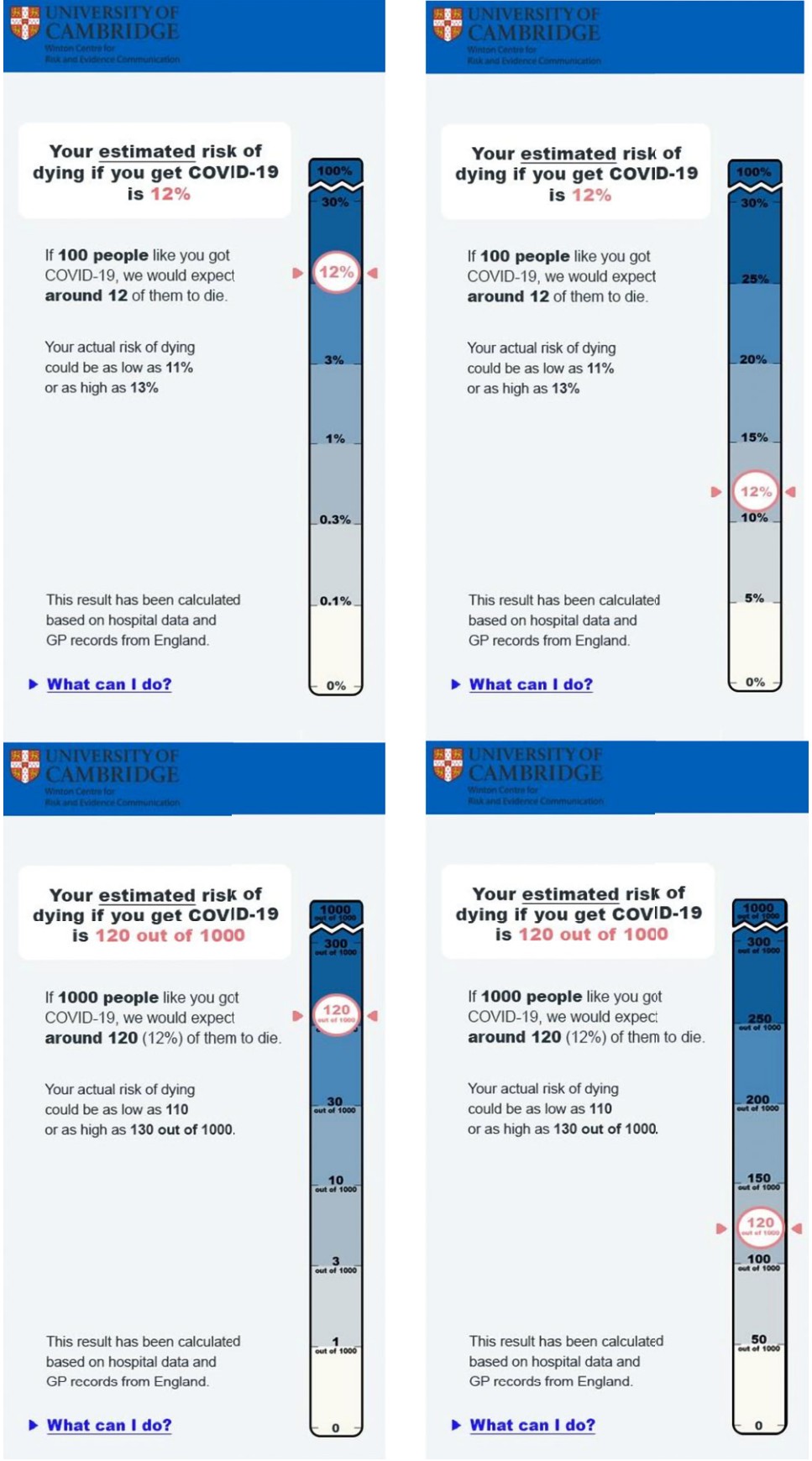

**Figure 7.** A 12% risk shown on a log or linear scale, in percentages or frequencies. Participants in Experiment 3.1 were randomized to see either a 12% or a 0.1% risk in one of these four formats.

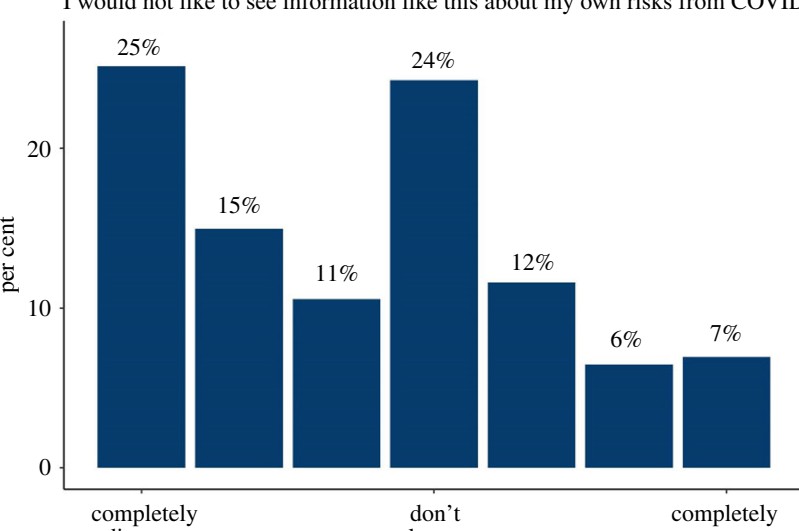

**Figure 8.** The proportion of people answering 1–7 on a Likert scale 'I would not like to see information like this about my own risks from COVID-19'. Results from surveys 2 and 3 combined (n = 2520).

those who wanted it. The view from interviews, though, was that too many numbers at once would probably be confusing.

When asked to rank possible information that could additionally be made available to them, after the risk of dying from COVID itself, the highest-ranked pieces of information was advice on risk mitigation strategies and details about how the score was calculated. These were followed by the request for information reminding people that they pose a risk to others and context about that person's risk score in comparison with everyone else in the UK. Below that was information about the uncertainty in the score and the data used for it (assessments of the quality of the evidence) (see figure 9).

### 5.3.2. Experiment 3.1

The two measures of subjective comprehension were correlated with $r_{1595} = 0.88$ and so were combined into a single measure, as were the subjective comprehension items, $r_{1601} = 0.74$. The three measures of trustworthiness had $\alpha = 0.95$ and so were combined.

The results for these four dependent variables for the four arms of the experiment (log and linear scales, with figures shown as percentage or frequencies on each), for the high (12%) risk result are shown in figure 10.

We planned separate $t$-tests to explore differences in these key dependent variables between log and linear presentations (pooling frequency and percentage conditions), and likewise between the frequency and percentage conditions (pooling log and linear presentations) for the high-risk group only, correcting for multiple comparisons with the Benjamini–Hochberg procedure. As previous literature [32] reported that the visual positioning of the risk on the scale is influential, and we expected risks to look higher on a logarithmic scale as a result, we planned these comparisons only for participants who saw the higher (12%) risk result.

After adjusting for multiple comparisons, we found no significant differences between participants who saw a 12% risk result displayed on a logarithmic versus linear scale, in terms of perceived risk ($M_{lin} = 38.89$, s.d. = 20.36, $M_{log} = 40.45$, s.d. = 21.27; $p = 0.30$), worry ($M_{lin} = 4.10$, s.d. = 1.59, $M_{log} = 4.28$, s.d. = 1.61; $p = 0.13$), subjective comprehension ($M_{lin} = 5.70$, s.d. = 1.35, $M_{log} = 5.58$, s.d. = 1.38; $p = 0.27$), or trust in the information ($M_{lin} = 4.22$, s.d. = 1.31 $M_{log} = 4.01$, s.d. = 1.32; $p = 0.06$).

Considering the effect of the format of the risk presented, those who saw the numerical information in frequency terms (120 out of 1000), compared with a percentage, perceived the risks to be significantly higher ($M_{freq} = 43.30$, s.d. = 21.47; $M_{perc} = 36.18$, s.d. = 19.56; $p < 0.001$; $d = 0.35$; CLES = 0.60) and less easily understood ($M_{freq} = 5.46$, s.d. = 1.42; $M_{perc} = 5.81$, s.d. = 1.29; $p < 0.01$; $d = 0.26$; CLES = 0.57) (figure 10). This was in line with the other experiments in this study.

Risk format did not have a significant effect on emotional risk perception (worry) ($M_{freq} = 4.32$, s.d. = 1.62; $M_{perc} = 4.06$, s.d. = 1.58; $p = 0.06$) or trust in the information ($M_{freq} = 4.03$, s.d. = 1.37; $M_{perc} = 4.20$, s.d. = 1.26; $p = 0.13$).

Out of all the information that could be available about a person's risk, which do you think is the most important?

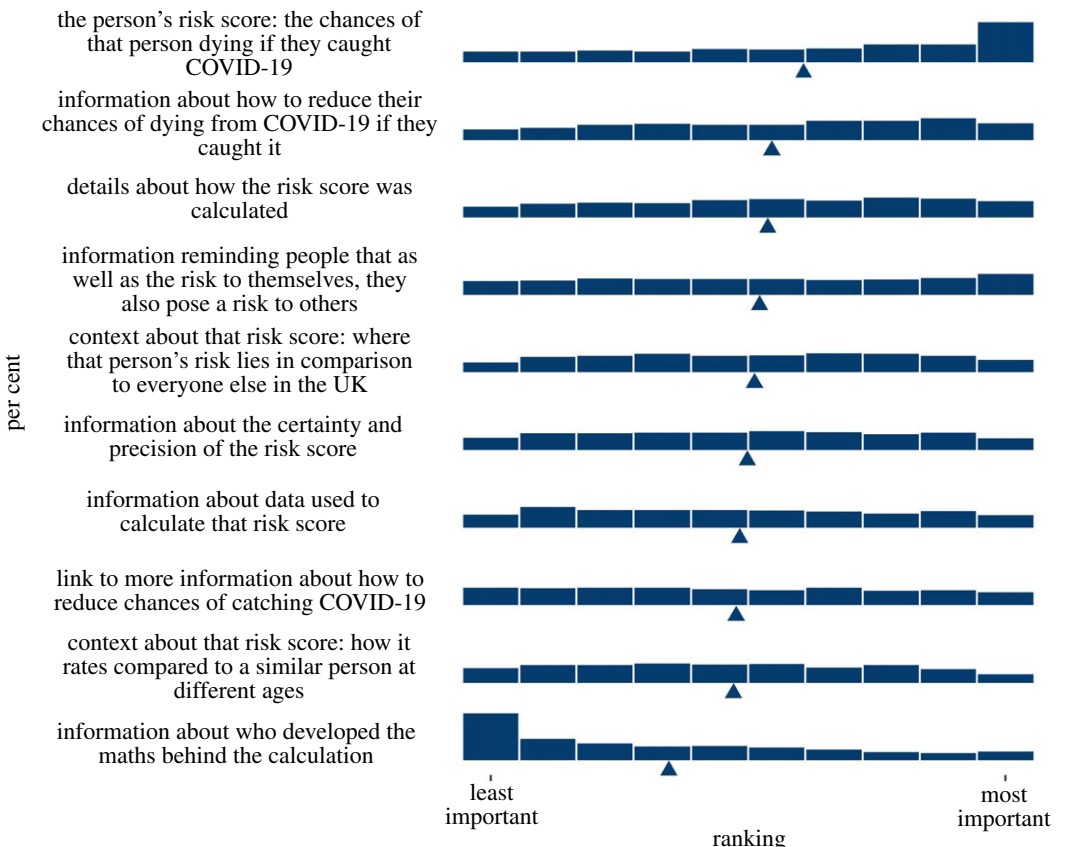

**Figure 9.** Participants' ranking of the importance of different pieces of information in a hypothetical personal COVID-19 risk communication tool. Triangle indicates mean. From surveys 2 and 3 combined ($n = 2520$).

The same analyses for those who saw the 0.1% risk result were not performed due to a mistake in the graphic for this condition shown to some participants.

## 5.4. Interim discussion

The data on the audience's information preferences confirmed the findings of survey 1: that the majority of people were keen to be given this kind of information (even, this time, when shown what that information might look like). It also confirmed the importance—if only a single number were being shown—of the risk of death if a person catches COVID-19 as the main outcome of interest.

In Experiment 3.1, the findings of the qualitative interviews were confirmed: there was a small but significant difference in trust between logarithmic and linear scales (with the linear scales being more trusted). However, there was no difference in subjective understanding or—surprisingly—risk perception. This went against our expectation that the higher position of the '12%' marker on the graphical logarithmic scale compared with the linear scale would create a perception of a higher risk.

The findings that the risks expressed as a frequency out of 1000 were found to be less clear and perceived as higher was also in keeping with previous research and the results in survey 2 (which was run concurrently).

Together these suggested that we concentrate our design on a risk ladder with a linear axis and using percentages rather than frequencies as the main method of communication of numbers on the axis. We wanted to continue work on the potential comparators that were most useful, and also have a final set of endpoints that would help us assess the communication overall in terms of Weinstein & Sandman's criteria [42] and communication efficacy [58]. These formed the core experiment within our final study, alongside some smaller experiments designed to complement the work.

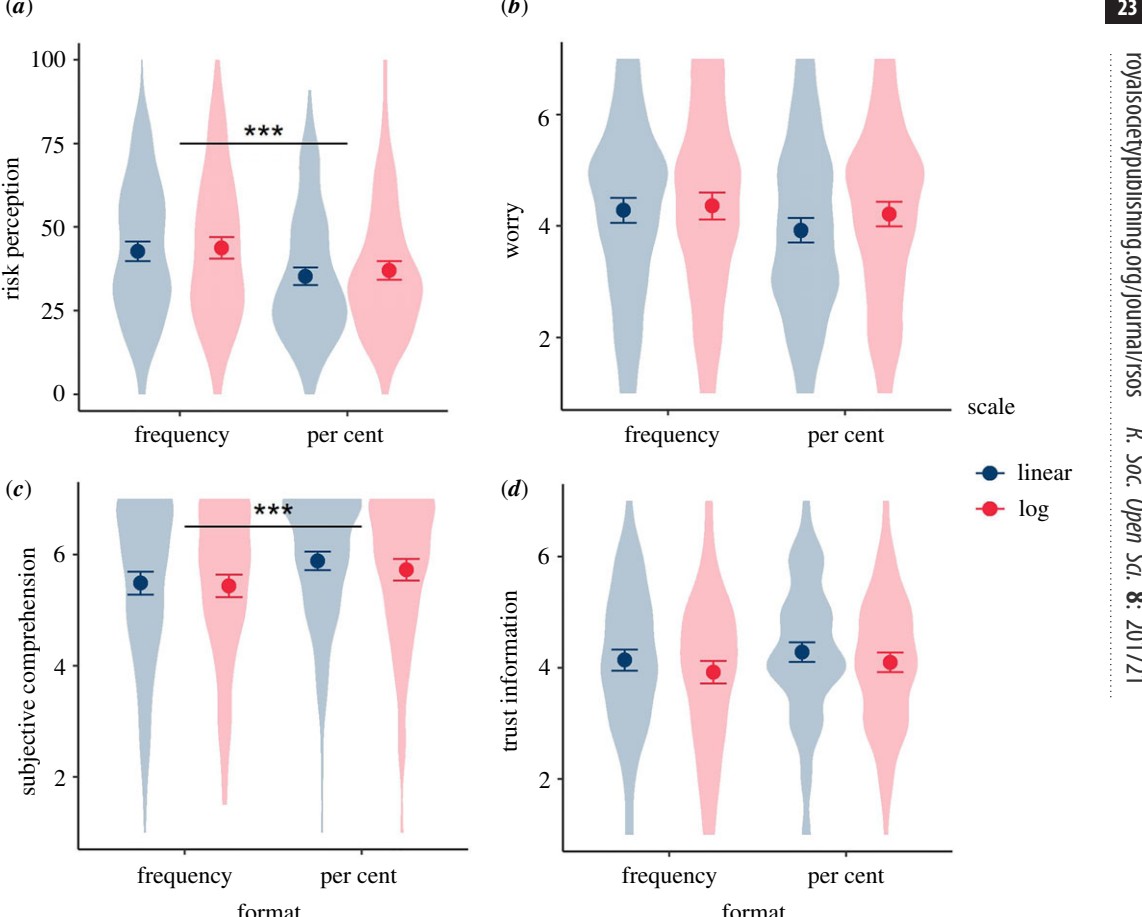

**Figure 10.** Mean risk perception (*a*), worry (*b*), subjective comprehension (*c*) and trust (*d*) among participants responding to a high (12%) risk result presented as either a frequency or percentage with either a linear or logarithmic scale. Error bars show 95% confidence intervals. 'Violin' plots represent underlying data distribution. Asterisks denote significant difference between frequency and per cent groups (no significant difference between scale formats) (survey 3, *n* = 1820).

# 6. Survey 4

## 6.1. Introduction

The main aim of survey 4 was to create a final experimental test of the potential formats for a communication of an individual's personal risk of dying from COVID-19 if they caught it.

Learning from our previous findings, we decided to use percentages as the main form of communicating the risk, but—because of the difference in perception of percentages versus frequencies—we also decided to include a single instance in which we 'translated' the percentage into a frequency format in an attempt to help balance the audience's perception of the number.

In order to give communicators a firm evidence base as to the effects of choosing different formats of frequency (e.g. '1 in x', which we had not previously tested) for the presentation of the range of relevant probabilities (i.e. ranging from 0.01% to 20%) we decided to run one experiment specifically to test the risk perception of each of four different formats: 'x in 1000', 'x in 100', 'x%', '1 in x'.

From our previous findings, it appeared that the most likely useful comparators for people to have on a risk ladder were 'personas': descriptions of people who symbolized different levels of risk. To test whether the level of description of these personas that we were considering providing was adequate to elicit a level of risk in the minds of the audience that was as consistent as possible across individual participants, we designed a second experiment presenting participants with persona descriptions and asking participants to provide a numerical probability to represent the risk they thought that that persona faced of dying if they caught COVID-19.

We additionally randomized some participants to provide their answer in a percentage format and others in a frequency format in order to elicit participants' quantitative interpretations of the level of risk associated with the different presentation formats (as opposed to previous experiments in which we tested participants' interpretations of risks that we provided quantitative information on).

Finally, as the main experiment of the survey, we set out to conduct an evaluation study to test additional options for how the communication could be structured, specifically:

— whether a visual scale was an improvement over merely providing comparators as text;
— whether personas provided a better set of comparators over having none, or the individual's chance of dying from seasonal 'flu (the only other comparator that was considered reasonable by participants in our qualitative interviews);
— whether 'positive framing' (i.e. the number of people who survive rather than the number who die) made a difference compared with the negative framing we had so far been testing.

We aimed to evaluate realistic mock-ups of these options at a range of different risk levels covering the orders of magnitude that might be communicated in such a tool (allowing us to evaluate dose–response consistency according to the criteria of Weinstein & Sandman [42]), according to a communication efficiency scale based on Scheuner et al. [58], an 'actionability' scale modified from Recchia et al. [59], and measures of trustworthiness, perceived risk, perceived uncertainty, subjective comprehension, subjective liking, worry and concern about carrying out various high-risk behaviours (a collection of endpoints informed by our qualitative interviews).

The mock-ups were designed to be 'realistically complex', that is, to include additional information that such a tool would include in the real world, including (faux) links to further information based on participants' requests in previous surveys, information about the uncertainty around the score and some information about the data on which the calculation was based.

In addition to these experiments, we added two more questions to survey 4 about the information preferences of the audience. Since we were aware that—as mentioned in the main introduction—the aim of the communicator was key to many design decisions, we were interested in where the UK public thought the aim of such a tool should sit on the 'inform–persuade' spectrum.

## 6.2. Methods

Survey 4 started with the same risk perception and COVID-19 experience questions as the previous surveys. Additionally, participants were asked whether they expected a public tool designed to communicate personalized COVID-19 risks to try to persuade people to change their behaviour, or merely to inform them, with the following question:

'A tool like this can be designed in many different ways, which give people a different impression of the numbers. The tool can be made as neutral as possible, to try to just inform people and let them make up their own mind about the risk to them and how they should behave. Or the differences between numbers can be made very obvious, making the risk look larger or smaller. This would be more persuasive and make people more likely to change their behaviour. How do you think a national tool should be designed?'

They were provided a 7-point Likert scale for their answers, with the endpoints labelled 'As neutral as possible: just inform people'—'As persuasive as possible: change people's behaviour'.

Participants who marked an answer on the scale past the mid-point and towards the 'persuasive' end of the spectrum were then asked 'In which direction do you think the tool should try to persuade people?' and given three options: 'It should try to reassure people by showing the risk is generally low'; 'It should try to make people be more cautious by showing that even if the risk is low to them, they can spread it to others'; 'It should try to be persuasive to different people in different ways (reassuring some and making others more cautious)'.

Participants were then presented with survey questions corresponding to three randomized controlled experiments:

### 6.2.1. Experiment 4.1: elicitation of different numeric formats

Participants were presented with basic descriptions of five different individuals (the same as in Experiment 2.1) in a randomized order and asked to estimate each individual's chances of dying from COVID-19 if infected. Depending on the arm they were randomized to, participants were asked:

'Type a percentage, without the percentage sign'; 'Type the number of people out of 100 exactly like [*name*] you would expect to die if they all caught it.' Or 'Type the number of people out of 1000 exactly like [*name*] you would expect to die if they all caught it.'

We compared the mean estimated risk of dying (all converted to a percentage) between conditions and personas using a 3 (response format) × 5 (persona) factorial ANOVA with Tukey HSD *post hoc* comparisons. This analysis was pre-registered at https://osf.io/w2vks.

### 6.2.2. Experiment 4.2: interpretation of different numeric formats

Participants were asked to imagine they had received a personalized estimate of the risk of dying if infected with COVID-19 and were each presented with five different risk estimates in random order. They were randomized to one of five conditions: four formats of presentation of the level of risk (percentages, '1 in x', 'x in 100', 'x in 1000' in which they were asked to respond with a sliding scale labelled 'very low risk' at one end and 'very high risk' at the other) and one condition where they received the risk as a percentage but were asked to respond instead on an 11-point Likert scale with the same labelling as the slider. This was done to compare the effects of using a slider versus Likert scale as we had some concerns that those being given numbers as a percentage might merely use the slider scale to measure out the approximate percentage of the entire slider scale distance, rather than to indicate the degree of risk that they perceived.

We compared mean perceived risk scores between the four presentation conditions (excluding the exploratory response condition with the Likert scale response; between-subjects) and five risk levels (within-subjects) using a 4 × 5 two-way ANOVA. This analysis was pre-registered at https://osf.io/rdtc4.

### 6.2.3. Experiment 4.3: interpretation of risk information with and without context, visual scale and positive versus negative framing

A 5 × 4 factorial design randomized participants to view one of five formats of risk presentation (figure 11) at one of four risk levels.

The five formats were (1) positive framing, visual scale, no comparators; (2) negative framing, visual scale, no comparators; (3) negative framing, text only, age risks as comparators; (4) negative framing, visual scale, 'flu risk as comparator; (5) negative framing, visual scale, age risks as comparators. The four risk levels were 0.01%, 0.1%, 2% and 20% risk of death.

After viewing the mock-up, participants were asked the 13-question *communication efficacy* scale (electronic supplementary material, appendix S4), the five-question *actionability* scale (electronic supplementary material, appendix S4).

They were asked three questions about the perceived accuracy, trustworthiness and reliability of the numeric information as previously used in Experiments 2.2 and 3.1 (electronic supplementary material, appendix S4). We planned to combine the three answers, if Cronbach's alpha was acceptable, into an index to describe the *trustworthiness* rating of the information. *Trust in the producers* of the information was measured via a single item ('To what extent do you think that the people responsible for producing this number are trustworthy?' answered on a 7-point Likert scale marked 'not very trustworthy' to 'very trustworthy').

Participants were asked a single question about the *perceived uncertainty* of the information ('To what extent do you think that the result was certain or uncertain?' answered on a 7-point Likert scale marked 'very certain' to 'very uncertain').

As a measure of *cognitive risk perception*, they were asked 'If the person who got this result caught COVID-19, how likely do you think it is that they would die as a result?' and 'How would you describe the risk of this person dying from COVID-19 if they caught it?', both answered via a slider with no numerical cues and 'very low risk'–'very high risk' as the endpoints. We intended to combine these two into a single index if they correlated well enough.

As a measure of *emotional risk perception*, they were asked the same question about worry as in previous experiments (electronic supplementary material, appendix S4).

We also asked participants to complete two questions capturing *objective comprehension* of the information presented, while still viewing the information: 'Approximately what percentage of people who got a result like this would you expect to die of COVID-19 if they caught it?' and 'Out of 10 000 people who got a result like this, how many would you expect to die of COVID-19 if they caught it?' Responses to both were recorded as a number typed into a box. In the positive framing condition,

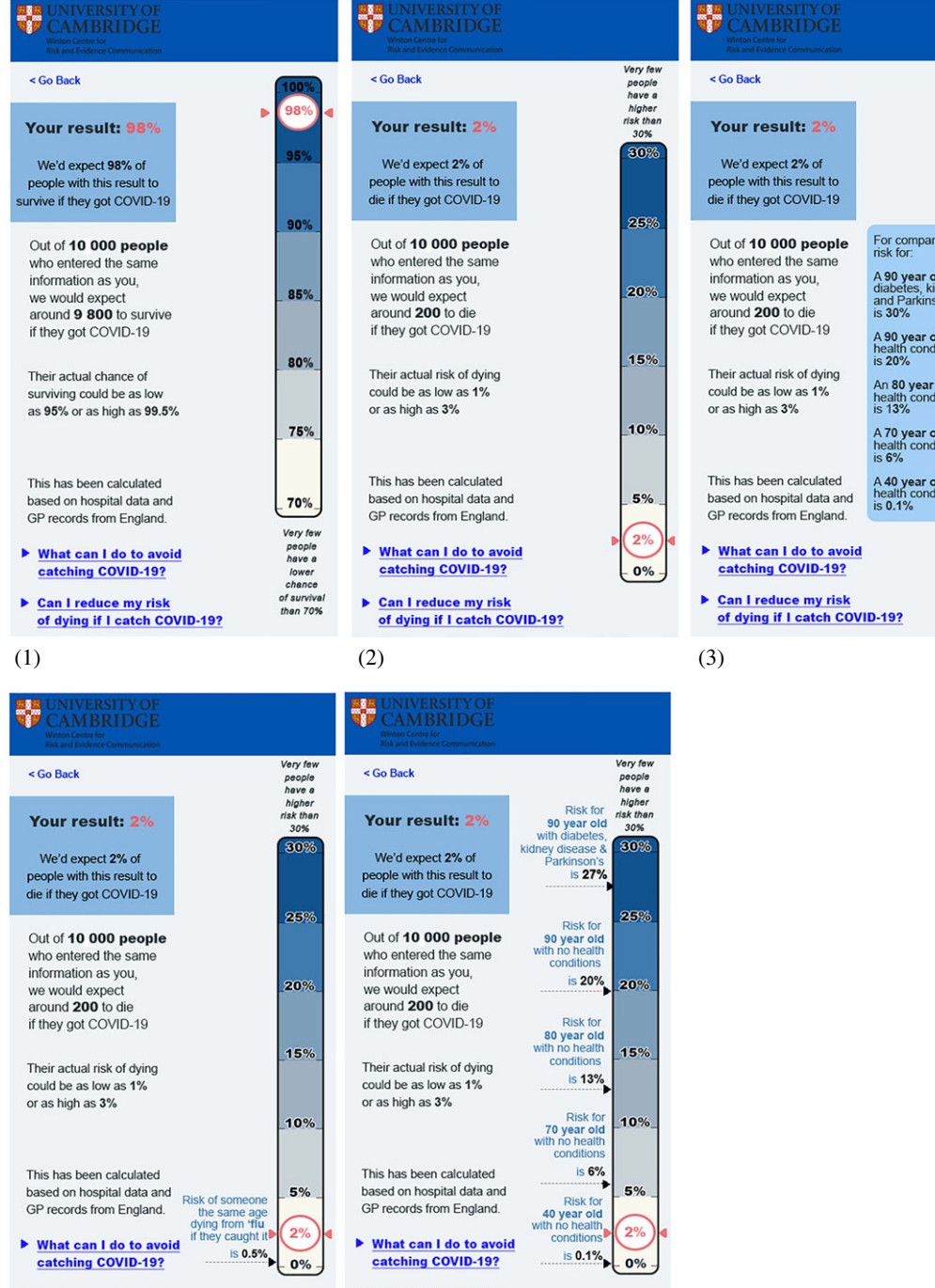

**Figure 11.** Visualization formats shown to participants in Experiment 4.1 (2% risk level category only). (1) positive framing, visual scale, no comparators; (2) negative framing, visual scale, no comparators; (3) negative framing, text only, age risks as comparators; (4) negative framing, visual scale, 'flu risk as comparator; (5) negative framing, visual scale, age risks as comparators. Planned contrasts were pre-registered between the following pairs of formats: (1, 2), (3, 5), (1, 3), (1, 4), (1, 5), for each dependent variable: risk perception, worry, communication efficacy and concern about higher-risk behaviours. Please note, the risks illustrated on these visualisations are all fictional (including the age comparators) this was made clear to participants. Please take care not to reproduce these visualisations in contexts where they may be taken as genuine.

these items were reframed as asking the percentage/frequency of people expected to survive rather than die, so as to match the information provided. The answers to these questions were directly available in the information provided (figure 11). For each of the two items, responses were coded as either correct or incorrect.

To assess *hazard–response consistency*, participants were asked: 'If this result applied to me I'd be more concerned about my risk from COVID-19 than from seasonal 'flu' (answered on a 7-point Likert scale: Completely disagree–Completely agree).

In order to assess their *concern over high-risk behaviours*, another way to try to assess their risk perception, participants were asked 'How worried would you feel doing each of the following because of the risk of catching or passing on coronavirus at the moment?' prior to viewing the mock-up and 'If you had received the result you just saw, how worried would you feel doing each of the following because of the risk of catching or passing on coronavirus right now?' after viewing the mock-up: 'Shopping in a busy supermarket; Eating indoors in a restaurant with a small group of friends; Drinking in a pub garden with a small group of friends; Going to a large cinema; Travelling on the London underground; Visiting an elderly person in a nursing home; Attending Accident and Emergency in a city hospital'. These options were presented in random order and answered via 7-point Likert scales (labelled 'not at all worried' to 'very worried').

After viewing the mock-up, participants were also asked how much they agreed or disagreed with the statement 'I think the tool should just tell people whether their risk is "high" or "low"' (answered on a 7-point Likert scale marked 'completely disagree' to 'completely agree').

Finally, participants were shown all five mock-ups, and asked to rank them, as a measure of *subjective preference*.

For a full list of question wordings and other questions asked but not reported here, see electronic supplementary material, appendix S4.

We ran four regression models in which perceived likelihood of death, worry, communication efficacy and concern about higher-risk behaviours were the dependent variables. In each, the independent variables were display format (using sum coding) and risk level (treated as a continuous variable).[3] We also conducted pairwise planned contrasts between different pairs of display formats for all four DVs, with independent variables as above. These analyses were pre-registered at https://osf.io/4yu73.[4] Exploratory analyses also investigated cognitive risk perception in more detail, trustworthiness, perceived uncertainty, hazard–response consistency, and objective comprehension.

### 6.2.4. Power calculation

Our power calculation for survey 4 was based on the experiment contained within it with the largest number of conditions and hence power requirements, Experiment 4.3, and we confirmed that powering for this experiment would provide equivalent or better power for the other experiments in the survey.

GPower was used to determine the number of participants needed in order to have 95% power to detect a small-to-medium effect size ($d = 0.3$, equivalent to CLES = 0.58) at alpha 0.05 for the main analyses and the pairwise planned contrasts. As we intended to use the Benjamini–Hochberg procedure to control the familywise error rate, we powered so as to be able to detect an effect of this size even for the most stringently corrected test of the 24 tests planned (tests of the main effect of display format on risk perception, worry, subjective clarity and behavioural intentions, plus the 20 planned contrasts). A total of 2495 subjects was found to be needed for the survey as a whole, which we rounded up to 2500.

## 6.3. Results

When asked their opinions on whether the aim of personal risk communication about COVID-19 should be as neutral as possible just to 'inform people' or whether the aim should be to 'persuade people' in order to change their behaviour, 63% of participants chose a mark on the Likert scale above the half-way point towards the tool being 'persuasive'. When this subset of participants were asked in which direction they wanted it to persuade people, 61% said that 'it should try to make people be more cautious by showing that even if the risk is low to them, they can spread it to others'.

[3]As perceived likelihood of death increased with risk level in a linear fashion, we treated risk level as a continuous variable rather than as a categorical variable. Treating risk level as a categorical variable led to the same pattern of results.

[4]In this paper, 'perceived likelihood of death' refers to the measure described in the pre-registration as 'risk perception', 'communication efficacy' refers to the measure described as 'subjective clarity' and 'concern about higher-risk behaviours' refers to the measure described as 'behavioural intentions'.

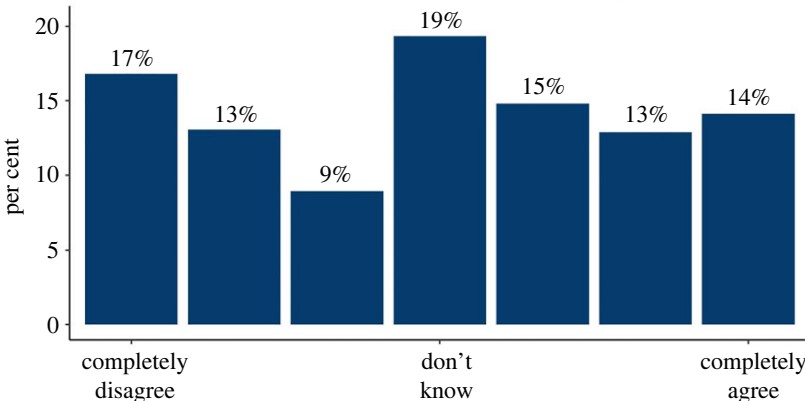

**Figure 12.** Participants' rating on a 7-point Likert scale of how much they agreed or disagreed with the statement that a personal risk communication tool 'should just tell people whether their risk is high or low'—asked after people had seen a mock-up of potential outputs giving an exact risk score (survey 4, $n = 2500$).

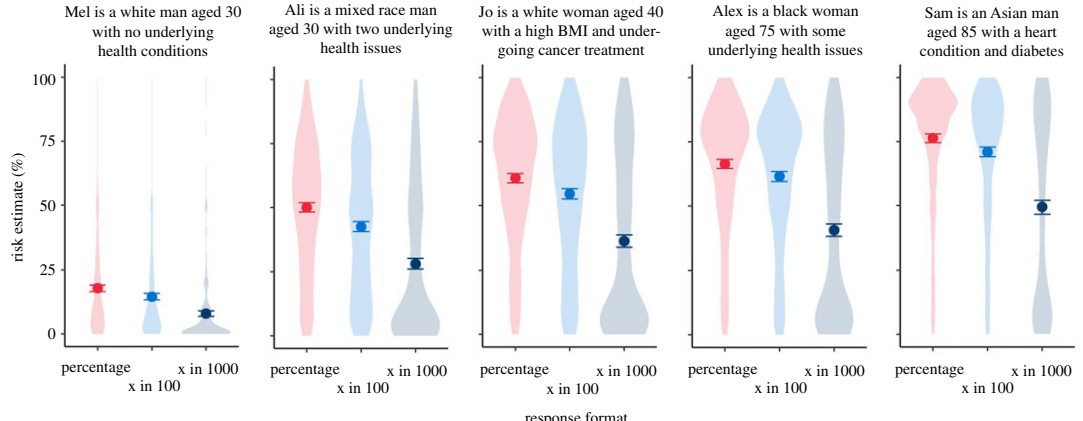

**Figure 13.** Mean (95% CI) estimates of different personas' risk of dying if infected with COVID-19. Participants provided their estimate as either a percentage, or a frequency out of 100 (x in 100) or 1000 (x in 1000); converted to percentage for comparison. Violin plots indicate underlying distribution. For each persona, all mean estimates differed significantly from each other by response format ($p < 0.01$) (survey 4, $n = 2500$).

When asked, after seeing the mock-up risk ladder showing them their individual score in a realistically complex display, participants were rather more divided on whether they would rather have just been told their score was 'high' or 'low' (figure 12).

### 6.3.1. Experiment 4.1

Participants were given the descriptions for five fictional personas in a randomized order and asked to estimate their risk of dying from COVID-19 if they caught it. The mean estimates that participants gave for the risk of each described persona, depending on the format they were asked to give their estimates in, are shown in figure 13.

As per our pre-registered analysis, we first conducted a two-way mixed ANOVA comparing participants' estimates of a hypothetical individual's risk of dying across the three response formats (between factor) and five personas (within factor).

This revealed a significant interaction ($F_{8,8824} = 39.56$, $p < 0.001$, $\eta_G^2 = 0.01$). One-way ANOVAs revealed a significant effect of response format for all personas ($Fs_{2,2260} = 69.55–175.30$, all $p < 0.001$), but this effect was more pronounced for personas with descriptions that included more risk factors and hence higher estimated risks; the effect of format was greatest for Sam ($\eta_G^2 = 0.14$; see full descriptions in figure 13 and electronic supplementary material, appendix S4) and Alex ($\eta_G^2 = 0.13$) compared with Jo, Ali and Mel ($\eta_G^2$s 0.12, 0.11 and 0.06, respectively). *Post hoc* analyses using Tukey's HSD revealed that risk estimates differed significantly between all response formats, with individuals

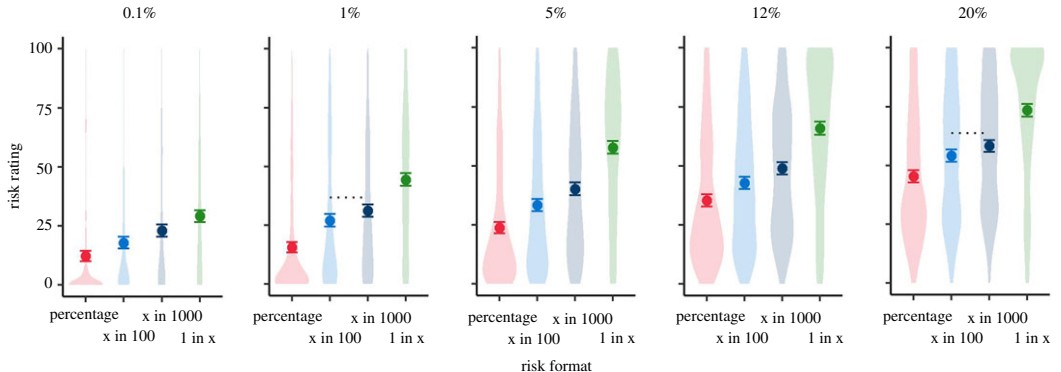

**Figure 14.** Mean (95% CI) ratings (from 'very low risk' (0) to 'very high risk' (100)) of five different COVID-19 infection fatality risk figures, presented as either a percentage or as one of three frequency formats. Violin plots indicate underlying distribution. Dotted lines indicate a non-significant difference between means, all other pairwise comparisons were significant at $p < 0.05$ (survey 4, $n = 2500$).

who entered their estimate as a percentage, on average, estimating a higher risk of death for each persona than those who gave their estimate as a frequency out of 100. In turn, estimates provided as a frequency out of 1000 were on average lower than those given in the percentage or 'out of 100' conditions (all $p < 0.01$).

In sum, when asked to estimate a hypothetical individual's risk of death from COVID-19, people, on average, gave the lowest estimates when asked to provide the figure as a frequency out of 1000, higher if out of 100, and the highest estimates if asked to provide the figure as a percentage, and these differences appear greater when the individual in question can be considered generally 'high risk'.

### 6.3.2. Experiment 4.2

To investigate how participants would perceive a range of realistic absolute risks of dying from COVID-19 (if infected), and to quantify the effects of numerical format, participants were asked to rate five different numerical risks (presented to them in a random order) on a sliding scale marked only 'very low risk' and 'very high risk' at the extremes. The position of the slider was coded as 0–100 by the survey software.

Means and distributions of participants' ratings of the risks are shown in figure 14.

As per our pre-registered analyses, we conducted a mixed 5 (risk level; within) × 4 (risk format; between) ANOVA. We report a significant interaction between conditions, $F_{12,7048} = 15.09$, $p < 0.001$, $\eta^2_G = 0.01$. One-way ANOVAs comparing perceived risk across formats for each risk level all returned a significant effect of format ($Fs_{3,1762} = 34.7$–$122.58$, all $p < 0.001$). The effect of format was greatest for ratings of the 5% risk level condition ($\eta^2_G = 0.17$) followed by 12% ($\eta^2_G = 0.14$), 1% ($\eta^2_G = 0.12$), 20% ($\eta^2_G = 0.12$) then 0.1% ($\eta^2_G = 0.06$).

*Post hoc* comparisons using Tukey's HSD revealed a consistent pattern across all risk levels: risk ratings, on average, were highest for risk levels presented in a '1 in x' format, followed by 'x in 1000', 'x in 100', then percentages. Mean ratings for nearly all formats in each risk level condition differed significantly from each other ($ps < 0.05$). The exception to this was the differences between the 'x in 100' and 'x in 1000' formats in the 20% and 1% risk level conditions, which were not statistically significant.

At risks higher than 0.1%, the absolute difference between format condition means appeared relatively consistent. For example, across all four of the higher risk levels, the average response in 'x in 100' condition was about 10% points higher than the percentage condition ($M_{\text{diff}}$ range: 9.0–11.3). Similarly, the mean responses in the 'x in 1000' condition were around 13% points higher than the percentage condition (10.8–16.5), and the '1 in x' condition mean scores were approximately 30% points higher than the percentage condition (28.7–34.0).

The exploratory condition using an 11-point Likert scale as a response format gave results not significantly different to the comparable condition using a slider response format except in the highest (20%) risk condition (electronic supplementary material, appendix S5).

Levene's test of variance equality indicated variances differed significantly between formats for ratings of the 0.1%, 1% and 5% risk levels ($Fs_{3,1762} = 17.69$, $23.90$, $11.41$, respectively, all $p < 0.001$), with

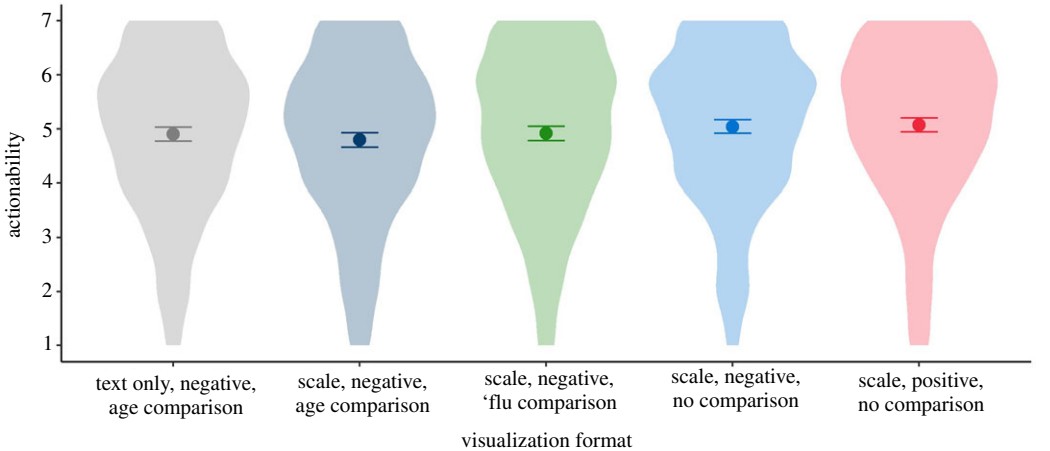

**Figure 15.** Mean (95% CI) actionability ratings for the five formats tested in Experiment 4.3. Violin plots indicate underlying distribution (survey 4, $n = 2500$).

lower variance of risk ratings among participants viewing percentage formats compared with frequency. Variances did not differ significantly between formats in the 12% ($F_{3,1762} = 2.17$, $p = 0.08$) and 20% risk levels ($F_{3,1762} = 0.79$, $p = 0.50$). See electronic supplementary material, appendix S5 for a table of pairwise comparisons.

### 6.3.3. Experiment 4.3

Following our pre-registration, we fitted regressions with display format and risk level as independent variables to look for effects on the perceived likelihood of death, emotional risk perception, concern about higher-risk behaviours and communication efficacy, as well as planned contrasts comparing five pairs of mock-ups. After alpha adjustment, we found no main effect of format on emotional risk perception nor on communication efficacy, but did find main effects of format on concern about higher-risk behaviours and perceived likelihood of death. The main effects are reported in full in the corresponding subsections.

None of the planned contrasts found significant differences. However, the stringency of our pre-registered correction procedure may have caused us to miss true effects. In cases where uncorrected contrasts found differences, this is also mentioned in the following sections, along with the results of *post hoc* tests and exploratory analyses.

#### 6.3.3.1. Communication efficacy

We found no main effect of format on communication efficacy, $F_{4,2217} = 2.1$, $p = 0.08$, $\eta_{G}^2 = 0.004$. Uncorrected contrasts suggested that the negatively framed scale with no comparator risk information scored modestly better on communication efficacy ($M = 3.27$, s.d. = 0.60) than the versions of the scale that added comparisons with individuals of different ages ($M = 3.17$, s.d. = 0.60), $p = 0.02$, or with the 'flu ($M = 3.16$, s.d. = 0.66), $p = 0.01$.

#### 6.3.3.2. Actionability

Possible scores on the actionability measure could range from 1 to 7. The formats did not appear to differ in actionability; the largest and smallest means were 5.07 and 4.85, respectively (figure 15). For context, this means that on average, participants answered questions such as 'How clear are you about what actions you could take if you had received this result in real life?', 'Do you feel you would have the necessary information to decide what actions to take if you had received this result in real life?', and so on (full list in electronic supplementary material, appendix S4) with answers that were somewhat closer to 'completely' than to 'not at all'.

#### 6.3.3.3. Trust

Perceived trustworthiness was measured via an index including perceived accuracy, trustworthiness and reliability of the numeric information ($\alpha = 0.95$). In the planned comparisons of the main pre-registered

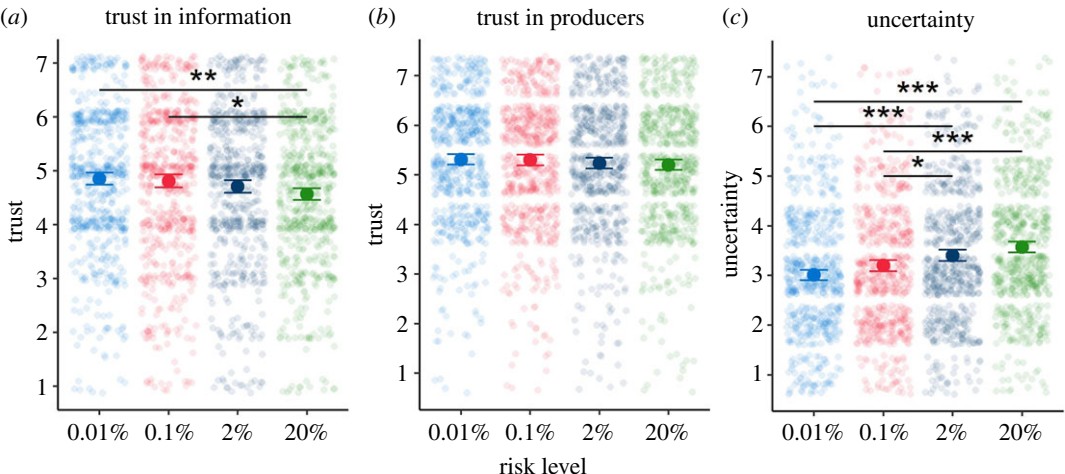

**Figure 16.** Mean (95% CI) ratings of trust in the information (*a*), trust in the producers of the information (*b*) and perceived uncertainty (*c*) (collapsed across all five formats tested in Experiment 4.3) for each of the four risk levels indicated. Jittered points indicate underlying distribution. Horizontal bars indicate significant difference between conditions, $^*p < 0.05$, $^{**}p < 0.01$, $^{***}p < 0.001$ (survey 4, $n = 2500$).

analysis, we did not find any significant differences between formats in either perceived trustworthiness of the information nor trust in the producers of the information.

Comparing effects for the various risk levels, however, revealed a significant effect for trust in the information ($F_{3,2219} = 4.87$, $p = 0.002$, $\eta_G^2 = 0.007$). Tukey HSD *post hoc* testing revealed trust being significantly lower for the 20% risk level compared with the 0.01% risk level ($p = 0.003$, $d = 0.22$, CLES = 0.56) as well as compared with the 0.1% risk level ($p = 0.015$, $d = 0.18$, CLES = 0.55) (figure 16*a*). Trust in the producers of the information showed a similar descriptive trend with trust decreasing with increasing risk levels; however, no significant differences were observed (figure 16*b*). There were no interactions between format and risk level for either perceived trust in the information, nor trust in the producers.

### 6.3.3.4. Perceived uncertainty

There was a positive relationship between risk level and perceived uncertainty of the result; the higher the risk, the higher the perceived uncertainty ($F_{3,2219} = 19.38$, $p < 0.001$, $\eta_G^2 = 0.026$). Differences were significant between all groups, except the two at the edges, i.e. 0.01% versus 0.1% and 2% versus 20% (*p* range: <0.001–0.043; *d* range: 0.15–0.44; CLES range: 0.54–0.62) (figure 16*c*). This reflects the relative sizes of the uncertainty intervals illustrated, which increased with the size of the absolute risk portrayed: 0.01% (0–0.03%); 0.1% (0.01–0.3%); 2% (1–3%); 20% (11–23%).

When contrasting the comparable positively and negatively framed visualizations (1 and 2 in figure 11) people perceived information in the negatively framed condition as more uncertain ($M = 3.32$, s.d. = 1.33) than in the positively framed condition ($M = 3.09$, s.d. = 1.32, $t_{888.52} = -2.66$, $p = 0.008$; $d = 0.18$; CLES = 0.55). See figure 17. No other comparisons were significant.

### 6.3.3.5. Risk perception

We looked at the relationship between the measures of cognitive and emotional risk perception (rescaled from 1–7 to 0–100), described in electronic supplementary material, appendix S4. These were well-correlated ($r_{2215} = 0.81$). A Levene median test indicated unequal variances ($F_{1,4437} = 59.8$, $p < 0.001$), with emotional risk perception—worry about the result—($M = 39.7$, s.d. = 30.6) having a greater variance than cognitive risk perception ($M = 28.1$, s.d. = 26.2).

*Perceived likelihood of death.* There was a main effect of format on perceived likelihood of death, $F_{4,2214} = 3.9$, $p = 0.004$, $\eta_G^2 = 0.007$. Tukey HSD tests suggested that the perceived likelihood of death was lower for those viewing the positively framed scale ($M = 2.55$, s.d. = 1.75) than for those viewing text only ($M = 2.90$, s.d. = 1.67), $p = 0.01$.

*Cognitive risk perception.* Cognitive risk perception showed a weak but significant inverse correlation with age ($r_{2201} = -0.08$, $p < 0.001$), with older participants perceiving less risk. It also showed inverse

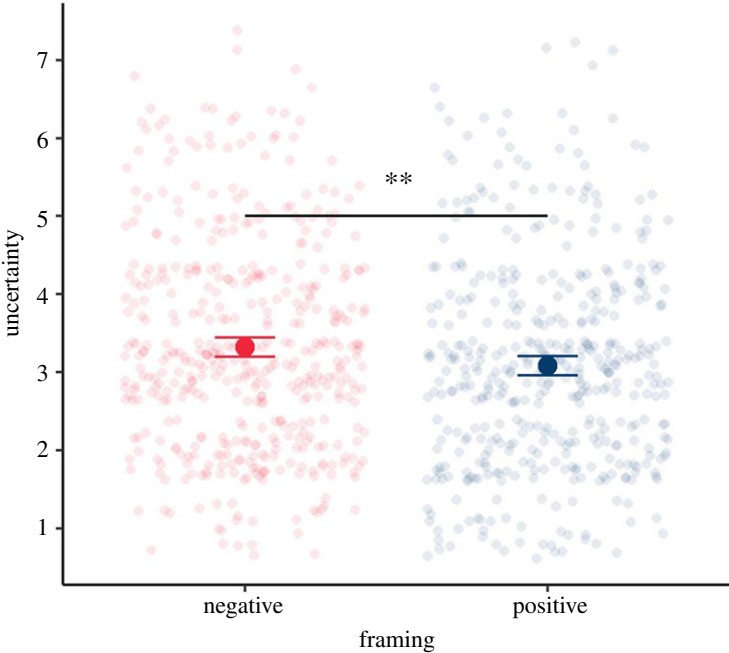

**Figure 17.** Mean (95% CI) ratings of the perceived uncertainty of the information presented in a positively framed and a negatively framed mock-up. Jittered points indicate underlying distribution (survey 4, $n = 891$).

correlations with numeracy ($r_{2195} = -0.32$, $p < 0.001$), with more numerate participants perceiving less risk, as in our earlier experiments.

*Emotional risk perception.* We found no main effect of format on emotional risk perception, $F_{4,2216} = 2.2$, $p = 0.07$, $\eta_G^2 = 0.004$. Emotional risk perception also did not show a correlation with age, but did show an inverse correlation with numeracy ($r_{2199} = -0.21$, $p < 0.001$), once again with more numerate participants perceiving less risk.

*Exploratory regressions.* The overall perception of risk associated with COVID-19 as reported prior to the experiment (assessed using an index from [5]) predicted both cognitive and emotional risk perception when entered into a regression with several covariates: sex, format, risk level, age and numeracy. Even without taking covariates into account, males perceived a lower level of overall risk from COVID-19 ($M = 4.77$, s.d. = 1.04) prior to the experiment than women did ($M = 5.08$, s.d. = 0.93), $t_{2221} = -7.27$, $p < 0.001$. Sex, age, numeracy and prior COVID-19 risk perception were, therefore, included as predictors in a set of exploratory regressions reported in electronic supplementary material, appendix S7; the format *scale, negative, no comparison* was treated as the reference level. Multicollinearity was not likely to be a major issue, as no variables in the model had correlations of greater than 0.43 and all variance inflation factors (VIFs) were less than 1.6.

The use of this relatively simple condition as the reference level allowed us to explore the potential effect of adding an age comparison (electronic supplementary material, appendix S7, row 1), a 'flu comparison (row 2), switching from a positive to a negative scale (row 3), or replacing the visual scale with a textual list of age comparisons (row 4) on various dependent variables. For example, column 1, row 4 indicates that being shown the risks with the textual list of age comparisons rather than the visual scale predicted higher cognitive risk perception, as compared with the reference level.

Analogous analyses on five other dependent variables (electronic supplementary material, appendix S7, final five columns) suggested that risk level was predictive of increasing concern in four of them. Specifically, it predicted less agreement with 'If this result applied to me, I would not be worried about catching COVID-19' ($\beta = -0.06$, $p < 0.001$), more agreement with 'If this result applied to me, I would likely be anxious and it might affect my mental health' ($\beta = +0.06$, $p < 0.001$), more agreement with 'If this result applied to me, I would do everything I could to avoid catching the virus' ($\beta = +0.04$, $p < 0.001$), and more agreement with 'If this result applied to me I'd be more concerned about my risk from COVID-19 than from seasonal 'flu' ($\beta = +0.07$, $p < 0.001$). We found a slightly counterintuitive result that higher risk levels predicted more agreement with 'If this result applied to me, I would likely change my behaviour to be less cautious of catching the virus' ($\beta = +0.02$, $p < 0.01$).

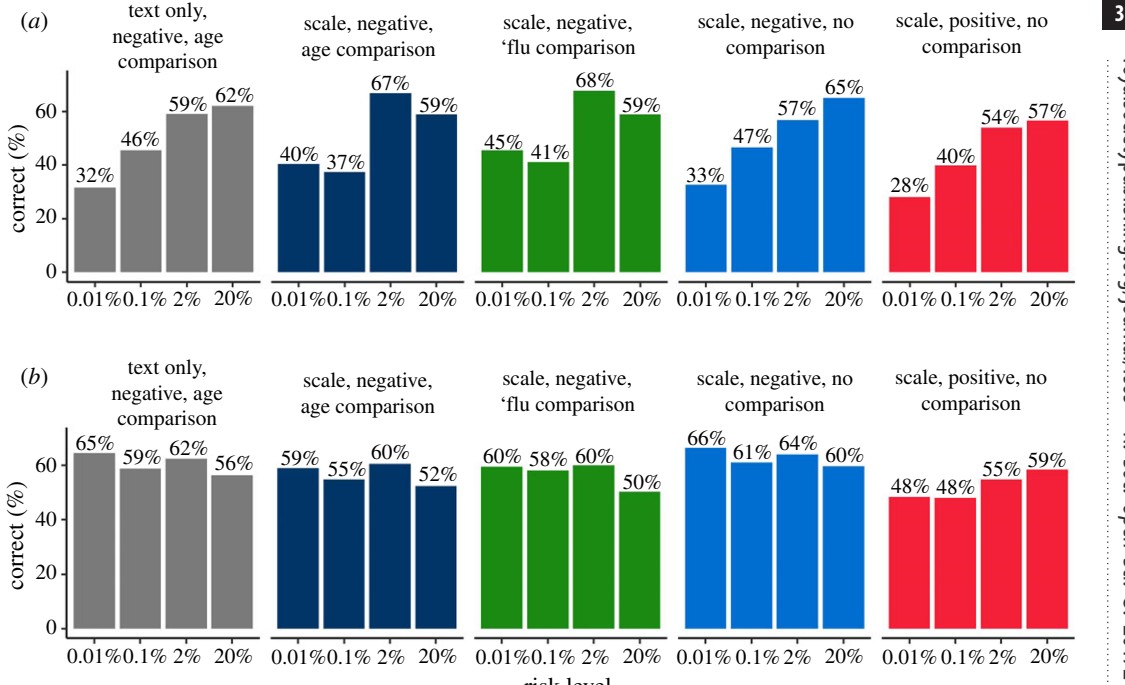

**Figure 18.** Percentage of participants in each condition correctly reporting the (*a*) percentage and (*b*) frequency presented in the information provided.

Likewise, we found that increased risk level was predictive of increased emotional risk perception ($\beta = 1.49$, $p < 0.001$), as was male sex ($\beta = 2.89$, $p < 0.05$) and high prior overall perception of the risk associated with COVID-19 ($\beta = 6.96$, $p < 0.001$). Higher levels of numeracy were predictive of decreased emotional risk perception ($\beta = -3.53$, $p < 0.001$).

Although no particular display format predicted responses on our measure of emotional risk perception, the exploratory regressions on the five 'If this result applied to me…' questions from the previous paragraph suggested that having been shown the display with the positive rather than negative framing predicted less concern overall. Specifically, having been shown the positive framing predicted more agreement with 'If this result applied to me, I would not be worried about catching COVID-19' ($\beta = 0.39$, $p < 0.01$), less agreement with 'If this result applied to me, I would likely be anxious and it might affect my mental health' ($\beta = -0.36$, $p < 0.01$), and less agreement with 'If this result applied to me I'd be more concerned about my risk from COVID-19 than from seasonal 'flu' ($\beta = -0.25$, $p < 0.05$). Betas on the remaining two questions were not significant in either direction (electronic supplementary material, appendix S7).

#### 6.3.3.6. Objective comprehension

The proportion of participants correctly reporting the risk level presented in the information (both as a percentage and a frequency) varied across format and risk level. The percentage of participants correctly answering each of the two questions, across formats and risk levels are shown in figure 18.

For each of the two comprehension items, we conducted a logistic regression with risk level and format as predictors of the log-odds of a correct response, with the text-only format and 0.01% risk level as reference categories.

Considering correct responses to the percentage question (asking participants to report the percentage of expected to die—or survive, in the positive format conditions—presented in the information): we find that participants presented with the graphical scale showing a comparison with the risk from 'flu were significantly more likely to provide a correct response relative to those getting the text-only format (OR = 1.79, 95 CI [1.07–3.04], $p = 0.028$. There were no other main effects of format.

Participants presented with higher risk levels were more likely to provide a correct response [$OR_{0.1\%} = 1.05$ [0.61–1.80], $p = 0.027$; $OR_{2\%} = 3.12$ [1.85–5.32], $p < 0.001$; $OR_{20\%} = 3.53$ [2.10–6.01], $p < 0.001$). We note one significant interaction, between the 'flu risk comparison format and the 0.1% risk level (OR = 0.47 [0.22–0.96], $p = 0.039$) where the effect of the 0.1% risk level (relative to the reference 0.01%) appeared

diminished. A similar interaction was detected for the age comparison format and 0.1% risk level, though this was not significant. Full regression results are reported in electronic supplementary material, appendix S6.

Considering correct responses to the frequency question (asking participants to report the number, out 10 000 people, expected to die/survive presented in the information), we find that participants presented with a positively framed graphical scale with no comparators were less likely to provide a correct response relative to the text-only format (OR = 0.52, 95 CI [0.31–0.86], $p = 0.011$. There were no other main effects of format or risk level.

We note one significant interaction between the positively framed graphical scale with no comparators and the 20% risk level (OR = 2.11 [1.03–4.34], $p = 0.041$) such that the 20% risk level (relative to the 0.01% reference category) appeared to have a greater effect in this format.

*Exploratory regressions.* In addition to the regressions described above, we also conducted two exploratory logistic regressions predicting objective comprehension (as percentages and as frequencies, respectively) using the same covariates and reference levels as the exploratory regressions previously reported in the section on risk perception; these are reported in full in electronic supplementary material, appendix S8. In these regressions, numeracy was predictive of higher comprehension (percentages: OR = 1.77 [1.67–1.90], $p < 0.001$; frequencies: OR = 1.79 [1.68–1.90], $p < 0.001$), as was age (percentages: OR = 1.01 [1.00–1.01], $p < 0.05$; frequencies: OR = 1.01 [1.01–1.02], $p < 0.01$) and prior COVID-19 risk perception (significant for frequencies only: OR = 1.12 [1.01–1.23], $p < 0.05$). Factors predictive of lower comprehension were having been shown the scale with positive framing (percentages: OR = 0.74 [0.55–0.99], $p < 0.05$; frequencies: OR = 0.50 [0.37–0.68], $p < 0.001$), having been shown the scale with the influenza comparison (significant for frequencies only, OR = 0.72 [0.53–0.98], $p < 0.05$) and male sex (percentages: OR = 0.73 [0.60–0.90], $p < 0.01$; frequencies: OR = 0.62 [0.50–0.76], $p < 0.001$).

### 6.3.3.7. Concern about high-risk behaviours

There was a main effect of format on concern about higher-risk behaviours, $F_{4,2217}) = 4.7$, $p < 0.001$, $\eta^2_G = 0.008$. Uncorrected planned contrasts suggested that those viewing the positively framed scale felt less concern about engaging in higher-risk behaviours (M = 4.61, s.d. = 1.54) than those viewing the corresponding negatively framed scale (M = 4.89, s.d. = 1.63), $p = 0.008$. Tukey HSD tests also suggested that that concern about higher-risk behaviours was lower for those viewing the positively framed scale than those viewing either text only (M = 5.02, s.d. = 1.53), $p = 0.001$, or the age comparison scale (M = 4.89, s.d. = 1.63), $p = 0.03$.

Participants reported at the start of the survey how worried they would be about engaging in seven different behaviours on account of the risk of catching or passing on coronavirus (e.g. scale 'not at all worried' (1) to 'very worried' (7)) (figure 19).

After being randomized to view one of the five visual mock-ups, participants were asked to indicate how worried they would be about the same behaviours if they had received the risk result communicated to them by the mock-up (either a 0.01%, 0.1%, 2% or 12% chance of dying if infected).

Adopting a pre–post design, we created an index of responses to the seven items before and after the presentation of the mock-up (pre $\alpha = 0.91$; post $\alpha = 0.94$) and collapsed scores across the different presentation formats to investigate the impact of just the risk level communicated.

A 2 (pre–post; within) × 4 (risk level; between) mixed ANOVA revealed a significant interaction between time point and risk level, $F_{3,2493} = 63.71$, $p < 0.001$, $\eta^2_G = 0.001$. Paired $t$-tests examining the difference between pre and post scores for each risk level (with Benjamini–Hochberg correction) indicated that participants who viewed a 20% risk result reported they would be more worried had they received that result (M = 5.23, s.d. = 1.41), than their actual level of worry reported prior to the experiment (M = 4.71, s.d. = 1.43; $t_{628} = -12.48$, $p < 0.001$, $d = -0.50$; CLES = 0.60). For participants who viewed a 2% risk result, there was no significant difference in terms of worry scores before and after the experiment ($M_{pre} = 4.90$, s.d. = 1.41; $M_{post} = 4.93$, s.d. = 1.52; $t_{624} = -0.72$, $p = 0.47$). For participants who viewed a 0.1% risk, reported level of worry, given the result, was lower than that reported prior to the experiment ($M_{pre} = 4.68$, s.d. = 1.53; $M_{post} = 4.86$, s.d. = 1.40; $t_{622} = 4.09$, $p < 0.001$, $d = 0.16$, CLES = 0.53). A similar effect was seen for participants who viewed a 0.01% result ($M_{pre} = 4.78$, s.d. = 1.48; $M_{post} = 4.52$, s.d. = 1.59; $t_{619} = 5.39$, $p < 0.001$, $d = 0.22$, CLES = 0.55) (figure 20).

To summarize, on average participants said they would be more worried than they currently are if they were told they had a 20% chance of dying if infected with COVID-19. Conversely, if they were told they had 0.1% or 0.01% chance of dying, they would be less worried than they currently are.

How worried would you feel doing each of the following because of the risk of catching or passing on coronavirus at the moment?

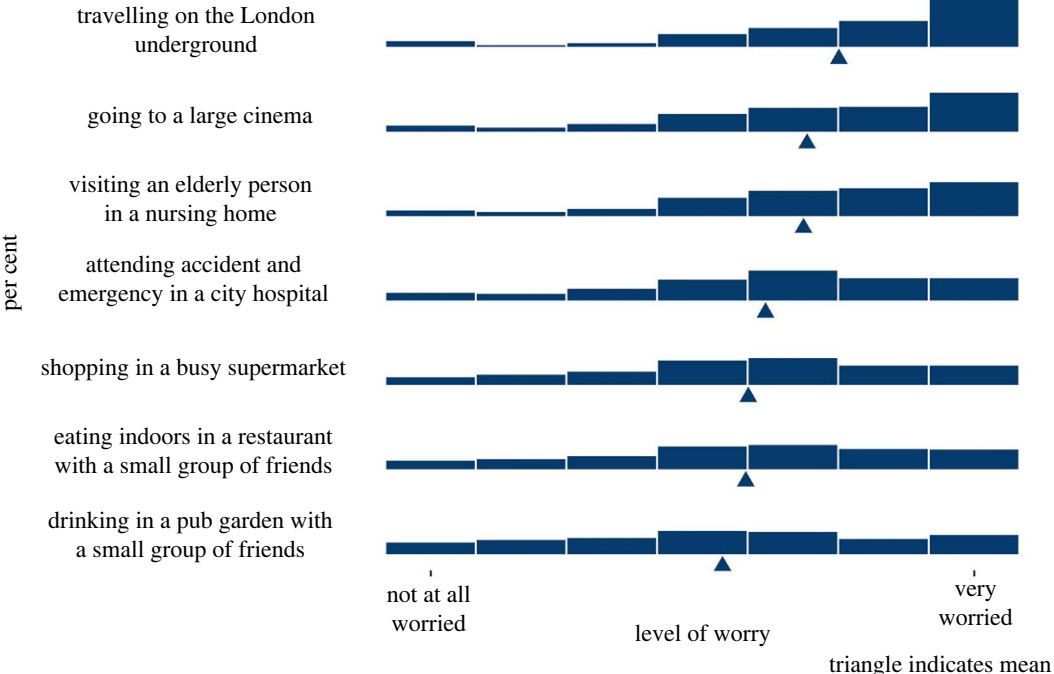

**Figure 19.** Distribution of participants' ratings of how worried they would be (on a 7-point Likert scale) to carry out each kind of behaviour because of the risk of catching or passing on coronavirus (UK participants, July 2020, asked before having seen any risk information about the virus) (survey 4, $n = 2500$).

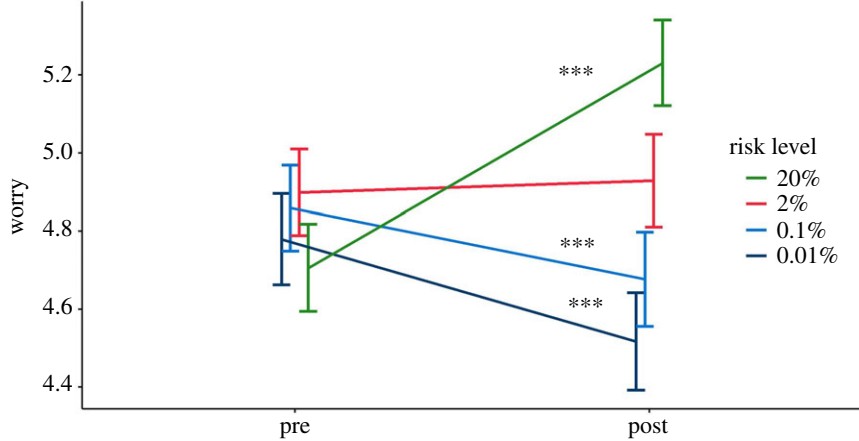

**Figure 20.** Mean actual worry before (pre) and hypothetical worry after (post) receiving risk result. Error bars represent 95% CI. Asterisks indicate a significant difference between pre and post means. $^{***}p < 0.001$ (survey 4, $n = 2500$).

### 6.3.3.8. Hazard–response consistency

When communications are evaluated on this measure, 'the goal of the communication is usually to help people see that one hazard is a considerably greater or smaller risk than another' [42, p. 110]. To get a sense of whether hazard–response consistency was present, we asked participants to what extent they were more concerned with catching COVID-19 than seasonal 'flu.

In an exploratory analysis, we found no significant difference between the responses on any of our independent variables of those who were shown the comparative risk of seasonal 'flu ($M = 4.59$, s.d. = 1.93) and those who were shown the same visualization without this comparison ($M = 4.41$, s.d. = 1.94), $t_{896} = 1.397$, $p = 0.2$; agreement was high in both cases. Similarly, seeing the comparative risk of seasonal 'flu was not predictive of responses to this question in the exploratory regression on this dependent variable (electronic supplementary material, appendix S7).

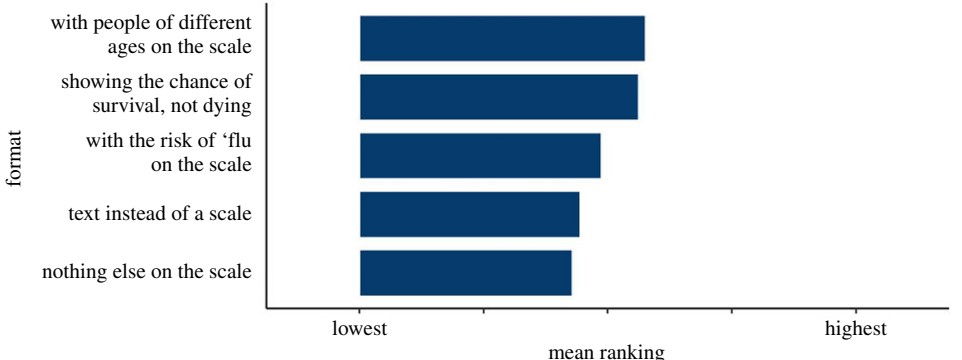

**Figure 21.** Participants' preferences across the five presentation formats tested in Experiment 4.3 when shown all five and asked to rank them (survey 4, $n = 2500$).

### 6.3.3.9. Subjective preference

When participants were shown all five visual format options and asked to rank them in order of preference, the formats with visual scales were clearly preferred (figure 21).

### 6.3.3.10. Additional findings from exploratory regressions

In exploratory regressions (electronic supplementary material, appendix S7), higher levels of numeracy predicted higher ratings of communication efficacy ($\beta = 0.03$, $p < 0.001$), lower ratings of actionability ($\beta = -0.04$, $p < 0.05$), higher ratings of subjective clarity ($\beta = 0.12$, $p < 0.001$) and subjective comprehension ($\beta = 0.17$, $p < 0.001$). Not surprisingly, then, higher numeracy participants generally found the formats easier to read and understand. Increased age was predictive of higher ratings of communication efficacy ($\beta = 0.003$, $p < 0.001$), actionability ($\beta = 0.01$, $p < 0.01$).

Additionally, higher risk levels were very slightly but significantly predictive of lower ratings of communication efficacy ($\beta = -0.01$, $p < 0.001$) and actionability ($\beta = -0.01$, $p < 0.001$), perhaps suggesting that participants who were shown high levels of risk found the tool to be marginally less helpful; they also showed greater objective comprehension on 'percentage' questions ($\beta = 0.05$, $p < 0.001$), perhaps indicating that individuals shown very low probabilities had difficulty expressing these low probabilities as percentages.

## 6.4. Interim discussion

The majority of the UK population wanted a tool giving people their personalized risk from COVID-19 to be persuasive, so that it would make others more cautious in their behaviour. This is against a backdrop of high levels of concern about the virus that we saw in the answers to the four surveys we carried out during the term of this study, and previously [5].

When given five different personas and asked to rate the chances of each person dying if they caught COVID-19 (as a percentage, as a frequency out of 100 or as a frequency out of 1000) in Experiment 4.1, it was clear that although people broadly recognized the main risk factors, as our interviews had suggested, they estimated fatality rates to be very high. As suggested in previous literature, frequencies appeared to represent higher risks to people than percentages (and 'x out of 1000' higher than 'x out of 100')—in this case shown by participants who were asked to rate the chance of death in percentage points inputting a higher fatality rate than those who were asked to rate the chance as a frequency (as each percentage point appeared to convey a lesser sense of increased risk than each additional single person 'out of 100' in the frequency format and so participants needed to input a higher percentage to convey the same level of risk).

Experiment 4.2 confirmed the elicitation experiment reported above, and revealed that the '1 in x' format is consistently rated as conveying a greater risk than the others we tested, for the range of values explored. Note that when rating how they perceived a risk of 1%, those in the 'x in 100' and '1 in x' presentation conditions were both presented with the risk '1 in 100', and yet rated it, on average, very differently. This is an effect of context—comparison with other risks they were presented with within the experiment (the presentation order was randomized). When only the subset of participants who

saw the 1% value first was analysed (removing contextual effects), there was no difference between the ratings of that value by those who were randomized to the 'x in 100' and the '1 in x' groups. The fact that the variance around the answers of those rating the risks displayed as a percentage was lower than those rating them as a frequency does lend credence to the idea that these participants were perhaps using the answer slider as a 0–100 number line (even though it was not explicitly labelled), and hence were judging the percentage as a distance along this line. However, the response of participants using a Likert scale rather than a slider was comparable, and the reverse (elicitation) method used in Experiment 4.1 was also designed to counteract that potential effect and also showed frequency presentations to convey a greater sense of likelihood, giving us confidence that this is a robust finding.

Experiment 4.3 was the culmination of our study, in which we hoped both to evaluate the effectiveness of our overall design in communicating an individual's risk, and also to determine the effects of a few remaining key design decisions.

Weinstein & Sandman [42] suggested a number of criteria that should help define effective risk communication: objective comprehension, agreement with recommendations/advice, dose–response consistency, hazard–response consistency, uniformity in response, audience evaluation (subjective measures) and a regard for types of failure of the communication and how acceptable those might be.

The findings of Experiment 4.3 can be used to assess our mock-ups according to those criteria:

(1) Objective comprehension: When assessed by asking the question 'Approximately what percentage of people who got a result like this would you expect to die of COVID-19 if they caught it?', our primary analysis (electronic supplementary material, appendix S7) did not find that positive framing condition was a significant predictor relative to the text-only negatively framed condition, though exploratory logistic regressions including further independent variables (electronic supplementary material, appendix S8) did find that viewing the positive scale predicted poorer comprehension as compared with the negatively framed visual scale. When assessed by asking the question 'Out of 10 000 people who got a result like this, how many would you expect to die of COVID-19 if they caught it?', participants were less likely to provide a correct response if they had viewed the positively framed visual scale than if they had viewed the (negatively framed) text format (electronic supplementary material, appendix S6), or the negatively framed visual scale (electronic supplementary material, appendix S8; see also figure 18). There may be an effect of familiarity with negative framing in media reports—the graphs reveal interesting patterns of comprehension which may relate to people's prior beliefs about likely death and survival rates from COVID-19, and present interesting avenues for further research. Another possible part of the explanation may be that people pay more attention to negative stimuli. Research has suggested that negative information is more thoroughly processed compared with positive information [60], and that people are loss averse [61,62], meaning they are more sensitive to losses compared with gains, which might explain attentional differences to the negatively framed information versus the positively framed information.

Results were more ambivalent with respect to the visual scale with the influenza comparison, with the primary analysis finding a possible advantage (versus the text-only condition, only for percentages) and the exploratory regressions a possible disadvantage (versus the same scale without the 'flu comparison, only for frequencies).

(2) Although the tool was not giving specific advice or recommendations, we examined whether it affected people's levels of concern about specific behaviours (such as shopping in a crowded supermarket) and found that the portrayed risk level did affect the level of worry that participants expressed about carrying out these different behaviours, suggesting a degree of (hypothetical) behavioural intent as a result of seeing the communication.

(3) Dose–response consistency was good: those who were presented with higher levels of risk perceived the risk to be higher, as measured by both their perceived likelihood of the risk (objective risk perception) and the levels of worry they expressed about the risk (subjective risk perception).

(4) Hazard–response consistency refers to whether 'people facing a hazard that is higher in risk perceive the risk as greater and/or show a greater readiness to take action than people exposed to a hazard that is lower in risk' [42]. To get a sense of whether hazard–response consistency was present, we asked participants to what extent they were more concerned with catching COVID-19 than seasonal 'flu. There was no difference in the responses between those who saw the comparative risk of seasonal 'flu and those who didn't in Experiment 4.3: in both cases, the mean response indicated agreement (rather than disagreement) with the idea that COVID-19 risk was more

concerning than the risk from seasonal 'flu, suggesting that hazard–response consistency was present irrespective of whether the 'flu comparator was included.

(5) We found no differences between the variances in the cognitive or emotional risk perception responses to the five different formats tested in Experiment 4.3, implying that they were all roughly equivalent in how homogeneously they were interpreted by participants, and there was also no significant difference between the variances in the responses to the formats tested in Experiment 4.3 and participants being given the bald absolute risks as a percentage in Experiment 2.1 (testing the 0.1% and 20% risk levels individually). This suggests that the communication via the formats we tested was as uniform in its messaging as communication via a simple percentage.

(6) Regarding subjective audience response, the negatively framed scale with age comparators was preferred when the formats were overtly ranked, but there were no significant differences between the subjective ratings of clarity and understanding between the different formats.

(7) Possible failures of the communication would be most serious if people misunderstood their risk level and were either greatly worried or greatly over-reassured as a result. As figure 20 shows, participants imagining that they had been told they had a 20% chance of dying if they caught COVID-19 would be considerably more likely to be concerned about their behaviour as a result, while those imagining they had been told the lower risk levels were less likely to be concerned about their behaviour. So, failures in either direction could have an impact on worry and behaviour. The most problematic format in this regard may be the scale with positive framing: there is some evidence that it was more poorly comprehended, and also evidence that those who viewed it were less concerned: they demonstrated a lower perceived likelihood of death, and less concern about engaging in high-risk behaviours. Exploratory regressions also suggested they exhibited more agreement with 'If this result applied to me, I would not be worried about catching COVID-19', less agreement with 'If this result applied to me, I would likely be anxious and it might affect my mental health' and less agreement with 'If this result applied to me I'd be more concerned about my risk from COVID-19 than from seasonal 'flu'.

'Actionability' is a key component of good risk communication: the audience need to know what they can do with the information they have just received (e.g. [63,64]). Unsurprisingly, our mock-ups didn't differ significantly in actionability, given how similar they were in the information they gave about potential actions that could be taken, but all achieved means well above the mid-point of the scale.

In terms of the remaining design decisions, we had hoped to clarify whether presenting the risk information in a visual format (such as a risk ladder) had an advantage over a purely text-based format. Research has long emphasized the advantages of visualizing numerical information, providing at-a-glance estimation and context [19]. In the case of very small risks, and those where comparative information is likely to be helpful, risk ladders are usually recommended (e.g. [19]). Our exploratory regression suggested that participants viewing a risk ladder visualization had lower perceptions of the risks as compared with the text-only version—whether that is desirable or not is a matter of the communicator's aims. However, subjectively, the audience preferred the presentations with visualizations.

We also hoped to determine the effects of positive versus negative framing. Although we saw no significant differences for our measure of emotional risk perception, levels of agreement/disagreement with questions such as 'If this result applied to me, I would not be worried about catching COVID-19' suggested that the positive framing reduced concern (which again, may be seen as either desirable or undesirable by different communicators), and our regression analyses suggested that it may have reduced comprehension as well.

Finally, we hoped to determine whether comparators were useful, and, if so, whether 'personas' displaying different risk factors (mainly age) or the individual's own risk of dying from 'flu provided more useful comparisons. The audience appeared, subjectively, to prefer the personas of different risk factors, in line with our qualitative interviews.

# 7. Discussion

This extensive study, employing a mix of qualitative interviews alongside pre-registered quantitative experiments, was designed to help provide professional communicators with an empirical evidence base from which to inform design decisions when trying to communicate individuals' personalized

risk of dying from COVID-19. Within the study we investigated a number of areas where we hope our findings will prove of use and here lay out what we consider to be our main findings:

## 7.1. What are the information needs of the public? Perception of risk of dying from COVID-19

When asked to give a numerical estimate, the UK population surveyed tended to greatly overestimate the absolute risks of dying from COVID-19 if infected (for example, in Experiment 4.1, the chance of death of a persona with no risk factors was deemed by many to be around 5–20%, while higher risk personas were rated as having above 50% chance of death). This overestimate of the likelihood could be the result of a combination of factors such as the risk being 'dread' (new, little-understood, potentially fatal, etc.) [1], and extremely well-covered in the media, causing availability bias [65]. However, it is also likely that for many people 'average' risk 'feels like' 50%, or the middle of a slider—similar to the well-known effect of '50 : 50' meaning 'I don't know' [66,67]. Several of the free-text comments that people gave suggested that some participants thought in this way. Participants' estimates of their own chance of dying from COVID-19 if infected (given their own major risk factors) was also often inaccurate (survey 1).

All these attempts to elicit a subjective feeling of risk are subject to problems of context. When asked to give an answer via a position along a slider bar, how do people interpret the slider: their own risk within the range of risks from COVID-19 faced by the UK population as a whole? Or their own risk from COVID-19 as compared with their own risk of death from another cause? The qualitative interviews revealed that people generally do not think numerically about their risk of dying from COVID-19, thus we expect that very few are likely to think of it as 'a percentage chance of death'. Efforts to communicate risk precisely should aim to help people translate from the precise numerical language to their own subjective experience [53].

## 7.2. Which format should probabilities be presented in? Perception of risk conveyed by numerical formats

The first part of communicating a numerical concept is the format of the numbers used. In accordance with previous findings (e.g. [24–28]), in Experiment 4.2 (and also in other experiments within the study) we found that the same number expressed as a percentage, as 'x in 100', 'x in 1000' and '1 in x' were perceived as representing increasingly higher levels of risk. The '1 in x' format has long been known to evoke higher risk perceptions than other formats [24,68]. Differences between the other formats are usually ascribed to 'ratio-bias' and 'denominator neglect' [25], where the numerator is more salient than the denominator. We were able to quantify the difference in risk perception as an 11%, 13% and 30% point increase from the perception of the percentage format, respectively. Our experiments as a whole covered four orders of magnitude (from 0.01% to 20%)—the range of numbers most likely to be used in this particular context—but the smallest percentage in Experiment 4.2 was 0.1%, and the relatively high perception of the risk when presented as '1 in x' may reduce as the risk gets smaller (and hence 'x' gets bigger), as there may be a trend in decreasing difference between the formats at the 0.1% risk level, suggested in figure 14.

## 7.3. How should context be provided for the numbers?

Previous literature has stressed the need for an appropriate choice of comparator when providing context for a risk [30,31,33–35]. We approached the problem through qualitative interviews followed by quantitative testing. Participants in the qualitative interviews rejected the concept of comparing COVID-19 with most other causes of death, with only seasonal 'flu being considered a similar kind of risk (possibly because of coverage in the media as a comparison). Our results from Experiment 2.1 suggested that while giving comparative information in the form of the risks of COVID-19 to people of different ages appeared to assist people make an assessment of the risk, giving contextual information in the form of a description of the person at risk (including their main risk factors), had a greater effect (particularly at the higher risks, where the 'persona' carried more information about additional health conditions). This echoed the findings of the qualitative interviews, where interviewees tended to think of risk in the form of personas—having in their heads an impression of the risk factors of a 'high risk' and 'low risk' person. Providing numerical absolute risks for these kinds of instinctive personas may help calibrate the perception of 'high' and 'low' risk to the numerical scale.

## 7.4. Should the numbers be visualized and how?

Another way to help translate numerical concepts more instinctively, particularly for those of lower numeracy, is to use visual representations [69,70]. We attempted to combine persona descriptions and age comparators with a visual scale, to maximize the potential for assisting comprehension of the risks.

Adding the linear visual scale alongside the presentation of absolute risks appeared to reduce participants' perceptions of the risks in exploratory regressions comparing the two matched formats in Experiment 4.3 ('age comparators, text-only' versus 'age comparators plus visual scale'). This is in contrast to findings from Lee & Mehta [71], who found no difference in risk perception between written and visual risk communication in a similar scenario, and Siegrist *et al.* [26] who found that a visual scale increased perception of risk. However, in both these studies, the logarithmic 'Paling perspective scale' was used, and in the latter study the 'text only' condition did not convey any comparator risks, unlike the scale tested in our experiment.

We did not see a difference in perception of the risk based on its position on the scale (in Experiment 3.1) as expected following the work of Sandman *et al.* [32] when testing logarithmic and linear scales, but we did find that the logarithmic scale was slightly less trusted, which again added to the comments in qualitative interviews around logarithmic scales seeming potentially misleading.

Adding a visual scale did not seem to enhance objective comprehension of the risk (either as a percentage or as a frequency) in Experiment 4.3—although participants could have been reading this information from the text-only part of the communication rather than the visual part.

Following suggestions from the qualitative interviews, we made sure that where the top of the scale was cut off, this was adequately explained (very few people have a risk higher than x), and that a persona of the sort of person whom participants could imagine as the highest risk was placed near the top with their absolute risk shown, to underline this.

## 7.5. Should positive or negative framing of the numbers be used?

Positive and negative framing is known to be able to cause a difference in people's perceptions of risks (e.g. [40]). In Experiment 4.3, we found that people who had been shown the number likely to survive rather than the number likely to die found that the risk seemed lower and was less worrying when it came to concern 'if this result applied to me…'. It was also relatively well-liked (though not the favourite) when five formats were ranked by participants. However, participants seeing positive framing may have also comprehended the numbers more poorly. This may have been for a number of reasons, such as that a negative framing is more familiar to people in the context of COVID-19 (e.g. media and governments typically reporting deaths, and not survival rates), and is only an exploratory finding.

## 7.6. Trustworthiness

Interviews suggested that trustworthiness was a crucial dimension to consider—something previously emphasized in the risk communications literature (e.g. [72]). In line with previous research (e.g. [73]), our interviews suggested that trustworthiness and relevance were enhanced by appropriate branding and making it clear that the results presented were based on research (ideally from a trustworthy source) and on relevant data. Previous experience (e.g. [73]) suggests that people are sensitive to cues of quality of evidence (such as sizes of datasets and relevance of the population on which the evidence is based), as well as assessing the information's source.

As already mentioned, in Experiment 3.1, we saw that a linear scale was significantly more trusted than its logarithmic alternative. In Experiment 4.3, we found that although trustworthiness was not affected by the different graphical formats presented (now using linear scales only), it was affected by the level of risk being communicated. Prospect theory [61,74] and negativity bias [60,75] predict that losses loom larger than gains, and research by Slovic [72] showed a 'trust asymmetry' where bad news affects trust more than good news does. Eiser & White [76] found that this asymmetry was more pronounced for more concrete and tangible events, as well as greater hazard risk potential. This may help explain the effect of the highly negative information (i.e. the high risk of death) over the more positive information (i.e. the low risk). Additionally, denial or avoidance are ways of coping with stress [77,78], and trusting more negative information less could help to psychologically mitigate a potential self-threatening situation.

## 7.7. Other findings

In exploratory regressions in Experiment 4.3 (electronic supplementary material, appendix S7), male sex was also predictive of higher cognitive risk ratings, which may seem surprising given that regressions also show that male sex was predictive of lower worry in responses to questions about engaging in various 'higher-risk' behaviours such as going to a large cinema, and given that men perceived a lower level of overall risk from COVID-19 than women, in line with previous literature [79,80]. One possible explanation is that ratings of risk to individuals are relative to prior expectations: if a person perceives COVID-19 to be very risky, then they may rate even a 20% risk of death as 'low' because it is lower than their expectations. However, in our data, higher prior COVID-19 risk perception was predictive of higher cognitive risk ratings. Another explanation could be that males and females differ in both their cognitive assessment of the likelihoods of death (with males being higher) and their worry about that likelihood (with females being higher). However, the picture may be more complex, as our regressions suggested that being male predicted higher levels of emotional risk perception (worry), despite predicting lower levels of concern about higher-risk behaviours.

Older age was predictive of lower cognitive risk ratings but not emotional risk rating (worry), which again is slightly counterintuitive as one might expect the same absolute risk to feel less risky to an older person given the lower relative risk that it would present to them.

# 8. Conclusion

This series of rounds of qualitative interviews and quantitative experiments give useful insights into how personalized information about the risks of COVID-19 might be clearly communicated to individuals.

Communicators first need to consider where their aims fall on the spectrum from *purely informing individuals of the risk* to *outright persuasion to follow a certain behavioural outcome*. Many of the design and communication choices that need to be made will be based on whether the communicator wants the message to be persuasive, and if so, in what direction.

Many of the effect sizes that we found are small (apart from those around the difference that the different format of the number, such as percentages versus '1 in x' makes), as is common for experimental work on risk communication. However, when scaled up to population levels, for instance for mass communications, where tens of thousands (or more) people are forming the audience, small effect sizes can still have an impact on outcomes that is worth considering.

Here, we provide some guidelines based on our findings:

## 8.1. How to express the numbers

Percentages appear to be the clearest format, having the smallest variance in perceived risk. They also make the risk seem lowest, with chances expressed as 'x out of 100', 'x out of 1000' or '1 in x' conveying increasingly higher likelihoods (on the range of orders of magnitude we tested—down to 1 in 10 000). In order to provide balance, then, communicators might choose to use both a percentage and a frequency with a large denominator for the main risk score. However, beware using too many numbers on a single scale or presentation format as they can become visually busy and overwhelming, which can have negative effects on comprehension due to cognitive load [81].

The result should not be described as 'your risk' as statistics necessarily rely on limited and incomplete information, describing subpopulations rather than individuals. We suggest using 'Risk result' or 'Risk level' as descriptions.

## 8.2. Framing: the number of people who die or the number who survive?

Participants subjectively liked being shown the number of people likely to survive rather than the number likely to die, and seemed to find the result less concerning. However, participants seeing positive framing may have also understood the numbers less well. We would recommend using negative framing but including a single translation to the positive within the format (e.g. 'We would expect 2 in 1000 people to die… that means that 998 out of 1000 would survive'), once again with the caveat that this may decrease comprehension if there are already many numbers being presented.

## 8.3. Using a visual scale: log versus linear

Despite the difficulties of representing several orders of magnitude on the same scale, participants generally found the linear scale more easily understandable and slightly more trustworthy.

Representing multiple orders of magnitude on a linear scale generally requires cutting it at a suitable maximum point, and participants in qualitative interviews stressed the importance of explaining why that maximum point had been chosen and using a description of the type of person who would have that very highest level of risk to help them calibrate their perceptions to the numerical scale in front of them.

## 8.4. Using a visual scale: colour

Colour can affect the impact and interpretation of a scale. In this study, we did not investigate these effects systematically. Issues of accessibility to those with visual impairments as well as the principles of good design should guide the use of colour, as well as empirical testing.

## 8.5. Giving context

An absolute risk figure (e.g. '2% chance of dying if you catch COVID-19') is, on its own, generally unhelpful to members of the public. The public are very unfamiliar with the absolute risks posed by COVID-19, but their interpretation of the absolute risks of those individuals changed when given some contextual information.

A visual scale with a well-chosen and well-explained maximum point helps give context, but participants in these studies also found that the most useful comparators were the absolute risks faced by individuals of defined risk factors (predominantly age, in the case of COVID-19) covering the full range from very low-risk individuals to very high-risk individuals—in other words, putting numbers against a series of 'personas' that are their natural mental models.

Giving people context in the form of other risks that they might face was deemed slightly less helpful. This may be because people were unclear of the magnitude of those other risks, or because those risks were seen to be qualitatively different in important dimensions [28], although seasonal 'flu risk was an acceptable comparator to some. Similarly, choosing personas that weren't easily imagined ('you if you didn't have these risk factors', or 'an average person of this age') were less helpful than those which could be brought to mind clearly and distinctly (such as those which defined both age and health conditions).

## 8.6. Limitations

All mock-ups of results shown to participants in this study were hypothetical, and participants were asked to imagine receiving that result in real life. This removes both the prior beliefs that that participant may have had about their own risk (making it impossible to assess the effects of those prior beliefs and potential conflict with the information being communicated), and the emotional component of receiving a result relating to one's own mortality.

Before implementing any such personalized risk communication it would be important to test the proposed format on participants receiving their real results, with appropriate ethical permissions and support in place.

Our experiments also only covered the communication of risks between 0.01% and 20%. Depending on the absolute risks being communicated (e.g. whether it is 'the risk of dying from COVID-19 if you catch the virus' or 'the risk of catching and then dying as a result of the virus'), the absolute risks for many people could be much lower than this (especially at times of low prevalence of the virus). Further research would be necessary to extend this work to lower percentages where different visual and numerical formats may be required.

Ethics. The study was approved by the Psychological Research Ethics Committee at the University of Cambridge (PRE.2020.070). The letter of approval is saved in the OSF project.

Data accessibility. The questionnaires and data for the surveys are available at: https://osf.io/auf8h/.

Authors' contributions. A.L.J.F. conceived of the study and drafted the manuscript; J.K., G.R., C.R.S. and S.D. collected and analysed the quantitative surveys and wrote the relevant parts of the manuscript; L.F., A.C.E.L. and G.L. carried out and analysed the interviews and wrote the relevant parts of the manuscript; D.S. oversaw and advised the team as well as assisting with manuscript writing.

Competing interests. We declare we have no competing interests.

Funding. This work was funded by the Winton Centre for Risk & Evidence Communication at the University of Cambridge, which is financed by a donation from the David & Claudia Harding Foundation.

Acknowledgements. Interviews with some participants were additionally carried out by Ilan Goodman and María Climént Palmer. We would like to thank all the participants in this study who generously gave large amounts of time and thoughtful responses, and all those who were involved in the administration of the survey at Respondi.

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
