## [Peer Review File · Royal Society Open Science]

Review History

RSOS-201721.R0 (Original submission)

Review form: Reviewer 1

Is the manuscript scientifically sound in its present form?

No

Are the interpretations and conclusions justified by the results?

No

Is the language acceptable?

No

Do you have any ethical concerns with this paper?

No

Have you any concerns about statistical analyses in this paper?

No

Recommendation?

Major revision is needed (please make suggestions in comments)

Comments to the Author(s)

The authors of the present paper conducted an ambitious set of qualitative and quantitative studies to determine how best to communicate COVID-19's risk of death. The abstract was terrific and I much looked forward to reading and learning from the paper. The studies seemed well thought through although the methods were not always clear; more on this below. The paper, however, contains so much that it has an "all but the kitchen sink" feel to it. Overall, the quantity of different ideas, studies, combined with less-than-ideal organization and explanation, and results diluted the possible impact of each study. This is unfortunate because there do seem to be some very good ideas and data here.

Aim of the paper. The authors briefly discuss the aim of communication messages. The aim of their own paper, however, was unclear. Where did they land in terms of aims of being informative, trustworthy, or persuasive? Or perhaps they wanted to recognize the importance of all of these and allow their results (both qualitative and quantitative) to guide their ultimate recommendations. The authors should clarify the aim of the paper.

Lack of hypotheses. What hypotheses were the authors testing? These were never clearly presented. Nor were they justified clearly from the literature. Instead, the experiments mostly seemed to spring from the authors' imaginations. That's not quite fair as they did discuss frequency vs. percentage formats, for example, and the importance of visualizations, albeit in a fairly superficial way.

Lack of grounding in the literature. The authors investigated a number of interesting questions, but the paper's brevity did not allow for justification/explanation of what they tested and why. This lack of grounding interacts with the lack of hypotheses.

Great multi-method studies. I applaud the authors for their use of qualitative research as a grounding for their quantitative research. As researchers, we should do more of this. However, the research methods description was simply too sparse to appreciate the 6 experiments. There was very little explanation in the main text. And, as mentioned, below, the combination of main text and appendices lacked both organization and explanation.

Lack of organization and explanation. The methods simply did not contain enough information to evaluate (or understand) what was done in each study. As one example (but there are many), the authors mention a power calculation for Survey 4 based on the largest of 3 experiments. However, they did not define "largest;" I think they meant the largest number of conditions. Another example under Survey 2 is that at first I thought 2.1 and 2.2 represented question types and only realized later that they were experiments both conducted as part of Survey 2. In addition, in their Experiment 2.1: What is a result "concordant [or discordant] with the numerical risk"? Too many other examples exist to mention. Questions (and designs) are provided in the Appendices. But those appendices are not organized well enough to be able to follow from the main text to the methods in the appendix and back. Just as one example, Appendix 2 contains much of the experimental materials, but the experiments were not presented in order, different labels were used for questions in the main text and the appendix, graphics that were presumably used in the materials did not appear in the appendix (which promised full detailed methods), and experimental designs were difficult to follow. The designs were also complicated. Some methods showed up in the results section (e.g., bottom of page 9 to top of page 10). The qualitative results described in the text should also remind the reader of sample size (e.g., colour of scale on page 17 that also included recommendations in this results section; generally, I'd say this should be moved to the discussion section, but given that it's based on such a tiny sample size, any

recommendation should be deleted instead). I appreciated the attempt at brevity, but readers should be able to find and understand experimental designs and methods, including specific questions, easily in the appendices.

This one paper could be split into multiple papers. Greater clarity would result if the authors split this one paper into more papers. Results could be presented separately for experiments on similar topics (this is complicated by the complex designs, but is something the authors should consider carefully). The qualitative results could potentially be their own paper too with other papers referring to it. Separating topics out into different papers would allow the authors to explain their experiments place in the literature, expand on methods so that they're understandable, and increase the impact of their work.

Experiment 4.1 (a 5×4 factorial design itself!) seemed the most novel and important although, given the quantity of numbers and text, I wondered how well less numerate participants understood it.

Minor:

The Appendices should be renumbered in order of their appearance in the text. Currently, the order is something like Appendix 1, 2, 7, 3, 6,...

What did the authors consider their "numeracy" scale? In the full paragraph on page 6, they measure the Berlin numeracy test, but also 4 other numeracy items from two other numeracy scales.

Figure captions should include the experiment number (e.g., Experiment 4.1) and its sample size.

On page 17 "comparators that involved participants mentally moving away from their risk" was unclear and needs further explanation.

Given the large number of experiments and their complex designs, the authors should identify from where they derived their conclusions in the Discussion.

Review form: Reviewer 2

Is the manuscript scientifically sound in its present form?

No

Are the interpretations and conclusions justified by the results?

No

Is the language acceptable?

No

Do you have any ethical concerns with this paper?

No

Have you any concerns about statistical analyses in this paper?

No

Recommendation?

Major revision is needed (please make suggestions in comments)

Comments to the Author(s)

Review of “Communicating personalised risks from COVID-19: guidelines from an empirical study”

The manuscript has a clear applied focus and contains a lot of important information. However, I feel the current structure of the manuscript is not as clear as it could be. For example, it is unclear to what extent the interviews with the physicians have informed the four studies, whether each of the studies has informed the following one and if so how?

Further, I find the method section hard to follow. Many relevant information are not mentioned or only found in the Appendix, the latter meaning that the reader needs to constantly jump back and forth. I believe it would be clearer if the materials used in each study were described in more detail in the Method section, alongside the most important visualisation and other information presented to the participants. Also, some example items from the other scales would be useful (e.g., numeracy test, health literacy). Finally, more information on the interviews with the physicians would be useful in the Method section. One useful solution to structure the manuscript better seems to me to follow the layout of typical multi-study articles: Present the five studies separately, each one with a separate Method and Result section (and potentially brief Introduction and Discussion).

Currently, it clearly looks like the authors themselves were overwhelmed with the information: The first figure of the manuscript is labelled “Figure 7”, the Appendix does not provide sufficient information to understand the method, there is a Figure 10 in the Result section and one in the Appendix (the latter could be labelled “Figure A10”), the results of Experiment 4 are reported before those of Experiment 3, the page numbering is odd, and the formatting of the result section is all over the place (e.g., missing equal signs, inconsistent labelling of effect sizes; see below), just to name a few things. This makes it very hard to assess it and I stopped reading halfway through the Result section as I could not follow them, making it also pointless to read the Discussion.

Below are some relatively minor comments, listed in order of appearance:

- 1) Introduction: It might be worth to link research from (medical) risk communication with related recent research in psychology (e.g., Grice et al., 2020; Hanel & Mehler, 2019), to connect those two lines of research better; they seemed to have developed somewhat independently.
- 2) Survey 1: “the number of participants (500) was based on our previous experience of these types of surveys” – this sounds like a good opportunity for some justified self-citations. On a side note, I agree that 500 participants are sufficient.
- 3) It is unclear where the effect size used for the power calculations of $d = 0.30$ is coming from.
- 4) Page 7: “Around half of the healthcare providers interviewed” or “participants rarely used numbers” – here and elsewhere: why not be more specific (e.g., “X out of Y healthcare providers interviewed...”) ? Absolute frequencies are surely more informative than approximations? Absolute frequencies are reported later on in the Results (see below) – another example for inconsistent writing.
- 5) Page 7f: “As suggested in previous literature, frequencies appeared to represent ‘higher risks’ to people than percentages (and ‘x out of 1000’ higher than ‘x out of 100’) - in this case shown by participants who were asked to rate the chance of death in percentage points inputting a higher fatality rate than those who were asked to rate the chance as a frequency (see Figure 7).” The first part of the sentence seems to contradict the second one?
- 6) Figure 7 and other Figures: There are many issues with bars- and linegraphs (e.g., Weissgerber et al., 2015, 2019). A plot containing the raw data or at least some sort of density distribution (e.g., violinplot) together with a boxplot and a confidence interval would be more informative. On a side note, it is unclear whether the error bars are SEs or CIs.
- 7) Page 8: “As per our pre-registered analysis we first conducted...” – unclear to which study/experiment this is referring to. Reporting the methods and results for all studies separately would solve this issue.

- 8) Result section: Some “=”s are missing, e.g., “Fs(2, 2260) 69.55-175.30”.
- 9) Result section: It is unclear whether the authors are reporting generalized eta-squares or eta-squares, as the subscript “G” is missing for most but not all “eta-squares”.
- 10) Result section: If only two groups are compared I would suggest that the authors also report an effect size, perhaps one based on percentages to be consistent with the overall message of the manuscript (e.g., Cohen’s U3, Common Language Effect Size, Overlapping Coefficient; see Grice et al., 2020; Hanel & Mehler, 2019). Occasionally, Cohen’s d is reported, but this is not an effect size that many people can correctly interpret.
- 11) Figure 11: The x-axis labels are odd: Lowest (5) – Highest (1). Shouldn’t 1 and 5 be exchanged?
- 12) Page 18: “but 2 of the 6 (33%)” It is unclear to me why there are suddenly only 6 participants? Also, 2 vs 4 is not a significant difference (Pearson’s chi-square test).

Overall, very interesting data that are in my view not clearly presented.

References

- Grice, J. W., Medellin, E., Jones, I., Horvath, S., McDaniel, H., O’lansen, C., & Baker, M. (2020). Persons as effect sizes. *Advances in Methods and Practices in Psychological Science*, 2515245920922982. <https://doi.org/10.1177/2515245920922982>
- Hanel, P. H. P., & Mehler, D. M. (2019). Beyond reporting statistical significance: Identifying informative effect sizes to improve scientific communication: *Public Understanding of Science*, 28(4), 468–485. <https://doi.org/10.1177/0963662519834193>
- Weissgerber, T. L., Milic, N. M., Winham, S. J., & Garovic, V. D. (2015). Beyond bar and line graphs: Time for a new data presentation paradigm. *PLoS Biol*, 13(4), e1002128. <https://doi.org/10.1371/journal.pbio.1002128>
- Weissgerber, T. L., Winham, S. J., Heinzen, E. P., Milin-Lazovic, J. S., Garcia-Valencia, O., Bukumiric, Z., Savic, M. D., Garovic, V. D., & Milic, N. M. (2019). Reveal, don’t conceal: Transforming data visualization to improve transparency. *Circulation*. <https://www.ahajournals.org/doi/abs/10.1161/CIRCULATIONAHA.118.037777>

Decision letter (RSOS-201721.R0)

Dear Dr Freeman

The Editors assigned to your paper RSOS-201721 "Communicating personalised risks from COVID-19: guidelines from an empirical study" have now received comments from reviewers and would like you to revise the paper in accordance with the reviewer comments and any comments from the Editors. Please note this decision does not guarantee eventual acceptance.

We invite you to respond to the comments supplied below and revise your manuscript. Below the referees’ and Editors’ comments (where applicable) we provide additional requirements. Final acceptance of your manuscript is dependent on these requirements being met. We provide guidance below to help you prepare your revision.

Please submit your revised manuscript and required files (see below) no later than 21 days from today’s (ie 04-Jan-2021) date. Note: the ScholarOne system will ‘lock’ if submission of the revision

is attempted 21 or more days after the deadline. If you do not think you will be able to meet this deadline please contact the editorial office immediately.

on behalf of Professor Geoff Haddock (Associate Editor) and Essi Viding (Subject Editor)
openscience@royalsociety.org

Associate Editor Comments to Author (Professor Geoff Haddock):

Associate Editor: 1

Comments to the Author:

Thank you for submitting your paper to RSOS. I have now been able to secure feedback from two reviewers, both of whom have offered comprehensive evaluations of the work. Both reviewers have convergent views of the research; they see it is as highly important and ambitious, but that the current presentation of the paper makes it extremely unwieldy, with the manuscript being very difficult to follow. I share the reviewers' views. This is important and interesting work, but in its present form the take-home messages are difficult to discern. As a result, I wish to offer you the opportunity to revise and resubmit the paper.

For the sake of brevity, I will not repeat all of the reviewers' points. As noted above, an overarching theme is how to present the research in a manner that provides the reader a sufficiently clear and manageable message. As I see it, there are different paths to this goal. First, I believe it is best for the paper to follow the typical format of a multi-study paper, where each study is presented in a sequential manner, with each study building upon the earlier ones. This view is echoed by both of the reviewers. This would have the likely effect of lengthening the paper, but would allow it to be digested in a manageable way. If this re-organization results in a paper that is simply too big, one reviewer suggested that you consider scaling back the number of studies presented in the current manuscript, allowing the studies to be discussed in different papers with a particular theme. This is something that you might wish to consider; I see value in this possibility. Regardless, re-structuring would also have an impact on the presentation of the results and discussion.

As part of this theme about clarifying the message, the reviewers and I share the view that the introduction would benefit from greater detail/clarity regarding the paper's aims, its grounding in the literature, and the justification for the hypotheses. Similarly, the description of the methods requires greater elaboration; like the reviewers I found that the lack of detail sometimes made it difficult to evaluate what was done in the individual studies.

In addition to this overarching structural revision, the reviewers raise a series of additional points about issues such as the presentation of the results that would need to be addressed in a revision.

In sum, there is much to admire about the work – it is ambitious and important. However, major revisions are required to further the paper’s clarity. Should you revise and resubmit the paper, please include a cover letter that how you addressed all of the comments made in this decision and the reviewers’ comments.

Once again, thank you for submitting your work to RSOS.

Reviewer comments to Author:

Reviewer: 1

Comments to the Author(s)

The authors of the present paper conducted an ambitious set of qualitative and quantitative studies to determine how best to communicate COVID-19’s risk of death. The abstract was terrific and I much looked forward to reading and learning from the paper. The studies seemed well thought through although the methods were not always clear; more on this below. The paper, however, contains so much that it has an “all but the kitchen sink” feel to it. Overall, the quantity of different ideas, studies, combined with less-than-ideal organization and explanation, and results diluted the possible impact of each study. This is unfortunate because there do seem to be some very good ideas and data here.

Aim of the paper. The authors briefly discuss the aim of communication messages. The aim of their own paper, however, was unclear. Where did they land in terms of aims of being informative, trustworthy, or persuasive? Or perhaps they wanted to recognize the importance of all of these and allow their results (both qualitative and quantitative) to guide their ultimate recommendations. The authors should clarify the aim of the paper.

Lack of hypotheses. What hypotheses were the authors testing? These were never clearly presented. Nor were they justified clearly from the literature. Instead, the experiments mostly seemed to spring from the authors’ imaginations. That’s not quite fair as they did discuss frequency vs. percentage formats, for example, and the importance of visualizations, albeit in a fairly superficial way.

Lack of grounding in the literature. The authors investigated a number of interesting questions, but the paper’s brevity did not allow for justification/explanation of what they tested and why. This lack of grounding interacts with the lack of hypotheses.

Great multi-method studies. I applaud the authors for their use of qualitative research as a grounding for their quantitative research. As researchers, we should do more of this. However, the research methods description was simply too sparse to appreciate the 6 experiments. There was very little explanation in the main text. And, as mentioned, below, the combination of main text and appendices lacked both organization and explanation.

Lack of organization and explanation. The methods simply did not contain enough information to evaluate (or understand) what was done in each study. As one example (but there are many), the authors mention a power calculation for Survey 4 based on the largest of 3 experiments. However, they did not define “largest;” I think they meant the largest number of conditions. Another example under Survey 2 is that at first I thought 2.1 and 2.2 represented question types and only realized later that they were experiments both conducted as part of Survey 2. In addition, in their Experiment 2.1: What is a result “concordant [or discordant] with the numerical risk”? Too many other examples exist to mention. Questions (and designs) are provided in the

Appendices. But those appendices are not organized well enough to be able to follow from the main text to the methods in the appendix and back. Just as one example, Appendix 2 contains much of the experimental materials, but the experiments were not presented in order, different labels were used for questions in the main text and the appendix, graphics that were presumably used in the materials did not appear in the appendix (which promised full detailed methods), and experimental designs were difficult to follow. The designs were also complicated. Some methods showed up in the results section (e.g., bottom of page 9 to top of page 10). The qualitative results described in the text should also remind the reader of sample size (e.g., colour of scale on page 17 that also included recommendations in this results section; generally, I'd say this should be moved to the discussion section, but given that it's based on such a tiny sample size, any recommendation should be deleted instead). I appreciated the attempt at brevity, but readers should be able to find and understand experimental designs and methods, including specific questions, easily in the appendices.

This one paper could be split into multiple papers. Greater clarity would result if the authors split this one paper into more papers. Results could be presented separately for experiments on similar topics (this is complicated by the complex designs, but is something the authors should consider carefully). The qualitative results could potentially be their own paper too with other papers referring to it. Separating topics out into different papers would allow the authors to explain their experiments place in the literature, expand on methods so that they're understandable, and increase the impact of their work.

Experiment 4.1 (a 5 x 4 factorial design itself!) seemed the most novel and important although, given the quantity of numbers and text, I wondered how well less numerate participants understood it.

Minor:

The Appendices should be renumbered in order of their appearance in the text. Currently, the order is something like Appendix 1, 2, 7, 3, 6,...

What did the authors consider their "numeracy" scale? In the full paragraph on page 6, they measure the Berlin numeracy test, but also 4 other numeracy items from two other numeracy scales.

Figure captions should include the experiment number (e.g., Experiment 4.1) and its sample size.

On page 17 "comparators that involved participants mentally moving away from their risk" was unclear and needs further explanation.

Given the large number of experiments and their complex designs, the authors should identify from where they derived their conclusions in the Discussion.

Reviewer: 2

Comments to the Author(s)

Review of "Communicating personalised risks from COVID-19: guidelines from an empirical study"

The manuscript has a clear applied focus and contains a lot of important information. However, I feel the current structure of the manuscript is not as clear as it could be. For example, it is unclear to what extent the interviews with the physicians have informed the four studies, whether each of the studies has informed the following one and if so how?

Further, I find the method section hard to follow. Many relevant information are not mentioned or only found in the Appendix, the latter meaning that the reader needs to constantly jump back and forth. I believe it would be clearer if the materials used in each study were described in more detail in the Method section, alongside the most important visualisation and other information presented to the participants. Also, some example items from the other scales would be useful (e.g., numeracy test, health literacy). Finally, more information on the interviews with the physicians would be useful in the Method section. One useful solution to structure the manuscript better seems to me to follow the layout of typical multi-study articles: Present the five studies separately, each one with a separate Method and Result section (and potentially brief Introduction and Discussion).

Currently, it clearly looks like the authors themselves were overwhelmed with the information: The first figure of the manuscript is labelled "Figure 7", the Appendix does not provide sufficient information to understand the method, there is a Figure 10 in the Result section and one in the Appendix (the latter could be labelled "Figure A10"), the results of Experiment 4 are reported before those of Experiment 3, the page numbering is odd, and the formatting of the result section is all over the place (e.g., missing equal signs, inconsistent labelling of effect sizes; see below), just to name a few things. This makes it very hard to assess it and I stopped reading halfway through the Result section as I could not follow them, making it also pointless to read the Discussion.

Below are some relatively minor comments, listed in order of appearance:

- 1) Introduction: It might be worth to link research from (medical) risk communication with related recent research in psychology (e.g., Grice et al., 2020; Hanel & Mehler, 2019), to connect those two lines of research better; they seemed to have developed somewhat independently.
- 2) Survey 1: "the number of participants (500) was based on our previous experience of these types of surveys" – this sounds like a good opportunity for some justified self-citations. On a side note, I agree that 500 participants are sufficient.
- 3) It is unclear where the effect size used for the power calculations of $d = 0.30$ is coming from.
- 4) Page 7: "Around half of the healthcare providers interviewed" or "participants rarely used numbers" – here and elsewhere: why not be more specific (e.g., "X out of Y healthcare providers interviewed...")? Absolute frequencies are surely more informative than approximations? Absolute frequencies are reported later on in the Results (see below) – another example for inconsistent writing.
- 5) Page 7f: "As suggested in previous literature, frequencies appeared to represent 'higher risks' to people than percentages (and 'x out of 1000' higher than 'x out of 100') - in this case shown by participants who were asked to rate the chance of death in percentage points inputting a higher fatality rate than those who were asked to rate the chance as a frequency (see Figure 7)." The first part of the sentence seems to contradict the second one?
- 6) Figure 7 and other Figures: There are many issues with bars- and linegraphs (e.g., Weissgerber et al., 2015, 2019). A plot containing the raw data or at least some sort of density distribution (e.g., violinplot) together with a boxplot and a confidence interval would be more informative. On a side note, it is unclear whether the error bars are SEs or CIs.
- 7) Page 8: "As per our pre-registered analysis we first conducted..." – unclear to which study/experiment this is referring to. Reporting the methods and results for all studies separately would solve this issue.
- 8) Result section: Some "="s are missing, e.g., "Fs(2, 2260) 69.55-175.30".
- 9) Result section: It is unclear whether the authors are reporting generalized eta-squares or eta-squares, as the subscript "G" is missing for most but not all "eta-squares".
- 10) Result section: If only two groups are compared I would suggest that the authors also report an effect size, perhaps one based on percentages to be consistent with the overall message of the manuscript (e.g., Cohen's U_3 , Common Language Effect Size, Overlapping Coefficient; see Grice et al., 2020; Hanel & Mehler, 2019). Occasionally, Cohen's d is reported, but this is not an effect size that many people can correctly interpret.
- 11) Figure 11: The x-axis labels are odd: Lowest (5) – Highest (1). Shouldn't 1 and 5 be exchanged?

12) Page 18: “but 2 of the 6 (33%)” It is unclear to me why there are suddenly only 6 participants? Also, 2 vs 4 is not a significant difference (Pearson’s chi-square test).

Overall, very interesting data that are in my view not clearly presented.

References

- Grice, J. W., Medellin, E., Jones, I., Horvath, S., McDaniel, H., O’lansen, C., & Baker, M. (2020). Persons as effect sizes. *Advances in Methods and Practices in Psychological Science*, 2515245920922982. <https://doi.org/10.1177/2515245920922982>
- Hanel, P. H. P., & Mehler, D. M. (2019). Beyond reporting statistical significance: Identifying informative effect sizes to improve scientific communication: *Public Understanding of Science*, 28(4), 468–485. <https://doi.org/10.1177/0963662519834193>
- Weissgerber, T. L., Milic, N. M., Winham, S. J., & Garovic, V. D. (2015). Beyond bar and line graphs: Time for a new data presentation paradigm. *PLoS Biol*, 13(4), e1002128. <https://doi.org/10.1371/journal.pbio.1002128>
- Weissgerber, T. L., Winham, S. J., Heinzen, E. P., Milin-Lazovic, J. S., Garcia-Valencia, O., Bukumiric, Z., Savic, M. D., Garovic, V. D., & Milic, N. M. (2019). Reveal, don’t conceal: Transforming data visualization to improve transparency. *Circulation*. <https://www.ahajournals.org/doi/abs/10.1161/CIRCULATIONAHA.118.037777>

===PREPARING YOUR MANUSCRIPT===

===PREPARING YOUR REVISION IN SCHOLARONE===

Author's Response to Decision Letter for (RSOS-201721.R0)

See Appendix A.

RSOS-201721.R1 (Revision)

Review form: Reviewer 2

Is the manuscript scientifically sound in its present form?

Yes

Are the interpretations and conclusions justified by the results?

Yes

Is the language acceptable?

Yes

Do you have any ethical concerns with this paper?

No

Have you any concerns about statistical analyses in this paper?

No

Recommendation?

Accept with minor revision (please list in comments)

Comments to the Author(s)

The revised version of the manuscript has improved a lot. The structure is now much clearer, and the manuscript gives more justice to the impressive dataset. I only have a couple of minor comments.

- 1) The authors use several undefined terms in the Introduction that might be unclear to some readers. For example, Paling Perspective Scale, or dose-response. Perhaps add brief descriptions.
- 2) There are several formatting (?) issues throughout the manuscript "Error! Reference source not found." I believe some were referring to Tables and Appendices which I therefore couldn't assess.
- 3) Experiment 2.2: It is unclear to me why only a logarithmic scale was used, after participants in Survey 1 had criticised it as difficult to understand?
- 4) Experiment 2.2: I find the last three DVs confusing. In the mock-up, the wording directly addresses the participants "Your estimated risk of dying..." The items then read "If the person who got this result caught COVID-19, how likely do you think it is that they would die as a result?" and "If this result applied to you, how worried would you be?" In other words, the mock-up reads as they display information about the participants directly, but the items are about a generic person.
- 5) It is great that the authors are now also reporting the Common Language Effect Size. Since this effect size is (unfortunately) not very common, it might be worth providing a brief definition of it or 'spelling it out' as done here <https://rpsychologist.com/d3/cohend/>.

- 6) Experiment 2.1: In the Method section, Experiment 2.1 is described as a 4x2-design. At the beginning of the Result section as a 5x4x2-design.
- 7) Surveys 3 & 4, power calculations: “ $d = 0.3$, equivalent to $CLES = 0.55$ ”. A $d = 0.30$ is equivalent to $CLES = .58$, not $.55$.
- 8) Survey 4: “we decided to run one experiment specifically to test the risk perception of each of four different formats.” – to which four formats are you referring to?
- 9) Survey 4, result section: There are still a couple of “eta-squares”, rather than “generalised eta-squares” reported. There are also many Cohen’s d s that are not accompanied by CLES. I believe that reporting CLES in the abstract, results, and discussion sections is more important because lay-people are more likely to read the abstract or discussion than the power calculation. In this section there are some other minor typos such as missing degrees of freedoms (e.g., “ $F=59.8$ ”), including for correlation coefficients.
- 10) Survey 4, results: “the goal of the communication is usually...” The page number of the quote is missing.
- 11) Figure 21 could also be a violinplot with CIs.
- 12) Survey 4, interim discussion: Another reason why the negatively framed scale was associated with better comprehension than the positively framed scale is that people tend to focus more on negative-stimuli. There is some research on this in the cognitive and clinical psychological literature.
- 13) Overall, most effects were small or even smaller than small and only reached statistical significance because of the large sample sizes. This is not an issue of course, as small effects can have, on a country-level, a large impact and are in line with similar research (e.g., the <http://www.vizhealth.org/gallery/>). However, I feel this should be acknowledged.
- 14) Great that the authors made their rich datasets openly available!

Decision letter (RSOS-201721.R1)

Dear Dr Freeman

On behalf of the Editors, we are pleased to inform you that your Manuscript RSOS-201721.R1 "Communicating personalised risks from COVID-19: guidelines from an empirical study" has been accepted for publication in Royal Society Open Science subject to minor revision in accordance with the referees' reports. Please find the referees' comments along with any feedback from the Editors below my signature.

Please submit your revised manuscript and required files (see below) no later than 7 days from today's (ie 29-Mar-2021) date. Note: the ScholarOne system will 'lock' if submission of the revision is attempted 7 or more days after the deadline. If you do not think you will be able to meet this deadline please contact the editorial office immediately.

Please note article processing charges apply to papers accepted for publication in Royal Society Open Science (<https://royalsocietypublishing.org/rsos/charges>). Charges will also apply to

papers transferred to the journal from other Royal Society Publishing journals, as well as papers submitted as part of our collaboration with the Royal Society of Chemistry (<https://royalsocietypublishing.org/rsos/chemistry>). Fee waivers are available but must be requested when you submit your revision (<https://royalsocietypublishing.org/rsos/waivers>).

on behalf of Professor Geoff Haddock (Associate Editor) and Essi Viding (Subject Editor)
openscience@royalsociety.org

Associate Editor Comments to Author (Professor Geoff Haddock):

Thank you for sending your revised manuscript to RSOS. Given the scope of the revisions I sent this version of the paper to one of the original reviewers, who kindly agreed to read and comment upon it. Further, I read the revised version of the paper before and after having received the reviewer's comments. I very much appreciate how you have re-framed much of the paper; this has really enhanced its readability. There is still a lot of information to digest, but there is a substantially improved flow to the paper. As a result, I am happy to accept the paper pending some minor (and presumably final) revisions. The revisions are listed below. These are minor and should not require too much time. Given the nature of the topic, it would be nice to finalise these revisions as soon as possible.

Within the introduction's section of "Aim of communication", I wondered if the two paragraphs beginning with "The first approach ..." and "The second approach" could be merged into one paragraph. It might be a matter of taste, and I'll leave it to you should you wish (or not) to make that revision.

I wondered if Table 2 is better placed within a supplementary file, rather than the main text. Similarly, the appendices as can included within a supplemental file.

Most substantively, some of the effects are very small. This should be acknowledged more directly in the general discussion, perhaps within the section of limitations. While noting that the some of the effects are small, I would suggest highlighting that in the context of the pandemic these effects are still very impactful at a societal level. Please note that the reviewer makes a similar concern.

In addition, the reviewer has kindly noted a series of small issues to address.

Please send a revised version of the paper, along with a cover letter noting the revisions.

I hope this is helpful, I look forward to hearing back from you.

Reviewer comments to Author:

Reviewer: 2

Comments to the Author(s)

The revised version of the manuscript has improved a lot. The structure is now much clearer, and the manuscript gives more justice to the impressive dataset. I only have a couple of minor comments.

- 1) The authors use several undefined terms in the Introduction that might be unclear to some readers. For example, Paling Perspective Scale, or dose-response. Perhaps add brief descriptions.
- 2) There are several formatting (?) issues throughout the manuscript "Error! Reference source not found." I believe some were referring to Tables and Appendices which I therefore couldn't assess.
- 3) Experiment 2.2: It is unclear to me why only a logarithmic scale was used, after participants in Survey 1 had criticised it as difficult to understand?
- 4) Experiment 2.2: I find the last three DVs confusing. In the mock-up, the wording directly addresses the participants "Your estimated risk of dying..." The items then read "If the person who got this result caught COVID-19, how likely do you think it is that they would die as a result?" and "If this result applied to you, how worried would you be?" In other words, the mock-up reads as they display information about the participants directly, but the items are about a generic person.
- 5) It is great that the authors are now also reporting the Common Language Effect Size. Since this effect size is (unfortunately) not very common, it might be worth providing a brief definition of it or 'spelling it out' as done here <https://rpsychologist.com/d3/cohend/> .
- 6) Experiment 2.1: In the Method section, Experiment 2.1 is described as a 4x2-design. At the beginning of the Result section as a 5x4x2-design.
- 7) Surveys 3 & 4, power calculations: "d = 0.3, equivalent to CLES = 0.55". A d = 0.30 is equivalent to CLES = .58, not .55.
- 8) Survey 4: "we decided to run one experiment specifically to test the risk perception of each of four different formats." - to which four formats are you referring to?
- 9) Survey 4, result section: There are still a couple of "eta-squares", rather than "generalised eta-squares" reported. There are also many Cohen's ds that are not accompanied by CLES. I believe that reporting CLES in the abstract, results, and discussion sections is more important because lay-people are more likely to read the abstract or discussion than the power calculation. In this section there are some other minor typos such as missing degrees of freedoms (e.g., "F=59.8"), including for correlation coefficients.
- 10) Survey 4, results: "the goal of the communication is usually..." The page number of the quote is missing.
- 11) Figure 21 could also be a violinplot with CIs.
- 12) Survey 4, interim discussion: Another reason why the negatively framed scale was associated with better comprehension than the positively framed scale is that people tend to focus more on negative-stimuli. There is some research on this in the cognitive and clinical psychological literature.
- 13) Overall, most effects were small or even smaller than small and only reached statistical significance because of the large sample sizes. This is not an issue of course, as small effects can have, on a country-level, a large impact and are in line with similar research (e.g., the <http://www.vizhealth.org/gallery/>). However, I feel this should be acknowledged.
- 14) Great that the authors made their rich datasets openly available!

===PREPARING YOUR MANUSCRIPT===

one version identifying all the changes that have been made (for instance, in coloured highlight, in bold text, or tracked changes);
 a 'clean' version of the new manuscript that incorporates the changes made, but does not highlight them. This version will be used for typesetting.

===PREPARING YOUR REVISION IN SCHOLARONE===

- Any electronic supplementary material (ESM).
- If you are requesting a discretionary waiver for the article processing charge, the waiver form must be included at this step.
- If you are providing image files for potential cover images, please upload these at this step, and inform the editorial office you have done so. You must hold the copyright to any image provided.
- A copy of your point-by-point response to referees and Editors. This will expedite the preparation of your proof.

- Ensure that your data access statement meets the requirements at <https://royalsociety.org/journals/authors/author-guidelines/#data>. You should ensure that you cite the dataset in your reference list. If you have deposited data etc in the Dryad repository, please only include the 'For publication' link at this stage. You should remove the 'For review' link.
- If you are requesting an article processing charge waiver, you must select the relevant waiver option (if requesting a discretionary waiver, the form should have been uploaded at Step 3 'File upload' above).
- If you have uploaded ESM files, please ensure you follow the guidance at <https://royalsociety.org/journals/authors/author-guidelines/#supplementary-material> to include a suitable title and informative caption. An example of appropriate titling and captioning may be found at https://figshare.com/articles/Table_S2_from_Is_there_a_trade-off_between_peak_performance_and_performance_breadth_across_temperatures_for_aerobic_scope_in_teleost_fishes_/3843624.

Author's Response to Decision Letter for (RSOS-201721.R1)

See Appendix B.

Decision letter (RSOS-201721.R2)

Dear Dr Freeman,

It is a pleasure to accept your manuscript entitled "Communicating personalised risks from COVID-19: guidelines from an empirical study" in its current form for publication in Royal Society Open Science. The comments of the reviewer(s) who reviewed your manuscript are included at the foot of this letter.

Please ensure that your OSF link is made public during the proofing process.

You can expect to receive a proof of your article in the near future. Please contact the editorial office (openscience@royalsociety.org) and the production office (openscience_proofs@royalsociety.org) to let us know if you are likely to be away from e-mail contact – if you are going to be away, please nominate a co-author (if available) to manage the proofing process, and ensure they are copied into your email to the journal.

on behalf of Professor Geoff Haddock (Associate Editor) and Essi Viding (Subject Editor)
openscience@royalsociety.org

Associate Editor Comments to Author (Professor Geoff Haddock):
Associate Editor
Comments to the Author:

Thank you for submitting the revised version of your paper; I very much appreciate the attention devoted to the revisions. Overall, the revisions do a very good job of dealing with the points raised by myself and the reviewer. As a result, I am pleased to accept your paper for publication in RSOS. Compliments on this interesting and important piece of work.

There were two typos I found within the latest version. On page 9 (line 55) and page 48 of the PDF, two paragraphs need to be separated. I assume this will be picked up during the editing stage, but I simply wanted to bring it to your attention.

Best of luck with your ongoing work.

Sincerely,
Geoff Haddock
Associate Editor, RSOS

Appendix A

8th February 2021,

Dear Editors,

Thank you very much indeed for your patience as we have revised our manuscript “Communicating personalised risks from COVID-19: guidelines from an empirical study”. We very much appreciated comments from the editor and both reviewers and hope that we have brought the manuscript significantly closer to an acceptable format. In this letter we outline our responses. Since the manuscript has been so significantly restructured we suspect that the ‘marked up’ version will prove almost useless! However, we hope that the clean version will prove an easier read than the last version.

Associate Editor Comments to Author (Professor Geoff Haddock):

Associate Editor: 1

Comments to the Author:

Thank you for submitting your paper to RSOS. I have now been able to secure feedback from two reviewers, both of whom have offered comprehensive evaluations of the work. Both reviewers have convergent views of the research; they see it is as highly important and ambitious, but that the current presentation of the paper makes it extremely unwieldy, with the manuscript being very difficult to follow. I share the reviewers’ views. This is important and interesting work, but in its present form the take-home messages are difficult to discern. As a result, I wish to offer you the opportunity to revise and resubmit the paper.

For the sake of brevity, I will not repeat all of the reviewers’ points. As noted above, an overarching theme is how to present the research in a manner that provides the reader a sufficiently clear and manageable message. As I see it, there are different paths to this goal. First, I believe it is best for the paper to follow the typical format of a multi-study paper, where each study is presented in a sequential manner, with each study building upon the earlier ones. This view is echoed by both of the reviewers. This would have the likely effect of lengthening the paper, but would allow it to be digested in a manageable way. If this re-organization results in a paper that is simply too big, one reviewer suggested that you consider scaling back the number of studies presented in the current manuscript, allowing the studies to be discussed in different papers with a particular theme. This is something that you might wish to consider; I see value in this possibility. Regardless, restructuring would also have an impact on the presentation of the results and discussion.

As part of this theme about clarifying the message, the reviewers and I share the view that the introduction would benefit from greater detail/clarity regarding the paper’s aims, its grounding in the literature, and the justification for the hypotheses. Similarly, the description of the methods requires greater elaboration; like the reviewers I found that the lack of detail sometimes made it difficult to evaluate what was done in the individual studies.

In addition to this overarching structural revision, the reviewers raise a series of additional points about issues such as the presentation of the results that would need to be addressed in a revision.

In sum, there is much to admire about the work – it is ambitious and important. However, major revisions are required to further the paper’s clarity. Should you revise and resubmit the paper, please include a cover letter that how you addressed all of the comments made in this decision and the reviewers’ comments.

Once again, thank you for submitting your work to RSOS.

Reviewer comments to Author:

Reviewer: 1

Comments to the Author(s)

The authors of the present paper conducted an ambitious set of qualitative and quantitative studies to determine how best to communicate COVID-19’s risk of death. The abstract was terrific and I much looked forward to reading and learning from the paper. The studies seemed well thought through although the methods were not always clear; more on this below. The paper, however, contains so much that it has an “all but the kitchen sink” feel to it. Overall, the quantity of different ideas, studies, combined with less-than-ideal organization and explanation, and results diluted the possible impact of each study. This is unfortunate because there do seem to be some very good ideas and data here.

Thank you for your kind comments about the abstract and studies overall. Emboldened by the comments of reviewers and editor, we have lengthened the paper by including a full introduction and methods and hope that this makes our aims and the background to the work clearer. In our rush to share the resulting findings and guidelines with practitioners that we knew were working on such communications (and were less interested in the background to and theoretical grounding for the studies) we hadn’t put as much effort into these sections as we should have, and apologise. It was also difficult to know how best to structure the paper but given that practitioners are probably only going to read the final conclusions you are right that we should structure the rest of the paper in a more traditional study-by-study way to show the logical flow and development. Hopefully this has not resulted in a paper that is too unwieldy, as we do feel that keeping all the information together is probably best.

Aim of the paper. The authors briefly discuss the aim of communication messages. The aim of their own paper, however, was unclear. Where did they land in terms of aims of being informative, trustworthy, or persuasive? Or perhaps they wanted to recognize the importance of all of these and allow their results (both qualitative and quantitative) to guide their ultimate recommendations. The authors should clarify the aim of the paper.

We have substantially added to the Introduction section, hopefully explicitly introducing our aim (“*Our study was carried out during the COVID-19 pandemic with the*

aim of providing information in real time to communicators who were producing personalised COVID-19 risk calculators, so our methods involved some pragmatic design choices. We ran a series of qualitative interviews, first with members of the public and then with primary care physicians, which fed into design choices throughout the process. After the initial rounds of qualitative interviews we started potential designs of the communication and refined these through further interviews and a set of quantitative experiments. Overall we focussed a number of key research questions: "... By addressing this series of research questions and different endpoints, we hoped to be able to produce an evidence-based set of guidelines for practitioners trying to communicate people's individual risk from COVID-19 in such a way as to suit their own aims.")

Lack of hypotheses. What hypotheses were the authors testing? These were never clearly presented. Nor were they justified clearly from the literature. Instead, the experiments mostly seemed to spring from the authors' imaginations. That's not quite fair as they did discuss frequency vs. percentage formats, for example, and the importance of visualizations, albeit in a fairly superficial way.

In the new introduction we set out our five main research questions, giving a brief summary of the relevant literature underlying each; the dependent variables of interest, and state "We outline specific pre-registered hypotheses and analysis plans below as we set out each part of the study." Hopefully this new structure makes the theoretical basis for the studies clearer (although some experiments did spring from imagination, inspired by the qualitative interviews, we have tried to make that process clearer by laying the individual experiments out in order, especially relative to the qualitative research that helped supplement them).

Lack of grounding in the literature. The authors investigated a number of interesting questions, but the paper's brevity did not allow for justification/explanation of what they tested and why. This lack of grounding interacts with the lack of hypotheses.

This should now be addressed by the changes to the Introduction. The literature is, of course, far from an exhaustive review, but should give readers an understanding of the basis for our design decisions.

Great multi-method studies. I applaud the authors for their use of qualitative research as a grounding for their quantitative research. As researchers, we should do more of this. However, the research methods description was simply too sparse to appreciate the 6 experiments. There was very little explanation in the main text. And, as mentioned, below, the combination of main text and appendices lacked both organization and explanation.

Thank you – we appreciate the comment. We have now in the restructure outlined the methods (including for the qualitative work) in more detail in the main text, and

hopefully this makes all the experiments clear enough to follow (we have left the full list of relevant, reported, questions in Appendix 3 for ease of reference).

Lack of organization and explanation. The methods simply did not contain enough information to evaluate (or understand) what was done in each study. As one example (but there are many), the authors mention a power calculation for Survey 4 based on the largest of 3 experiments. However, they did not define “largest;” I think they meant the largest number of conditions.

We very much hope that our reorganisation of the manuscript has helped clarify the methods in enough detail. We have specifically separated the power calculations for each survey by a separate heading within ‘methods’ and given our justifications more clearly.

Another example under Survey 2 is that at first I thought 2.1 and 2.2 represented question types and only realized later that they were experiments both conducted as part of Survey 2. In addition, in their Experiment 2.1: What is a result “concordant [or discordant] with the numerical risk”? Too many other examples exist to mention. Questions (and designs) are provided in the Appendices. But those appendices are not organized well enough to be able to follow from the main text to the methods in the appendix and back. Just as one example, Appendix 2 contains much of the experimental materials, but the experiments were not presented in order, different labels were used for questions in the main text and the appendix, graphics that were presumably used in the materials did not appear in the appendix (which promised full detailed methods), and experimental designs were difficult to follow.

Again, hopefully this is now much clearer. We have moved most of the methods into the main text, using the Appendices only for a tabular list of the main questions asked in each experiment. Appendix 3 is now the Appendix that contains this tabular list. After some consideration, we have decided to leave these in ‘theme’ grouping but within each theme the experiments are now in numerical order. This should provide a useful contrast to the main manuscript where the order of experiments is purely numerical.

The designs were also complicated. Some methods showed up in the results section (e.g., bottom of page 9 to top of page 10).

All methodological information should now be found only in the Methods section for each survey (in the ‘results’ sections there is only a sentence of recap to help readers where deemed necessary).

The qualitative results described in the text should also remind the reader of sample size (e.g., colour of scale on page 17 that also included recommendations in this results section; generally, I’d say this should be moved to the discussion section, but given that it’s based on such a tiny sample size, any recommendation should be deleted instead).

Now all the qualitative results are reported together this should be slightly easier to read. Where we have employed a quantitative content analysis the sample sizes should now always be clear. Where we have employed a purely qualitative descriptive thematic analysis, such a quantitative approach would not be part of the methodology. The results from the two kinds of analysis should now be more clearly demarcated in the layout.

Regarding colour specifically, we do not draw conclusions or make recommendations other than that designers should be aware of the potential importance of colour. As we have now hopefully made clear, the aim of this study was to provide an empirical guide for practitioners approaching the task of reporting results from personalised COVID-19 risk calculators and we found that issue of colour emerged often. Although we could not make direct recommendations as it was not an area we investigated closely, we wanted to draw attention to the possible psychological affects of colour coding which might not otherwise be obvious to a designer.

I appreciated the attempt at brevity, but readers should be able to find and understand experimental designs and methods, including specific questions, easily in the appendices.

We did attempt brevity, and also addressing mainly a practitioner audience who would be less interested in methods, but we appreciate that this approach didn't work and so have attempted now to restructure so that the paper is more appropriate for those interested in the methods as much as the conclusions! Hopefully this has been successful.

This one paper could be split into multiple papers. Greater clarity would result if the authors split this one paper into more papers. Results could be presented separately for experiments on similar topics (this is complicated by the complex designs, but is something the authors should consider carefully). The qualitative results could potentially be their own paper too with other papers referring to it. Separating topics out into different papers would allow the authors to explain their experiments place in the literature, expand on methods so that they're understandable, and increase the impact of their work.

This is a difficult decision. We have attempted in this restructure to keep all the findings together because we believe that they are best seen in the context of what we were attempting: a single, rapid piece of work to provide an empirical guide for communicators in the COVID-19 pandemic. All our design decisions were taken within that context, and also we'd like to ensure that all our conclusions are justified in appearing together. The qualitative work in particular was analysed very quickly and although we could, in the future, do a full thematic analysis and publish that, we felt that reporting the pragmatic (but highly informative) analysis that we did within the scope of this paper was important because it underlay so many of the decisions we took during the design of the quantitative work (and the mock-ups we were evaluating). We hope that, although long, the way we have now laid out the paper makes our rationale clearer and justifies it staying as a single body of work.

Experiment 4.1 (a 5 x 4 factorial design itself!) seemed the most novel and important although, given the quantity of numbers and text, I wondered how well less numerate participants understood it.

This is a fair observation, and we have now emphasised in our introduction to Survey 4 that we attempted to make the mock-up representative of the true level of information that such a tool would have to try to communicate. (In fact, the NHS tool that was based on this work, attempts to convey far more information and numbers!)

Minor:

The Appendices should be renumbered in order of their appearance in the text. Currently, the order is something like Appendix 1, 2, 7, 3, 6,...

Indeed! This should now be sorted out, apart from the fact that Appendix 3 is referred to earlier than Appendix 1. We can renumber Appendix 3 to be number 1 if necessary, but the Appendices are now in the order of the manuscript as a whole.

What did the authors consider their “numeracy” scale? In the full paragraph on page 6, they measure the Berlin numeracy test, but also 4 other numeracy items from two other numeracy scales.

The sources are now clearly referenced, and the questions from each of these are now in Appendix 3.

Figure captions should include the experiment number (e.g., Experiment 4.1) and its sample size.

This is now done.

On page 17 “comparators that involved participants mentally moving away from their risk” was unclear and needs further explanation.

It was indeed. We have now rewritten this as: “One suggestion for context to attempt to make a person’s ‘relative risk’ clear has been to compare an individual’s own risk with that of someone who was ‘like them but without health conditions’ [53]. Participants found these sorts of hypothetical scenarios difficult to imagine.”

Given the large number of experiments and their complex designs, the authors should identify from where they derived their conclusions in the Discussion.

This is now done.

Reviewer: 2

Comments to the Author(s)

Review of “Communicating personalised risks from COVID-19: guidelines from an empirical study”

The manuscript has a clear applied focus and contains a lot of important information. However, I feel the current structure of the manuscript is not as clear as it could be. For example, it is unclear to what extent the interviews with the physicians have informed the four studies, whether each of the studies has informed the following one and if so how?

We have now restructured the manuscript to address this. The interviews with the physicians were carried out later than the initial interviews with members of the public so would not have influenced the design of the earlier surveys. We hope that the new structure makes the flow of information and thought from one stage of the study to the next much clearer.

Further, I find the method section hard to follow. Many relevant information are not mentioned or only found in the Appendix, the latter meaning that the reader needs to constantly jump back and forth. I believe it would be clearer if the materials used in each study were described in more detail in the Method section, alongside the most important visualisation and other information presented to the participants.

In our restructure we have now moved all of the key Methods information, including the visualisations tested, into the main manuscript. Hopefully this makes it easier for the reader.

Also, some example items from the other scales would be useful (e.g., numeracy test, health literacy).

These are now in Appendix 3 and referenced.

Finally, more information on the interviews with the physicians would be useful in the Method section. One useful solution to structure the manuscript better seems to me to follow the layout of typical multi-study articles: Present the five studies separately, each one with a separate Method and Result section (and potentially brief Introduction and Discussion).

We have now followed this structure, as suggested. The key questions asked of the physicians are summarised in Table 2 alongside those asked of the public participants.

Currently, it clearly looks like the authors themselves were overwhelmed with the information: The first figure of the manuscript is labelled “Figure 7”, the Appendix does not provide sufficient information to understand the method, there is a Figure 10 in the Result section and one in the Appendix (the latter could be labelled “Figure A10”), the results of Experiment 4 are reported before those of Experiment 3, the page numbering is odd, and the formatting of the result section is all over the place (e.g., missing equal signs,

inconsistent labelling of effect sizes; see below), just to name a few things. This makes it very hard to assess it and I stopped reading halfway through the Result section as I could not follow them, making it also pointless to read the Discussion.

You're right, we were rather overwhelmed with it! All figure numbers should now be correct. We have moved to a purely temporal flow of studies (previously we thought that it would be better, for a practitioner audience, to report our results by theme rather than by study number). We hope that you now feel able to read to the end!

Below are some relatively minor comments, listed in order of appearance:

1) Introduction: It might be worth to link research from (medical) risk communication with related recent research in psychology (e.g., Grice et al., 2020; Hanel & Mehler, 2019), to connect those two lines of research better; they seemed to have developed somewhat independently.

We agree that risk communication is a field that is approached by several different disciplines and there are definitely some independent lines of research. We hope that our Introduction now better places this manuscript within the relevant literature, which we feel comes from both medical risk communication and psychology, though does not perhaps represent parallel research from the science communication community with which we are less familiar. The particular references mentioned by the reviewer are specifically about verbal communication of statistical concepts around effect size and we didn't feel they were citable in the context of this study (although we are equally passionate about the importance of communicating effect sizes as opposed to merely p-values!)

2) Survey 1: "the number of participants (500) was based on our previous experience of these types of surveys" – this sounds like a good opportunity for some justified self-citations. On a side note, I agree that 500 participants are sufficient.

Thank you. We have now clarified this.

3) It is unclear where the effect size used for the power calculations of $d = 0.30$ is coming from.

We have now added a 'power calculation' section to each of our Methods.

4) Page 7: "Around half of the healthcare providers interviewed" or "participants rarely used numbers" – here and elsewhere: why not be more specific (e.g., "X out of Y healthcare providers interviewed...")? Absolute frequencies are surely more informative than approximations? Absolute frequencies are reported later on in the Results (see below) – another example for inconsistent writing.

We have now restructured these sections which might make the divide clearer. Where we were carrying out a quantitative content analysis, we counted the number of participants who made a specific comment. Where we move into a descriptive thematic analysis, this was a purely qualitative methodology.

5) Page 7f: “As suggested in previous literature, frequencies appeared to represent ‘higher risks’ to people than percentages (and ‘x out of 1000’ higher than ‘x out of 100’) - in this case shown by participants who were asked to rate the chance of death in percentage points inputting a higher fatality rate than those who were asked to rate the chance as a frequency (see Figure 7).” The first part of the sentence seems to contradict the second one?

We have tweaked this sentence to “As suggested in previous literature, frequencies appeared to represent higher risks to people than percentages (and ‘x out of 1000’ higher than ‘x out of 100’) - in this case shown by participants who were asked to rate the chance of death in percentage points inputting a higher fatality rate for each persona presented than those who were asked to rate the chance as a frequency.” To make the logic clearer.

For the same persona (i.e. the same risk), those rating it that risk as a percentage chose a higher number than those rating it as a frequency, showing that they ascribed a higher ‘risk value’ to the frequency.

6) Figure 7 and other Figures: There are many issues with bars- and linegraphs (e.g., Weissgerber et al., 2015, 2019). A plot containing the raw data or at least some sort of density distribution (e.g., violinplot) together with a boxplot and a confidence interval would be more informative. On a side note, it is unclear whether the error bars are SEs or CIs.

We have replotted the figures to ensure that the distributions of the data are represented appropriately in each case, and improved the figure legends to make the error bars explicit in each case.

7) Page 8: “As per our pre-registered analysis we first conducted...” – unclear to which study/experiment this is referring to. Reporting the methods and results for all studies separately would solve this issue.

Hopefully this has indeed been solved by our complete restructure of the manuscript.

8) Result section: Some “=”s are missing, e.g., “ $F_s(2, 2260) 69.55-175.30$ ”.

The particular example highlighted here was not actually missing an =, it is summarizing multiple ANOVA results by saying that the F-values were 69.55 to 175.30 (the context is “ $(F_s(2, 2260) 69.55-175.30, all p < 0.001)$ ”). However, we recognise that this was not a standard way to report multiple ANOVAs, and we believe that we have updated all instances of statistical reporting.

9) Result section: It is unclear whether the authors are reporting generalized eta-squares or eta-squares, as the subscript “G” is missing for most but not all “eta-squares”.

We have now worked through all reporting to ensure that it is all (we hope!) standardised and clearly labelled.

10) Result section: If only two groups are compared I would suggest that the authors also report an effect size, perhaps one based on percentages to be consistent with the overall message of the manuscript (e.g., Cohen's U_3 , Common Language Effect Size, Overlapping Coefficient; see Grice et al., 2020; Hanel & Mehler, 2019). Occasionally, Cohen's d is reported, but this is not an effect size that many people can correctly interpret.

We have now reported effect sizes consistently when reporting significance tests throughout the paper. We have used Cohen's d throughout as this is one of the most widely used effect size measures in psychological research, and one that many psychologists expect to see and have a good feel for due to its frequency of use. However, we can appreciate the arguments that some researchers have made that the field ought to move towards other measures. We therefore have included the Common Language Effect Size throughout as well.

11) Figure 11: The x-axis labels are odd: Lowest (5) – Highest (1). Shouldn't 1 and 5 be exchanged?

Since this was a graph of participants' rankings, '1' was indeed the highest, but to avoid confusion we have remade the figure without numerical labels.

12) Page 18: "but 2 of the 6 (33%)" It is unclear to me why there are suddenly only 6 participants? Also, 2 vs 4 is not a significant difference (Pearson's chi-square test).

We have added the phrase "The flexible nature of semi-structured interviews has many benefits, including enhancing the flow of the interview, however this means that at times not all participants are asked every question in the interview guide" into our interview methods, and the phrase "Of those that were shown the colours..." into the results. Hopefully this makes this clearer. With a quantified content analysis we would not conduct statistical tests (especially on such a small sample size), the numbers are given only to give the reader context to the statements made about the interviews. We do not draw any conclusions about colour usage, only drawn attention to the concern that two of our six interviewees raised.

Overall, very interesting data that are in my view not clearly presented.

Thank you – we hope the changes made to the presentation of the data have improved it.

We very much hope that this manuscript version meets the expectations of the editor and reviewers, and that it has not become too unwieldy in the process.

The authors.

Appendix B

Dear Editor,

Thank you very much indeed for your letter, and the minor revisions. We are delighted that you think the manuscript is now nearly in shape for publication and have appreciated the time that you and all the reviewers have spent getting it to this stage.

Below we respond to the suggestions:

Within the introduction's section of "Aim of communication", I wondered if the two paragraphs beginning with "The first approach ..." and "The second approach" could be merged into one paragraph. It might be a matter of taste, and I'll leave it to you should you wish (or not) to make that revision.

No problem, we have made the change.

I wondered if Table 2 is better placed within a supplementary file, rather than the main text. Similarly, the appendices as can included within a supplemental file.

We have moved Table 2 to a new Appendix. The Appendices were kept in the same document do as to make use of automatic number-linking to help us keep track of changes. Now we are (hopefully) approaching the final version of the manuscript we have switched this feature off and been able to separate the Appendices into a supplementary file.

Most substantively, some of the effects are very small. This should be acknowledged more directly in the general discussion, perhaps within the section of limitations. While noting that the some of the effects are small, I would suggest highlighting that in the context of the pandemic these effects are still very impactful at a societal level. Please note that the reviewer makes a similar concern.

We have now highlighted this in the Conclusions (further comments below where the reviewer makes the same point).

*Reviewer: 2
Comments to the Author(s)*

1) The authors use several undefined terms in the Introduction that might be unclear to some readers. For example, Paling Perspective Scale, or dose-response. Perhaps add brief descriptions.

Thank you. We have added a brief description of each ("a chart designed to try to help put new risks into a context of everyday experiences by plotting a number of 'familiar' risks on a logarithmic scale" and "dose-response consistency (do people facing a higher dose of the hazard perceive greater risk?), hazard-response consistency (do people facing a hazard that

is higher in risk perceive greater risk?), *uniformity in response* (do people exposed to the same level of risk tend to have similar responses to it?)”)

2) There are several formatting (?) issues throughout the manuscript “Error! Reference source not found.” I believe some were referring to Tables and Appendices which I therefore couldn’t assess.

This will be an error with the internal references within the document that we used to keep track of moving sections and tables! We have switched this off now and so it should cause no further issues.

3) Experiment 2.2: It is unclear to me why only a logarithmic scale was used, after participants in Survey 1 had criticised it as difficult to understand?

Thank you for raising this. We have now added the explanation “A logarithmic scale had to be used for this experiment because a realistic illustration of the population distribution of risk was impossible on a linear scale due to the very large number of people with a very low risk.”

4) Experiment 2.2: I find the last three DVs confusing. In the mock-up, the wording directly addresses the participants “Your estimated risk of dying...” The items then read “If the person who got this result caught COVID-19, how likely do you think it is that they would die as a result?” and “If this result applied to you, how worried would you be?” In other words, the mock-up reads as they display information about the participants directly, but the items are about a generic person.

We have now added a paragraph to explain this: “(Although the graphics represented the risk as ‘your estimated risk’, in all experiments for ethical reasons it was heavily emphasised that the representations were completely hypothetical and did not represent the participants’ risk, even though they were giving us information such as their age. Hence the questions were phrased carefully to re-emphasise that the results they were seeing were not related to them personally. Please also note that it was incorrect of us to represent the risk at any point as ‘your risk’ as it is only estimated on the basis of the characteristics entered and thus could never be truly personalised.)”

5) It is great that the authors are now also reporting the Common Language Effect Size. Since this effect size is (unfortunately) not very common, it might be worth providing a brief definition of it or ‘spelling it out’ as done here <https://rpsychologist.com/d3/cohend/>.

We have now added a brief description of what the Common Language Effect Size represents, along with a reference to McGraw & Wong 1992.

6) Experiment 2.1: In the Method section, Experiment 2.1 is described as a 4x2-design. At the beginning of the Result section as a 5x4x2-design.

Thank you for noting this discrepancy. We initially stated the within and between factors separately, which was confusing. We have now added a bracketed sentence giving the

overall design in the methods: “Each participant was shown a set of five hypothetical risk results, one after the other; the order of presentation was randomised.

Participants were also randomised to one of 8 conditions in a 2(format)x4(context) factorial design (resulting in an overall 5(risk level; within subjects)x2(format; between subjects)x4(context; between subjects) mixed design).”

7) Surveys 3 & 4, power calculations: “ $d = 0.3$, equivalent to $CLES = 0.55$ ”. A $d = 0.30$ is equivalent to $CLES = .58$, not .55.

Thanks for catching this – we had been using a different online calculator to convert from d to CLES, but upon digging deeper it appears that the calculator we’d used had been relying on an approximation. In the power calculation sections, we now use <https://rpsychologist.com/cohend/> to convert from d to CLES, which uses the conversion formula of Ruscio (2008).

Ruscio, J. (2008). A probability-based measure of effect size: robustness to base rates and other factors. *Psychological methods*, 13(1), 19–30.

8) Survey 4: “we decided to run one experiment specifically to test the risk perception of each of four different formats.” – to which four formats are you referring to?

Well spotted! We have added the clarification “‘x in 1000’, ‘x in 100’, ‘x%’, ‘1 in x’.”

9) Survey 4, result section: There are still a couple of “eta-squares”, rather than “generalised eta-squares” reported. There are also many Cohen’s ds that are not accompanied by CLES. I believe that reporting CLES in the abstract, results, and discussion sections is more important because lay-people are more likely to read the abstract or discussion than the power calculation. In this section there are some other minor typos such as missing degrees of freedoms (e.g., “ $F=59.8$ ”), including for correlation coefficients.

All remaining Cohen’s d s are now accompanied by CLES. (Note that because the d -to-CLES conversion formula used by <https://rpsychologist.com/cohend/> assumes normality and equal variances, the CLESs that we calculated directly from our data in R may not necessarily be precisely equal to what one would get if entering our Cohen’s d s into <https://rpsychologist.com/cohend/>). We have not been reporting any effect size statistics in the abstract or discussion in order to keep these as streamlined and simple as possible. However, we have now highlighted the fact that effect sizes were small in the abstract and the conclusions (see our response to reviewer comment #13 for the text that we added).

We have also added degrees of freedom to inferential statistics as requested and all ANOVA effect sizes are now correctly reported as generalised eta squared.

10) Survey 4, results: “the goal of the communication is usually...” The page number of the quote is missing.

Thank you – we have added the page reference (p110).

11) Figure 21 could also be a violinplot with CIs.

Thank you for this suggestion, we did recode this figure as violin plot as suggested but found it was not visually informative as there were only 5 possible values the raw data could take (resulting in a very ‘bumpy’ violin plot). Given that the aim of this figure is to descriptively outline the relative rankings of different options, we have decided to leave the simpler, original figure. However, if deemed necessary we can switch this to this a violin plot.

12) Survey 4, interim discussion: Another reason why the negatively framed scale was associated with better comprehension than the positively framed scale is that people tend to focus more on negative-stimuli. There is some research on this in the cognitive and clinical psychological literature.

Thank you for this suggestion. We have added a note on this possible explanation of the observed effects to the interim discussion of survey 4, referencing Baumeister et al.’s (2001) work showing more thorough processing of bad information compared to good information, as well as Kahneman and Tversky’s work (1979, 1991) on loss aversion, showing that people are more sensitive to losses compared to gains, which might explain attentional differences to the negatively framed information versus the positively framed information.

13) Overall, most effects were small or even smaller than small and only reached statistical significance because of the large sample sizes. This is not an issue of course, as small effects can have, on a country-level, a large impact and are in line with similar research (e.g., the <http://www.vizhealth.org/gallery/>). However, I feel this should be acknowledged.

Thank you. We have added the following to the abstract: “We note that observed effect sizes generally were small. However, even small effects are meaningful and relevant when scaled up to population levels.”, and the following to our conclusions: “Many of the effect sizes that we found are small (apart from those around the difference that the different format of number, such as percentages versus ‘1 in X’ makes), as is common for experimental work on risk communication. However, when scaled up to population levels, for instance for mass communications, where tens of thousands (or more) people are forming the audience, small effect sizes can still have an impact on outcomes that is worth considering.”

14) Great that the authors made their rich datasets openly available!

Thank you!

We hope that these revisions and the way that we have now submitted the manuscript is acceptable.